

# Gaussian approximation of dynamic cavity equations for linearly-coupled stochastic dynamics

**Mattia Tarabolo[1⋆] and Luca Dall'Asta[1,2,3†]**

**1** Institute of Condensed Matter Physics and Complex Systems,
Department of Applied Science and Technology, Politecnico di Torino,
C.so Duca degli Abruzzi 24, I-10129 Torino, Italy
**2** Italian Institute for Genomic Medicine (IIGM) and Candiolo Cancer Institute IRCCS,
str. prov. 142, km 3.95, I-10060 Candiolo (TO), Italy
**3** INFN, Turin Via Pietro Giuria 1, I-10125 Turin, Italy

⋆ mattia.tarabolo@polito.it , † luca.dallasta@polito.it

## Abstract

Stochastic dynamics on sparse graphs and disordered systems often lead to complex behaviors characterized by heterogeneity in time and spatial scales, slow relaxation, localization, and aging phenomena. The mathematical tools and approximation techniques required to analyze these complex systems are still under development, posing significant technical challenges and resulting in a reliance on numerical simulations. We introduce a novel computational framework for investigating the dynamics of sparse disordered systems with continuous degrees of freedom. Starting with a graphical model representation of the dynamic partition function for a system of linearly-coupled stochastic differential equations, we use dynamic cavity equations on locally tree-like factor graphs to approximate the stochastic measure. Here, cavity marginals are identified with local functionals of single-site trajectories. Our primary approximation involves a second-order truncation of a small-coupling expansion, leading to a Gaussian form for the cavity marginals. For linear dynamics with additive noise, this method yields a closed set of causal integro-differential equations for cavity versions of one-time and two-time averages. These equations provide an exact dynamical description within the local tree-like approximation, retrieving classical results for the spectral density of sparse random matrices. Global constraints, non-linear forces, and state-dependent noise terms can be addressed using a self-consistent perturbative closure technique. The resulting equations resemble those of dynamical mean-field theory in the mode-coupling approximation used for fully-connected models. However, due to their cavity formulation, the present method can also be applied to ensembles of sparse random graphs and employed as a message-passing algorithm on specific graph instances.

| | |
|---|---|
| Received | 2025-02-13 |
| Accepted | 2025-06-12 |
| Published | 2025-07-17 |

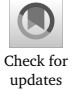

# 1 Introduction

Complex systems across various fields can be mathematically described as collections of random variables interacting through disordered network structures. These networks feature intricate local geometries and a variable number of interactions per degree of freedom, which can also change over time [1]. Despite the lack of finite-dimensional spatial embedding, which makes both equilibrium and non-equilibrium phenomena in these systems essentially mean-field in nature [2,3], they exhibit several unique properties, including anomalous critical exponents [4], spatial localization [5,6], and multiple relaxation scales [7–9]. In addition to numerical simulations, mean-field methods and moment closure techniques, in which local approximations of probability marginals are obtained assuming some level of decorrelation between variables, have been massively applied to investigate the dynamic properties of these systems [10–16].

For a wide range of stochastic processes with discrete degrees of freedom, the dynamic cavity method [17, 18] provides a very accurate description of the dynamics on locally tree-like graphs. However, this method typically involves significant computational complexity, which is reduced only for specific classes of non-recurrent or monotone processes, such as threshold models [19, 20] and epidemic processes [21, 22]. To address this limitation, further approximations of the dynamic cavity method have been introduced, resulting in more efficient algorithms for specific problems [23–27].

In contrast, there has been less progress in the study of stochastic dynamics with continuous degrees of freedom, despite their relevance in various fields such as epidemic diffusion in metapopulations, chemical reactions, and ecological networks. Typically, these cases involve analyzing potentially large systems of coupled (non-linear) stochastic differential equations (SDEs). Common approximation techniques for these systems include linear noise approximation [28, 29], and dimensional reduction techniques [30–33]. However, a long-standing tradition exists in statistical physics for addressing coupled SDEs, in the limit of fully-connected interaction graphs, using path integral mean-field methods [34]. These methods, unified under the general concept of Dynamical Mean Field Theory (DMFT), allow for the derivation of an effective process for a single representative degree of freedom. The representative process is described by a single stochastic differential equation containing memory terms and a colored noise, generated by the interaction of the representative degree of freedom with a self-consistently defined stochastic bath characterized by the very same statistical properties. DMFT has been successfully applied to study the dynamics of spherical p-spin models for aging and glassy dynamics in disordered systems [35–38], and more recently to characterize dynamical phases in neural networks [39, 40], ecological communities [41, 42] and gradient-based high-dimensional learning [43]. To our knowledge, a few works generalize path-integral and DMFT-like approaches beyond fully-connected graphs, most notably using dynamical replica approach [44], large-connectivity approximations [45, 46] and perturbative expansions [47].

In all these applications, DMFT is typically developed to study an effective process obtained after averaging over the disorder induced by the interaction structure. Algorithmic versions of DMFT were also introduced, like in the case of the Extended Plefka Expansion [48, 49], where two-time response/correlation functions are microscopically defined for every site of the system. The resulting equations are a kind of dynamic version of the TAP equations used for studying equilibrium properties of mean-field disordered systems. They provide an exact description of systems of linear SDEs with additive noise in the fully-connected limit, but they can be used also as an approximate algorithm for studying stochastic dynamics on generic graphs. Recent efforts have aimed at extending DMFT to properly account for sparsity. In particular, [50] proposed an algorithmic formulation of DMFT for sparse directed networks, employing a population dynamics approach to systematically average over disordered couplings.

The present work generalizes these results to the study of linearly-coupled SDEs on sparse undirected graphs introducing a message passing algorithm for local cavity moments, such as one-time averages, and two-time response functions and correlation functions. This is done first applying the dynamic cavity method to a graphical model representation obtained from the path-integral formulation of coupled SDEs (Martin-Siggia-Rose-Janssen-DeDominicis formalism [51–53]) and then employing a second-order expansion in the interaction strength to the obtained action, in the spirit of [27]. Notably, our expansion aligns with a Gaussian Ansatz for the cavity messages, which is exact for linear systems of SDEs with additive noise and in the presence of global constraints, like in the case of spherical 2-spin models. This formulation also highlights a deep relationship with the use of cavity method in random matrix theory [54,55]. Furthermore, we introduce a perturbative closure scheme to address the presence of non-linear terms, with applications to relaxation in a cubic force and to noise-driven

phase transitions. Gaussian approximations for local cavity actions, parametrized in terms of one-time and two-time cavity averages, have been previously proposed. These approximations were derived from supersymmetric formulations of stochastic dynamics [56] and within the framework of Bosonic–DMFT for quantum many-body systems using an imaginary-time formulation [57–59]. However, these earlier works did not develop the specific algorithmic formulation introduced in this study.

A related work [46] is based on a large-connectivity expansion of the disorder-averaged dynamic, that implies a Gaussian approximation for the corresponding action. In contrast, our method is formulated at the level of single disorder instances and is valid for finite, sparse graphs with arbitrary topologies. As such, it can be employed as a general-purpose algorithm that captures finite-size effects and heterogeneity-induced fluctuations. Another recent work [50] develops a DMFT formulation targeted to sparse directed networks with nonlinear interactions. In this respect, our method applies to both *directed* and *undirected* networks with linear interactions, and leads to a closed set of self-consistent equations for average quantities such as means, response functions, and correlations.

The manuscript is organized as follows: section 2 presents the method and its derivation. Section 2.3 illustrates the development of the small-coupling expansion of the dynamic cavity equations for linearly coupled SDEs with linear drift and additive noise. It also demonstrates the close relationship of the method with the theory of sparse random matrices. In section 2.4, we propose the perturbative closure technique to account for non-linearities. Results are presented in section 3, which include both linear dynamics with additive noise on Random Regular Graphs (RRGs) and heterogeneous structures, as discussed in section 3.1 and section 3.2, and a linear model with additive noise and cubic perturbation (section 3.3), the noise-driven phase transition in the Bouchaud-Mézard model of wealth distribution (section 3.4) and the spherical 2-spin model (section 3.5) on sparse graphs.

## 1.1 Notation

We summarise here the specific notation used throughout the paper. We consider graph instances $G = (V, E)$, where $V = \{1, \ldots, N\}$ is the set of $N$ nodes, while $E \subseteq V \times V$ is the set of edges. The discrete indices corresponding to the nodes of the graph are indicated by letters $i$, $j$, $k$ and $l$. Complex variables are denoted by $z$, and their real and imaginary part as $x = \mathrm{Re}\, z$ and $y = \mathrm{Im}\, z$ respectively, such that $z = x + iy$. The complex conjugate is $z^* = x - iy$, where i is the imaginary unit. We denote the Dirac distribution over the complex plane as $\delta(z) = \delta(x)\delta(y)$ and we use the complex derivatives $\partial_z = (\partial_x - i\partial_y)/2$ and $\partial_{z^*} = (\partial_x + i\partial_y)/2$. The measure over the complex plane is defined as $d^2z = dx\,dy$. We use boldface letters $\boldsymbol{x} = (x^1, \ldots, x^n)^\top$ for temporal column vectors, whose indices are denoted by the letters $n$ and $m$ and correspond to discretized times, and letters with an arrow on top $\vec{x} = (x_1, \ldots, x_N)^\top$ to denote large column vectors of size $O(N)$, whose indices are denoted by letters $i$, $j$, and $k$ and whose elements can be themselves time vectors (in that case we will use $\vec{\boldsymbol{x}}$). We defined the transpose operation as $\ldots^\top$. Boldface capital letters $\boldsymbol{A}$ are used to denote large matrices of size $O(N)$, sans serif bold letters A to denote $2 \times 2$ square matrices whose elements can be themselves square matrices, and capital letters $A$ to denote time matrices in which indices correspond to discretized times.

## 2 Methods

### 2.1 Dynamic cavity approach for linearly-coupled stochastic dynamics

We consider a complex dynamical system modeled by $N$ continuous degrees of freedom $x_i(t)$, $i = 1, \ldots, N$, which undergo a stochastic dynamics described by a system of coupled SDEs

$$\frac{d}{dt} x_i(t) = f_i(x_i(t)) + \alpha \sum_{j \neq i} a_{ij} J_{ij} x_j(t) + \eta_i(t), \tag{1}$$

where $A = \{a_{ij}\}_{i,j=1,\ldots,N}$ denotes the symmetric adjacency matrix of the underlying interaction graph $G$, and $J_{ij}$ is the interaction strength on the directed edge $j \rightarrow i$. The variable $\eta_i(t)$ is the $i$-th component of a Gaussian noise with mean zero and covariance $\langle \eta_i(t) \eta_{i'}(t') \rangle = 2 g_i(x_i(t)) \delta_{ii'} \delta(t - t')$, $i, i' = 1, \ldots, N$, where we have introduced the average over the noise distribution as $\langle \cdots \rangle$ and the Dirac delta function as $\delta(x)$. $f_i(x)$ represents a generic deterministic drift term, while $g_i(x)$ can be any generic, sufficiently smooth, function of $x$. A rescaling factor $\alpha$ is introduced in front of the interaction term to set a scale for the interaction terms. It will be used to perform a formal systematic series expansion around the non-interacting case, in a way reminiscent of Plefka's or other small-coupling expansions [48,60].

It is more convenient to adopt a discrete-time formulation of the stochastic process by first discretizing the SDE according to the Euler-Maruyama scheme, where time is discretized as $t = n\Delta$, with $\Delta$ a small time step approaching zero. The trajectory $x_i(t)$ of node $i$ becomes a time vector $\boldsymbol{x}_i$ whose $n$-th component is defined as $x_i^n = x_i(t = n\Delta)$. By adopting the Ito convention [61] the noise term $\eta_i(t)$ becomes $\Delta \boldsymbol{\eta}_i$ with $n$-th component defined as $\Delta \eta_i^n = \int_{n\Delta}^{(n+1)\Delta} dt\, \eta_i(t)$. We let $T = \mathcal{T}/\Delta$ be the dimension of the time vector trajectories, where $\mathcal{T}$ is the time horizon of the dynamics. The discretized version of the SDEs become

$$x_i^{n+1} = x_i^n + \left( f_i(x_i^n) + \alpha \sum_{j \neq i} a_{ij} J_{ij} x_j^n \right) \Delta + \Delta \eta_i^n, \tag{2}$$

with $\Delta \boldsymbol{\eta}_i$ being a Gaussian random variable with zero mean and covariance matrix elements $g_{ii',nn'} = \langle \Delta \eta_i^n \Delta \eta_{i'}^{n'} \rangle = 2 g_i(x_i^n) \Delta\, \delta_{ii'} \delta_{nn'}$, where we have introduced the Kronecker symbol as $\delta_{ij}$.

Classical path-integral representations of stochastic differential equations (SDEs), such as the Martin-Siggia-Rose-Janssen-De Dominicis (MSRJD) functional integral formalism [51–53], are constructed by interpreting the discretized version of the SDEs as a set of dynamical constraints on the degrees of freedom. The dynamical partition function is then defined as the sum over all possible trajectories, averaged over realizations of the noise. This can be expressed as

$$Z = \left\langle \int D\vec{x} \prod_i p_0(x_i^0) \prod_n \delta \left( x_i^{n+1} - x_i^n - f_i(x_i^n)\Delta - \alpha \Delta \sum_{j \neq i} a_{ij} J_{ij} x_j^n - \Delta \eta_i^n \right) \right\rangle \tag{3a}$$

$$\propto \left\langle \int D\vec{x} D\vec{\hat{x}} \prod_i p_0(x_i^0) \prod_n e^{-i \hat{x}_i^n \left( x_i^{n+1} - x_i^n - f_i(x_i^n)\Delta - \alpha \Delta \sum_{j \neq i} a_{ij} J_{ij} x_j^n - \Delta \eta_i^n \right)} \right\rangle, \tag{3b}$$

where we have defined the path-integral measures as $\int D\vec{x} = \prod_i \int D\boldsymbol{x}_i = \prod_{i,n} \int dx_i^n$ and $\int D\vec{\hat{x}} = \prod_i \int D\hat{\boldsymbol{x}}_i = \prod_{i,n} \int d\hat{x}_i^n$. The degrees of freedom are assumed to be independent and identically distributed at time $t = 0$ according to $p_0(x)$. The second line follows from the integral representation of the Dirac delta function, $\delta(x) \propto \int_{-\infty}^{\infty} d\hat{x}\, e^{-i\hat{x}x}$, which introduces

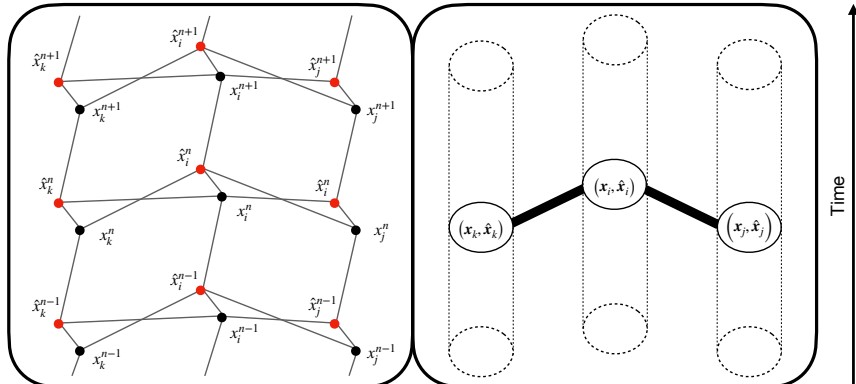

Figure 1: **Graphical model associated with the dynamical partition function**. Left panel: graphical model associated with the dynamical partition function of Eq. (5), where each node corresponds to a variable and each edge corresponds to a factor in the associated factor graph. The time direction is indicated by the arrow on the far right. The factor graph of the model exhibits numerous short loops, rendering a direct cavity approach infeasible. Right panel: graphical model associated with the dynamical partition function when grouping together, for each node $i$, the trajectories $\boldsymbol{x}_i$ and $\hat{\boldsymbol{x}}_i$. The factor graph maintains the same structure as the underlying interaction graph.

auxiliary fields $\hat{\boldsymbol{x}}_i$ to enforce the dynamical constraints at each time step. The average over the Gaussian noise can be performed applying the identity

$$\left\langle e^{i\vec{\hat{x}}^{\top}\vec{\Delta\eta}} \right\rangle_{\vec{\Delta\eta}} = e^{-\frac{1}{2}\vec{\hat{x}}^{\top}g\vec{\hat{x}}} = e^{-\frac{1}{2}\sum_{ii',nn'}\hat{x}_i^n g_{ii',nn'}\hat{x}_{i'}^{n'}}. \tag{4}$$

It is convenient to factorize the dynamical partition function as follows

$$Z \propto \int D\vec{x}\, D\vec{\hat{x}} \prod_i p_0\big(x_i^0\big) \prod_n e^{-i\hat{x}_i^n\big(x_i^{n+1}-x_i^n-f_i(x_i^n)\Delta\big)-\Delta g_i(x_i^n)(\hat{x}_i^n)^2}$$

$$\times \prod_{i<j}\prod_n e^{\alpha\Delta\big(a_{ij}J_{ij}i\hat{x}_i^n x_j^n + a_{ji}J_{ji}i\hat{x}_j^n x_i^n\big)}, \tag{5}$$

and use this form to build a corresponding graphical model. The graphical model associated with the partition function is represented in Fig. 1, left panel. By reordering the variables, it becomes clear that, by grouping together for each node $i$ the trajectories $\boldsymbol{x}_i$ and $\hat{\boldsymbol{x}}_i$ into a single variable node, the factor graph reproduces a tree-like structure when the underlying interaction graph is also a tree (right panel in Fig. 1). This is a consequence of the linear coupling between variables on neighboring nodes, which makes straightforward to disentangle the locally-loopy structure of the factor graph associated with the space-time problem.[1] For

---

[1]In some discrete-state stochastic dynamical processes, it has been shown that a correct cavity approach (Belief Propagation) requires to consider factor graphs with variable nodes containing pairs of trajectories on neighboring sites of the underlying interaction graph [20,23,62]. As an alternative, depending on the structure of the dynamics, one can consider a pair of trajectories formed by the physical variable on a node and an auxiliary local field acting on that node and representing the collective effects of the neighboring variables [27]. In the present case, similar to the application of the cavity approach (Belief Propagation) to the equilibrium Ising model [62], in which the spin variables are linearly coupled with neighboring ones in the energy term, only single-site trajectories are necessary. On the other hand, the conjugate variable $\hat{\boldsymbol{x}}_i$ is directly coupled to the neighboring nodes and plays a role similar to the local auxiliary field in discrete models [27].

nonlinear interactions, such a decoupling generally requires the introduction of auxiliary fields and leads to more complex factor graph representations.

According to this graphical model construction, we propose a *dynamic cavity* Ansatz to approximate marginals distributions of linearly coupled stochastic dynamics on sparse networks. This yields a set of fixed-point equations for the cavity messages

$$c_{i\backslash j}(\boldsymbol{x}_i, \hat{\boldsymbol{x}}_i) = \frac{p_0(x_i^0)}{Z_{i\backslash j}} e^{\sum_n \left\{ -\mathrm{i}\hat{x}_i^n \left[ x_i^{n+1} - x_i^n - f_i(x_i^n)\Delta \right] - g_i(x_i^n)\Delta(\hat{x}_i^n)^2 \right\}}$$
$$\times \prod_{k \in \partial i \backslash j} \int D\boldsymbol{x}_k D\hat{\boldsymbol{x}}_k c_{k\backslash i}(\boldsymbol{x}_k, \hat{\boldsymbol{x}}_k) e^{\alpha\Delta \sum_n \left( J_{ki}\mathrm{i}\hat{x}_k^n x_i^n + J_{ik}\mathrm{i}\hat{x}_i^n x_k^n \right)}, \tag{6}$$

where $\partial i \backslash j$ is the set of neighboring indices of $i$ except for $j$. The cavity equations should be interpreted as a message-passing procedure, where messages are iteratively exchanged between nodes until convergence to a fixed point. Importantly, the normalization of the messages at each step is not critical and can be adjusted after convergence. However, we choose to properly normalize the messages by defining the cavity normalization factor as

$$Z_{i\backslash j} = \int D\boldsymbol{x}_i D\hat{\boldsymbol{x}}_i \, p_0(x_i^0) e^{\sum_n \left\{ -\mathrm{i}\hat{x}_i^n \left[ x_i^{n+1} - x_i^n - f_i(x_i^n)\Delta \right] - g_i(x_i^n)\Delta(\hat{x}_i^n)^2 \right\}}$$
$$\times \prod_{k \in \partial i \backslash j} \int D\boldsymbol{x}_k D\hat{\boldsymbol{x}}_k c_{k\backslash i}(\boldsymbol{x}_k, \hat{\boldsymbol{x}}_k) e^{\alpha\Delta \sum_n \left( J_{ki}\mathrm{i}\hat{x}_k^n x_i^n + J_{ik}\mathrm{i}\hat{x}_i^n x_k^n \right)}. \tag{7}$$

This ensures that the cavity messages are properly normalized quasi-probability distributions and allows for the definition of averages over them.

Finally, completing the cavity and computing the total marginal over $i$ gives

$$c_i(\boldsymbol{x}_i, \hat{\boldsymbol{x}}_i) = \frac{p_0(x_i^0)}{Z_i} e^{\sum_n \left\{ -\mathrm{i}\hat{x}_i^n \left[ x_i^{n+1} - x_i^n - f_i(x_i^n)\Delta \right] - g_i(x_i^n)\Delta(\hat{x}_i^n)^2 \right\}}$$
$$\times \prod_{k \in \partial i} \int D\boldsymbol{x}_k D\hat{\boldsymbol{x}}_k c_{k\backslash i}(\boldsymbol{x}_k, \hat{\boldsymbol{x}}_k) e^{\alpha\Delta \sum_n \left( J_{ki}\mathrm{i}\hat{x}_k^n x_i^n + J_{ik}\mathrm{i}\hat{x}_i^n x_k^n \right)}, \tag{8}$$

where the normalization factor is defined as

$$Z_i = \int D\boldsymbol{x}_i D\hat{\boldsymbol{x}}_i \, p_0(x_i^0) e^{\sum_n \left\{ -\mathrm{i}\hat{x}_i^n \left[ x_i^{n+1} - x_i^n - f_i(x_i^n)\Delta \right] - g_i(x_i^n)\Delta(\hat{x}_i^n)^2 \right\}}$$
$$\times \prod_{k \in \partial i} \int D\boldsymbol{x}_k D\hat{\boldsymbol{x}}_k c_{k\backslash i}(\boldsymbol{x}_k, \hat{\boldsymbol{x}}_k) e^{\alpha\Delta \sum_n \left( J_{ki}\mathrm{i}\hat{x}_k^n x_i^n + J_{ik}\mathrm{i}\hat{x}_i^n x_k^n \right)}. \tag{9}$$

The cavity messages $c_{i\backslash j}(\boldsymbol{x}_i, \hat{\boldsymbol{x}}_i)$ are quasi-distributions defined on pairs of real-valued discrete-time trajectories, which become functionals in the limit of continuous time. We notice that when taking the limit $\Delta \to 0$, the time sums $\Delta \sum_{n=0}^T$ become time integrals $\int_0^{\mathcal{T}} dt$. For example, the cavity marginals become functionals of the trajectory $x_i(t), \hat{x}_i(t)$,

$$c_{i\backslash j}[x_i, \hat{x}_i] = \frac{p_0(x_i(0))}{Z_{i\backslash j}} e^{\int_0^{\mathcal{T}} dt \left\{ -\mathrm{i}\hat{x}_i(t) \left[ \frac{d}{dt} x_i(t) - f_i(x_i(t)) \right] - g_i(x_i(t))\hat{x}_i(t)^2 \right\}}$$
$$\times \prod_{k \in \partial i \backslash j} \int \mathcal{D}x_k \mathcal{D}\hat{x}_k c_{k\backslash i}(x_k, \hat{x}_k) e^{\alpha \int_0^{\mathcal{T}} dt (J_{ik}\mathrm{i}\hat{x}_i(t)x_k(t) + J_{ki}\mathrm{i}\hat{x}_k(t)x_i(t))}, \tag{10}$$

where the integrals have to be understood as path integrals over the trajectories of the neighbors $x_k(t), \hat{x}_k(t)$. For the sake of simplicity, we will only use in what follows the discretized notation unless specified.

Without any specific information on the parametric form of the cavity marginals, the dynamic cavity equations in Eq. (6) are of very limited utility in case of general linearly-coupled stochastic dynamics, because their algorithmic implementation is computationally demanding. In what follows, we discuss a small coupling expansion of the dynamic cavity equations that leads to a more useful set of equations for physically measurable quantities.

## 2.2 Small-coupling expansion

We exploit the presence of the "small" parameter $\alpha$ to perform an expansion of the exponential term containing the coupling between variables on neighboring sites. The expansion, reminiscent of the Plefka approach [60], can thus be interpreted as a small-coupling expansion around the non-interacting case or, equivalently, as a large-connectivity expansion.

By introducing a local statistical average over cavity marginals

$$\langle \mathcal{O}(\boldsymbol{x}_k, \hat{\boldsymbol{x}}_k)\rangle_{k\setminus i} = \int D\boldsymbol{x}_k D\hat{\boldsymbol{x}}_k c_{k\setminus i}(\boldsymbol{x}_k, \hat{\boldsymbol{x}}_k)\, \mathcal{O}(\boldsymbol{x}_k, \hat{\boldsymbol{x}}_k),\tag{11}$$

where $\mathcal{O}(\boldsymbol{x}_k, \hat{\boldsymbol{x}}_k)$ is any generic function of the trajectories $\boldsymbol{x}_k$, $\hat{\boldsymbol{x}}_k$, we can define the *cavity averages*

$$\mu_{i\setminus j}(t) = \langle x_i(t)\rangle_{i\setminus j},\tag{12}$$

$$\hat{\mu}_{i\setminus j}(t) = \langle \hat{x}_i(t)\rangle_{i\setminus j},\tag{13}$$

the *connected (local) two-time cavity correlation function*

$$C_{i\setminus j}(t, t') = \langle x_i(t)x_i(t')\rangle_{i\setminus j} - \langle x_i(t)\rangle_{i\setminus j}\langle x_i(t')\rangle_{i\setminus j}\tag{14a}$$

$$:= C_{i\setminus j}^{dc}(t, t') - \mu_{i\setminus j}(t)\mu_{i\setminus j}(t'),\tag{14b}$$

the *(local) cavity response function*

$$R_{i\setminus j}(t, t') = \langle x_i(t)\mathrm{i}\hat{x}_i(t')\rangle_{i\setminus j},\tag{15}$$

and the two-time correlation of conjugate variables

$$B_{i\setminus j}(t, t') = \langle \mathrm{i}\hat{x}_i(t)\mathrm{i}\hat{x}_i(t')\rangle_{i\setminus j}.\tag{16}$$

In principle, all these terms should be retained. However, under the assumption of causal dynamics, $\hat{\mu}_{i\setminus j}(t)$ and $B_{i\setminus j}(t, t')$ are found to vanish self-consistently within our approximation. This mirrors what happens in the exact generating functional formalism, where the vanishing of conjugate moments follows from normalization [48, 53, 63], though we cannot rigorously prove normalization of the cavity messages due to their approximate nature. In addition, it can be demonstrated that, since $R_{i\setminus j}(t, t')$ is the response function of $x_i(t)$ to a perturbing field,[2] it vanishes for any $t' \geq t$. Defining the discretized averages as $\mu_{i\setminus j}^n = \mu_{i\setminus j}(t = n\Delta)$, $C_{i\setminus j}^{n,n'} = C_{i\setminus j}(t = n\Delta, t' = n'\Delta)$ and $R_{i\setminus j}^{n,n'} = R_{i\setminus j}(t = n\Delta, t' = n'\Delta)$, the cavity marginal can be expanded at second order in $\alpha$, obtaining

$$c_{i\setminus j}(\boldsymbol{x}_i, \hat{\boldsymbol{x}}_i) \propto e^{\sum_n\left(-\mathrm{i}\hat{x}_i^n(x_i^{n+1} - x_i^n - f_i(x_i^n)\Delta) - \Delta g_i(x_i^n)(\hat{x}_i^n)^2\right) + \alpha\Delta\sum_{k\in\partial i\setminus j}\sum_n J_{ik}\mathrm{i}\hat{x}_i^n\mu_{k\setminus i}^n}$$

$$\times e^{\frac{1}{2}\alpha^2\Delta^2\sum_{k\in\partial i\setminus j}\sum_{n,n'}\left(J_{ik}^2\mathrm{i}\hat{x}_i^n C_{k\setminus i}^{n,n'}\mathrm{i}\hat{x}_i^{n'} + 2J_{ik}J_{ki}\mathrm{i}\hat{x}_i^n R_{k\setminus i}^{n,n'}x_i^{n'}\right)}.\tag{17}$$

Having already discussed some possible justifications for performing the expansion, we will henceforth set $\alpha = 1$.

---

[2]Adding a small external field $h_i(t)$ to the SDE Eq. (1) a term $\mathrm{i}\hat{x}_i^n h_i^n$ appears in the exponent of the cavity measure, from which $R_{i\setminus j}(t, t') = \delta\langle x_i(t)\rangle_{i\setminus j}/\delta h_i(t')\big|_{h=0}$.

## 2.3 Gaussian expansion cavity method for linear dynamics with additive noise

We consider the simplest dynamical scenario of a system of linearly interacting Ornstein-Uhlenbeck processes [64, 65], which are described by a system of linearly coupled SDEs with additive thermal noise. Such systems arise in various domains: in physics, they model the velocities of Brownian particles subject to friction and hydrodynamic fluctuations [64]; in finance, they describe correlated interest rates and systemic risk in interbank networks [66, 67]; in evolutionary biology, they model phenotypic evolution under stabilizing selection and genetic drift [68, 69]. Such a system is exactly solvable by direct diagonalization, so that its long-term dynamical properties can be fully determined by examining the spectral properties of the underlying interaction matrix. In this context, the second-order truncation of the small-coupling expansion yields Gaussian cavity messages. These messages are parametrized by one-time and two-time averages of the variables $x_i$ and $\hat{x}_i$, from which a closed set of dynamical equations can be derived. We will demonstrate that solving these equations provides sufficient information to reconstruct (within the cavity approximation) the spectral density of the underlying random interaction matrix. This reveals a close relationship between our method and some well-established applications of the cavity method in random matrix theory.

### 2.3.1 Gaussian cavity Ansatz

The Gaussian structure of the cavity marginals in the space of pairs of trajectories $\boldsymbol{x}_i$ and $\hat{\boldsymbol{x}}_i$ becomes apparent once we substitute $f_i(x_i(t)) = -\lambda_i x_i(t)$ and $g_i(x_i(t)) = D$, $i = 1, \ldots, N$, into the expression in Eq. (17), where we have introduced the relaxation rates $\lambda_i$ and the thermal noise coefficient $D$. For shortness of notation one can define

$$c_{i\backslash j}(\boldsymbol{x}_i, \hat{\boldsymbol{x}}_i) = \frac{1}{Z_{i\backslash j}} e^{-S^0_{i\backslash j}(\boldsymbol{x}_i, \hat{\boldsymbol{x}}_i)}, \tag{18}$$

where $S^0_{i\backslash j}(\boldsymbol{x}_i, \hat{\boldsymbol{x}}_i)$ is the Gaussian cavity action

$$S^0_{i\backslash j}(\boldsymbol{x}_i, \hat{\boldsymbol{x}}_i) = \sum_n \left[ i\hat{x}_i^n \left( x_i^{n+1} - x_i^n + \lambda_i x_i^n \Delta \right) + \Delta D (\hat{x}_i^n)^2 \right] - \Delta \sum_{k \in \partial i \backslash j} \sum_n J_{ik} i\hat{x}_i^n \mu_{k\backslash i}^n$$
$$- \frac{1}{2} \Delta^2 \sum_{k \in \partial i \backslash j} \sum_{n,n'} \left( J_{ik}^2 i\hat{x}_i^n C_{k\backslash i}^{n,n'} i\hat{x}_i^{n'} + 2 J_{ik} J_{ki} i\hat{x}_i^n R_{k\backslash i}^{n,n'} x_i^{n'} \right). \tag{19}$$

Taking the limit $\Delta \to 0$, the action becomes a quadratic functional of the pair of trajectories $\boldsymbol{x}_i$ and $\hat{\boldsymbol{x}}_i$

$$S^0_{i\backslash j}(x_i, \hat{x}_i) = \int dt \left[ i\hat{x}_i(t) \left( \frac{\partial}{\partial t} x_i(t) + \lambda_i x_i(t) \right) + D\hat{x}_i^2(t) \right] - \sum_{k \in \partial i \backslash j} \int dt \, J_{ik} i\hat{x}_i(t) \mu_{k\backslash i}(t) \tag{20}$$
$$- \frac{1}{2} \sum_{k \in \partial i \backslash j} \int dt dt' \left( J_{ik}^2 i\hat{x}_i(t) C_{k\backslash i}(t,t') i\hat{x}_i(t') + 2 J_{ik} J_{ki} i\hat{x}_i(t) R_{k\backslash i}(t,t') x_i(t') \right).$$

Motivated by this result, it is convenient to consider again the discretized Euler-Maruyama version of the stochastic dynamics and put forward a Gaussian parametric Ansatz for the corresponding dynamic cavity marginal,

$$c_{i\backslash j}(\boldsymbol{x}_i, \hat{\boldsymbol{x}}_i) = \frac{1}{Z_{i\backslash j}} e^{-\frac{1}{2}(\mathsf{X} - \mathsf{M}_{i\backslash j})^\top \mathsf{G}_{i\backslash j}^{-1}(\mathsf{X} - \mathsf{M}_{i\backslash j})}, \tag{21}$$

with normalization factor

$$Z_{i\backslash j} = \left( (2\pi)^{2(T+1)} \det \mathsf{G}_{i\backslash j} \right)^{-1/2}, \tag{22}$$

where $X_i^\top = (\boldsymbol{x}_i, i\hat{\boldsymbol{x}}_i)$ is a $2(T+1)$ dimensional vector. According to the previous derivation, $M_{i\setminus j}^\top = (\boldsymbol{\mu}_{i\setminus j}, i\hat{\boldsymbol{\mu}}_{i\setminus j})$ is a $2(T+1)$ dimensional vector and the elements of $G_{i\setminus j}$ are $(T+1)\times(T+1)$ time matrices

$$G_{i\setminus j} = \begin{pmatrix} C_{i\setminus j} & R_{i\setminus j} \\ R_{i\setminus j}^\top & B_{i\setminus j} \end{pmatrix}, \tag{23}$$

where

$$C_{i\setminus j}^{n,n'} = \langle x_i^n x_i^{n'} \rangle_{i\setminus j} - \langle x_i^n \rangle_{i\setminus j} \langle x_i^{n'} \rangle_{i\setminus j},$$
$$R_{i\setminus j}^{n,n'} = \langle x_i^n i\hat{x}_i^{n'} \rangle_{i\setminus j},$$

and

$$B_{i\setminus j}^{n,n'} = \langle i\hat{x}_i^n i\hat{x}_i^{n'} \rangle_{i\setminus j} - \langle i\hat{x}_i^n \rangle_{i\setminus j} \langle i\hat{x}_i^{n'} \rangle_{i\setminus j}.$$

Again, due to the causality of the dynamics, $\hat{\boldsymbol{\mu}}_{i\setminus j}$ and $B_{i\setminus j}$ are assumed to be zero for all times. The validity of this assumption can be verified self-consistently.

In matrix form, the cavity expression in Eq. (6) becomes

$$c_{i\setminus j}(\boldsymbol{x}_i, \hat{\boldsymbol{x}}_i) \propto e^{-\frac{1}{2}X_i^\top G_{0,i}^{-1} X_i} \prod_{k\in\partial i\setminus j} \int DX_k e^{-\frac{1}{2}(X_k - M_{k\setminus i})^\top G_{k\setminus i}^{-1}(X_k - M_{k\setminus i})} e^{-X_i^\top J_{ik} X_k}, \tag{24}$$

in which we introduced a matrix notation for the interaction terms,

$$J_{ik} = \begin{pmatrix} 0 & J_{ki}\Delta\mathbb{I} \\ J_{ik}\Delta\mathbb{I} & 0 \end{pmatrix}, \tag{25}$$

and the matrix operator

$$G_{0,i}^{-1} = \begin{pmatrix} 0 & \mathbb{E}^{-1} - \mathbb{I} + \lambda_i\Delta\mathbb{I} \\ \mathbb{E}^{+1} - \mathbb{I} + \lambda_i\Delta\mathbb{I} & -2\Delta D\mathbb{I} \end{pmatrix}, \tag{26}$$

where $\mathbb{I}$ is the $(T+1)\times(T+1)$ identity matrix and $\mathbb{E}^{\pm 1}$ are shift operators represented by $(T+1)\times(T+1)$ square matrices

$$\mathbb{E}^{-1} = \begin{pmatrix} \ddots & 0 & 0 & 0 \\ 1 & 0 & 0 & 0 \\ 0 & 1 & 0 & 0 \\ 0 & 0 & 1 & \ddots \end{pmatrix}, \qquad \mathbb{E}^{+1} = \begin{pmatrix} \ddots & 1 & 0 & 0 \\ 0 & 0 & 1 & 0 \\ 0 & 0 & 0 & 1 \\ 0 & 0 & 0 & \ddots \end{pmatrix}. \tag{27}$$

The integral in Eq. (24) has to be intended as $\int DX_i = \int D\boldsymbol{x}_i D\hat{\boldsymbol{x}}_i$. Computing the multivariate Gaussian integrals, and using $X_i^\top J_{ik} = \left(J_{ik}^\top X_i\right)^\top$, the cavity marginal can be written as follows

$$c_{i\setminus j}(\boldsymbol{x}_i, \hat{\boldsymbol{x}}_i) \propto e^{-\frac{1}{2}X_i^\top G_{0,i}^{-1} X_i} \prod_{k\in\partial i\setminus j} \int DX_k e^{-\frac{1}{2}(X_k - M_{k\setminus i})^\top G_{k\setminus i}^{-1}(X_k - M_{k\setminus i}) + X_i^\top J_{ik} X_k} \tag{28a}$$

$$\propto e^{-\frac{1}{2}X_i^\top G_{0,i}^{-1} X_i + \frac{1}{2}\sum_{k\in\partial i\setminus j} X_i^\top J_{ik} G_{k\setminus i} J_{ik}^\top X_i + \sum_{k\in\partial i\setminus j} X_i^\top J_{ik} M_{k\setminus i}}. \tag{28b}$$

Comparing it with Eq. (21) we obtain an identity for the exponents. Since the identity must be verified for any configuration $X_i$, we can equate separately the terms appearing with different

powers of $X_i$, that results in two matrix relations

$$
\mathsf{G}_{i\setminus j} = \left( \mathsf{G}_{0,i}^{-1} - \sum_{k\in\partial i\setminus j} \mathsf{J}_{ik}\mathsf{G}_{k\setminus i}\mathsf{J}_{ik}^\top \right)^{-1}, \tag{29}
$$

$$
\mathsf{G}_{i\setminus j}^{-1}\mathsf{M}_{i\setminus j} = \sum_{k\in\partial i\setminus j} \mathsf{J}_{ik}\mathsf{M}_{k\setminus i}. \tag{30}
$$

Substituting Eq. (29) into Eq. (30), after some algebra, we obtain two sets of local equations for the cavity propagator (or cavity Green's function) $\mathsf{G}_{i\setminus j}$ and for the cavity means $\mathsf{M}_{i\setminus j}$, that is

$$
\mathsf{G}_{0,i}^{-1}\mathsf{G}_{i\setminus j} = \mathsf{I} + \sum_{k\in\partial i\setminus j} \mathsf{J}_{ik}\mathsf{G}_{k\setminus i}\mathsf{J}_{ik}^\top\mathsf{G}_{i\setminus j}, \tag{31}
$$

$$
\mathsf{G}_{0,i}^{-1}\mathsf{M}_{i\setminus j} = \sum_{k\in\partial i\setminus j} \mathsf{J}_{ik}\mathsf{M}_{k\setminus i} + \sum_{k\in\partial i\setminus j} \mathsf{J}_{ik}\mathsf{G}_{k\setminus i}\mathsf{J}_{ik}^\top\mathsf{M}_{i\setminus j}, \tag{32}
$$

where $\mathsf{I}$ stays for the identity matrix

$$
\mathsf{I} = \begin{pmatrix} \mathbb{I} & 0 \\ 0 & \mathbb{I} \end{pmatrix}. \tag{33}
$$

Performing the matrix product in Eq. (31) explicitly results in a set of dynamical equations for the cavity response functions and the cavity correlation functions,

$$
R_{0,i}^{-1}R_{i\setminus j} = \mathbb{I} + \Delta^2 \sum_{k\in\partial i\setminus j} J_{ik}J_{ki}R_{k\setminus i}R_{i\setminus j}, \tag{34}
$$

$$
R_{0,i}^{-1}C_{i\setminus j} = 2D\Delta R_{i\setminus j}^\top + \Delta^2 \sum_{k\in\partial i\setminus j} J_{ik}^2 C_{k\setminus i}R_{i\setminus j}^\top + \Delta^2 \sum_{k\in\partial i\setminus j} J_{ik}J_{ki}R_{k\setminus i}C_{i\setminus j}, \tag{35}
$$

where the "non-interacting" response function $R_{0,i}$, satisfying the relation $R_{0,i}^{-1} = \mathbb{E}^{+1} - \mathbb{I} + \lambda_i\Delta\mathbb{I}$, was also introduced. From Eq. (32) we obtain instead a dynamical equation for the cavity mean $\boldsymbol{\mu}_{i\setminus j}$,

$$
R_{0,i}^{-1}\boldsymbol{\mu}_{i\setminus j} = \Delta \sum_{k\in\partial i\setminus j} J_{ik}\boldsymbol{\mu}_{k\setminus i} + \Delta^2 \sum_{k\in\partial i\setminus j} J_{ik}J_{ki}R_{k\setminus i}\boldsymbol{\mu}_{i\setminus j}. \tag{36}
$$

Equations (31) and (32) also demonstrate that the assumption that the quantities $\hat{\boldsymbol{\mu}}_{i\setminus j}$ and $B_{i\setminus j}$ vanish is self-consistently verified. In the supplemental material, a more general version of the equations, in which $\hat{\boldsymbol{\mu}}_{i\setminus j}$ and $B_{i\setminus j}$ are not zero, is reported.

In the continuous-time limit $\Delta \to 0$, the non-interacting response takes the form $R_{0,i}^{-1}/\Delta \to \delta(t-t')(\partial_t + \lambda_i)$, and the previous matrix equations can be replaced by a (closed) set of integro-differential equations

$$
\frac{d}{dt}\mu_{i\setminus j}(t) = -\lambda_i\mu_{i\setminus j}(t) + \sum_{k\in\partial i\setminus j} J_{ik}\mu_{k\setminus i}(t) + \sum_{k\in\partial i\setminus j} \int_0^t dt' J_{ik}R_{k\setminus i}(t,t')J_{ki}\mu_{i\setminus j}(t'), \tag{37}
$$

$$
\frac{\partial}{\partial t}R_{i\setminus j}(t,t') = -\lambda_i R_{i\setminus j}(t,t') + \sum_{k\in\partial i\setminus j} \int_{t'}^t dt'' J_{ik}R_{k\setminus i}(t,t'')J_{ki}R_{i\setminus j}(t'',t') + \delta(t-t'), \tag{38}
$$

$$
\frac{\partial}{\partial t}C_{i\setminus j}(t,t') = -\lambda_i C_{i\setminus j}(t,t') + \sum_{k\in\partial i\setminus j} \int_0^t dt'' J_{ik}R_{k\setminus i}(t,t'')J_{ki}C_{i\setminus j}(t'',t') + 2DR_{i\setminus j}(t',t)
$$
$$
+ \sum_{k\in\partial i\setminus j} \int_0^{t'} dt'' R_{i\setminus j}(t',t'')J_{ik}^2 C_{k\setminus i}(t,t''), \tag{39}
$$

which is the main result of the *Gaussian Expansion Cavity Method (GECaM)* in the case of linear dynamics with additive noise.

A set of self-consistent equations for the full averages (in terms of the cavity ones) can be similarly obtained,

$$\frac{d}{dt}\mu_i(t) = -\lambda_i\mu_i(t) + \sum_{k\in\partial i}J_{ik}\mu_{k\setminus i}(t) + \sum_{k\in\partial i}\int_0^t dt'J_{ik}R_{k\setminus i}(t,t')J_{ki}\mu_i(t'),\qquad(40)$$

$$\frac{\partial}{\partial t}R_i(t,t') = -\lambda_iR_i(t,t') + \sum_{k\in\partial i}\int_{t'}^t dt''J_{ik}R_{k\setminus i}(t,t'')J_{ki}R_i(t'',t') + \delta(t-t'),\qquad(41)$$

$$\frac{\partial}{\partial t}C_i(t,t') = -\lambda_iC_i(t,t') + \sum_{k\in\partial i}\int_0^t dt''J_{ik}R_{k\setminus i}(t,t'')J_{ki}C_i(t'',t') + 2DR_i(t',t)$$
$$+ \sum_{k\in\partial i}\int_0^{t'} dt''R_i(t',t'')J_{ik}^2C_{k\setminus i}(t,t''),\qquad(42)$$

where we have defined the *full (local) mean* $\mu_i(t) = \langle x_i(t)\rangle_i$, the *full (local) response function* $R_i(t,t') = \langle x_i(t)\mathrm{i}\hat{x}_i(t')\rangle_i$ and the *full (local) two-times correlation function* $C_i(t,t') = \langle x_i(t)x_i(t')\rangle_i - \langle x_i(t)\rangle_i\langle x_i(t')\rangle_i$. Here the average has to be intended over the full marginal

$$\langle\mathcal{O}(\boldsymbol{x}_i,\hat{\boldsymbol{x}}_i)\rangle_i = \int D\boldsymbol{x}_i D\hat{\boldsymbol{x}}_i c_i(\boldsymbol{x}_i,\hat{\boldsymbol{x}}_i)\mathcal{O}(\boldsymbol{x}_i,\hat{\boldsymbol{x}}_i).\qquad(43)$$

These GECaM equations are an intuitive cavity generalization of the dynamical TAP equations derived in [48] by means of the Extended Plefka Expansion, and reduce to them for sufficiently dense interaction graphs (in particular for fully-connected ones).

In the long time limit $t, t' \to \infty$, assuming that the system reaches a steady state with no memory of the initial conditions, the response functions and correlation functions become time translational invariant (TTI). In this limit, response/correlation functions depend only on time differences, that is $R(t,t') = R(t-t'=\tau) = R(\tau)$ and $C(t,t') = C(t-t'=\tau) = C(\tau)$. The TTI response functions obey the equations

$$\frac{d}{d\tau}R_{i\setminus j}(\tau) = -\lambda_iR_{i\setminus j}(\tau) + \sum_{k\in\partial i\setminus j}\int_0^\tau ds\,J_{ik}R_{k\setminus i}(\tau-s)J_{ki}R_{i\setminus j}(s) + \delta(\tau).\qquad(44)$$

Introducing the (one-sided) Laplace transform $\tilde{R}(z) = \int_0^{+\infty}\frac{d\tau}{2\pi}R(\tau)e^{-z\tau}$, and moving to the Laplace domain, we get an algebraic equation for the response function, whose solution is

$$\tilde{R}_{i\setminus j}(z) = \frac{1}{z + \lambda_i - \sum_{k\in\partial i\setminus j}J_{ik}J_{ki}\tilde{R}_{k\setminus i}(z)}.\qquad(45)$$

Similarly, the TTI equations for correlation functions read

$$\frac{d}{d\tau}C_{i\setminus j}(\tau) = -\lambda_iC_{i\setminus j}(\tau) + \sum_{k\in\partial i\setminus j}\int_{-\infty}^\tau ds\,J_{ik}R_{k\setminus i}(\tau-s)J_{ki}C_{i\setminus j}(s) + 2DR_{i\setminus j}(-\tau)$$
$$+ \sum_{k\in\partial i\setminus j}\int_\tau^\infty ds\,R_{i\setminus j}(s-\tau)J_{ik}^2C_{k\setminus i}(s).\qquad(46)$$

Introducing the two-sided Laplace transform $\tilde{C}(z) = \int_{-\infty}^{+\infty}\frac{d\tau}{2\pi}C(\tau)e^{-z\tau}$, the algebraic equation for the correlation function is solved by

$$\tilde{C}_{i\setminus j}(z) = \tilde{R}_{i\setminus j}(-z)\tilde{R}_{i\setminus j}(z)\left[2D + \sum_{k\in\partial i\setminus j}J_{ik}^2\tilde{C}_{k\setminus i}(z)\right].\qquad(47)$$

Equivalently, the full response functions and correlation functions obey the following equations in the Laplace space

$$\tilde{R}_i(z) = \frac{1}{z + \lambda_i - \sum_{k \in \partial i} J_{ik} J_{ki} \tilde{R}_{k \setminus i}(z)}, \tag{48}$$

$$\tilde{C}_i(z) = \tilde{R}_i(-z) \tilde{R}_i(z) \left[ 2D + \sum_{k \in \partial i} J_{ik}^2 \tilde{C}_{k \setminus i}(z) \right]. \tag{49}$$

If the interaction matrix $J$ is symmetric, i.e. $J_{ij} = J_{ji}$ for every $i, j = 1, \ldots, N$, the system satisfies detailed balance and it eventually reaches equilibrium after a sufficient long time. Within this regime we can apply the Fluctuation Dissipation Theorem (FDT),

$$D R_{i \setminus j}^{\text{eq}}(\tau) = -C_{i \setminus j}^{\dot{e}q}(\tau) \Theta(\tau), \tag{50}$$

where we have introduced the Heaviside step function $\Theta(\tau)$, which is equal to 1 for $\tau > 0$ and 0 otherwise. The cavity equilibrium correlations are therefore obtained by solving the set of equations

$$\text{sgn}(\tau) \frac{d}{dt} C_{i \setminus j}^{eq}(\tau) = -\lambda_i C_{i \setminus j}^{eq}(\tau) + \sum_{k \in \partial i \setminus j} \frac{J_{ik}^2}{D} C_{k \setminus i}^{eq}(\tau) C_{i \setminus j}^{eq}(0) - \sum_{k \in \partial i \setminus j} \frac{J_{ik}^2}{D} \int_0^\tau ds\, C_{k \setminus i}^{\dot{e}q}(s) C_{i \setminus j}^{eq}(\tau - s), \tag{51}$$

where $\text{sgn}(\tau)$ is the sign function, which is equal to 1 for $\tau \geq 0$, $-1$ for $\tau < 0$. The full equilibrium correlations are obtained from the cavity ones as

$$\text{sgn}(\tau) \frac{d}{dt} C_i^{eq}(\tau) = -\lambda_i C_i^{eq}(\tau) + \sum_{k \in \partial i} \frac{J_{ik}^2}{D} C_{k \setminus i}^{eq}(\tau) C_i^{eq}(0) - \sum_{k \in \partial i} \frac{J_{ik}^2}{D} \int_0^\tau ds\, C_{k \setminus i}^{\dot{e}q}(s) C_i^{eq}(\tau - s). \tag{52}$$

A complete derivation of the equilibrium cavity equations, along with their numerical implementation, is provided in the supplemental material.

### 2.3.2 Relation with random matrix theory

In the previous derivation, we have considered single instances of the sparse coupling matrix $J$, with elements $a_{ij} J_{ij}$, $i, j = 1, \ldots, N$. However, when considering disordered systems [70], at least one between the underlying interaction graph $G$ and the couplings is usually assumed to be drawn from a probability distribution. Within this setting the coupling matrix $J$ becomes a sparse random matrix [71, 72], sampled from an ensemble of random matrices with distribution $p_{\text{dis}}(J)$.

It is convenient to use a vectorial form to write a general linear system of SDEs with additive thermal noise,

$$\frac{d}{dt} \vec{x}(t) = -\lambda \vec{x}(t) + J \vec{x}(t) + \vec{\eta}(t) + \vec{h}(t), \tag{53}$$

where we introduced the diagonal relaxation rates matrix $\lambda$, with elements $\lambda_{ij} \delta_{ij}$, $i, j = 1, \ldots, N$, and a vanishing linear term $\vec{h}$ needed to compute responses. Given an initial condition $\vec{x}(0) = \vec{x}_0$ the solution can be formally written as follows

$$\vec{x}(t) = e^{-\lambda t} \vec{x}_0 + \int_0^t dt'\, e^{(J-\lambda)(t-t')} \left[ \vec{\eta}(t') + \vec{h}(t') \right]. \tag{54}$$

In the long time limit, assuming that all the relaxation rates $\lambda_i$ are positive, the term depending on the initial condition vanishes. The system has a stable solution if all the eigenvalues of the

matrix $\tilde{J} = J - \lambda$ have negative real parts. Within this assumption, the response matrix is obtained as

$$R(t, t') = \frac{\delta}{\delta \vec{h}(t')} \langle \vec{x}(t) \rangle \Big|_{\vec{h}=0} = \Theta(t - t') e^{(J-\lambda)(t-t')}, \tag{55}$$

which has manifestly a TTI form. Its one-sided Laplace transform can be written as

$$\tilde{R}(z) = \int_0^\infty d\tau\, e^{(J-\lambda)\tau} e^{-z\tau} = [z - (J - \lambda)]^{-1}, \tag{56}$$

and can be interpreted as the resolvent of the random matrix $\tilde{J}$. Its diagonal elements, which correspond in our case to the Laplace transformed response functions $\tilde{R}_i(z)$, allow us to directly compute the average spectral distribution of the random matrix $\tilde{J}$. These diagonal elements can be computed from Eqs. (45) and (48), which coincide with the cavity equations for the diagonal resolvent of sparse symmetric random matrices [54,73]. If $\tilde{J}$ is symmetric the spectral distributions will have support on the real axis, which corresponds to the cuts of the resolvent $\tilde{R}(z)$. The spectral distribution of $\tilde{J}$ is then recovered as the inverse Stieltjes transform of the trace of the resolvent [71,74,75]

$$\rho_{\tilde{J}}(x) = \frac{1}{N} \sum_{\alpha=1}^N \delta\left(x - \nu_\alpha(\tilde{J})\right) = \frac{1}{\pi N} \lim_{\varepsilon \to 0^+} \operatorname{Im} \sum_{i=1}^N \tilde{R}_i(x - i\varepsilon), \tag{57}$$

where $\nu_\alpha(\tilde{J})$ is the eigenvalue of $\tilde{J}$ associated to its eigenvector $\nu_\alpha$, i.e. $\tilde{J} \nu_\alpha = \nu_\alpha(\tilde{J}) \nu_\alpha$. The empirical spectral distribution of the random matrix $J$ is easily obtained through a change of variable of the Dirac delta function in Eq. (57)

$$\rho_J(x) = \frac{1}{N} \sum_{\alpha=1}^N \delta\left(x - \lambda_\alpha - \nu_\alpha(\tilde{J})\right) = \frac{1}{\pi N} \lim_{\varepsilon \to 0^+} \operatorname{Im} \sum_{i=1}^N \tilde{R}_i(x - \lambda_i - i\varepsilon). \tag{58}$$

When dealing with disordered systems, one is typically interested in the spectral distribution of the entire ensemble of random matrices. This means that instead of analyzing the spectral properties of a single realization of $J$, we consider the statistical properties of the spectrum across many different samples drawn from the disorder distribution $p_{\text{dis}}(J)$. The spectral distribution of the ensemble can be obtained as

$$\rho(x) = \overline{\rho_J(x)} = \frac{1}{\pi N} \lim_{\varepsilon \to 0^+} \operatorname{Im} \sum_{i=1}^N \overline{\tilde{R}_i(x - \lambda_i - i\varepsilon)}, \tag{59}$$

where we have introduced the average over the random matrices ensemble $\overline{\cdots}$. We reiterate that the system has a stable solution if all the eigenvalues of the matrix $\tilde{J}$ have negative real parts. This condition is equivalent to requiring that the spectral distribution $\rho(x)$ is supported entirely on the left half of the complex plane.

If $J$ is non-hermitian we need to resort to hermitization methods [55,76]. We show in the supplemental material how this can be adapted to our setting.

The relation between the GECaM approach and random matrix theory is double sided. One can compute the response/correlation functions of the system knowing the spectral distribution of the interaction matrix. In the case of a system of linear SDEs with additive thermal noise, the dynamics is diffusive and depends on the spectrum of the matrix $J$. Knowing the spectral density is therefore sufficient to compute the response and correlation functions of the linear system.

Let $\rho(x)$ be the average spectral density of the random matrix ensemble from which $J$ was sampled, towards which the empirical spectral distribution $\rho_J(z)$ converges in the thermodynamic limit. We consider here only symmetric real random matrices. The response matrix of

the system can be formally written as Eq. (55). An analogous equation can be found for the correlation matrix (also known as the correlator of the matrix)

$$\boldsymbol{C}^{\mathrm{dc}}(t,t') = \langle \vec{x}(t)\vec{x}^\top(t') \rangle = \int_0^t dt_1 \int_0^{t'} dt_2 \, e^{(\boldsymbol{J}-\boldsymbol{\lambda})(t-t_1)} \langle \vec{\eta}(t_1)\vec{\eta}(t_2) \rangle e^{(\boldsymbol{J}^\top-\boldsymbol{\lambda})(t'-t_2)} \tag{60a}$$

$$= 2D \int_0^{\min(t,t')} dt_1 \, e^{(\boldsymbol{J}-\boldsymbol{\lambda})(t-t_1)} e^{(\boldsymbol{J}^\top-\boldsymbol{\lambda})(t'-t_1)}, \tag{60b}$$

whose diagonal elements correspond to the local (disconnected) correlation function $C_i^{\mathrm{dc}}(t,t')$. If the coupling matrix $\boldsymbol{J}$ is symmetric, the correlator is symmetric as well. Moreover, in the long time limit, it becomes explicitly TTI, i.e.

$$\boldsymbol{C}^{\mathrm{dc}}(\tau) = D \int_{|\tau|}^\infty dw \, e^{(\boldsymbol{J}-\boldsymbol{\lambda})w}, \tag{61}$$

where we have defined $w = t + t' - 2t_1$. In this case, the fluctuation-dissipation theorem (FDT) is expected to hold, because the system satisfies detailed balance. Hence, we have

$$D\boldsymbol{R}(\tau) = -\Theta(\tau)\frac{\partial}{\partial \tau}\boldsymbol{C}^{\mathrm{dc}}(\tau), \tag{62}$$

which can be easily checked comparing Eq. (55) with Eq. (61).

The disorder averaged response and correlation functions of the system can be computed from the spectral density as

$$\overline{R(\tau)} = \frac{1}{N}\sum_i \overline{R_i(\tau)} = \mathrm{Tr}\,\overline{\boldsymbol{R}(\tau)} = \int dx \, \rho(x)\Theta(\tau)e^{(x-\lambda)\tau}, \tag{63}$$

$$\overline{C^{\mathrm{dc}}(\tau)} = \frac{1}{N}\sum_i \overline{C_i^{\mathrm{dc}}(\tau)} = \mathrm{Tr}\,\overline{\boldsymbol{C}^{\mathrm{dc}}(\tau)} = \int dx \, \rho(x)D \int_{|\tau|}^\infty dw \, e^{(x-\lambda)w}, \tag{64}$$

where we have assumed for simplicity $\lambda_i = \lambda$ for each site $i$.

## 2.4 Perturbative closure technique in the presence of non-Gaussian terms

When terms higher than quadratic (or bilinear) in $x_i$ and $\hat{x}_i$ appear in the local effective action at the exponent of the cavity messages $c_{i\setminus j}(x_i, \hat{x}_i)$, the corresponding cavity probability is not Gaussian. This can be due to the presence of non-linear drift terms or multiplicative noise terms in the original SDEs. In such cases, we introduce a perturbative parameter $\varepsilon$ to systematically control the expansion of the cavity measure around its Gaussian form. The cavity probability can then be written as

$$c_{i\setminus j}(x_i, \hat{x}_i) \propto e^{-S_{i\setminus j}^0(x_i, \hat{x}_i) - \varepsilon S_{i\setminus j}^{NG}(x_i, \hat{x}_i)}, \tag{65}$$

where $S_{i\setminus j}^0(x_i, \hat{x}_i) = \frac{1}{2}\left(\mathsf{X}_i - \mathsf{M}_{i\setminus j}\right)^\top \mathsf{G}_{i\setminus j}^{-1}\left(\mathsf{X}_i - \mathsf{M}_{i\setminus j}\right)$ is the usual Gaussian action in Eq. (20), while $S_{i\setminus j}^{NG}(x_i, \hat{x}_i)$ is the non Gaussian part of the action, coming from non-linear drift terms or non-additive noise terms in the system of SDEs.

The non-quadratic terms can be addressed by employing a perturbative expansion and a self-consistent calculation of the response and correlation functions, similar to the methodology used in the *mode-coupling theory* of disordered systems [34,77]. Without loss of generality, we will assume in what follows that $M_{i\setminus j} = 0$ for every edge $(i, j)$ of the interaction graph $G$, since one can always perform the change of variable $x_i \to x_i - \mu_{i\setminus j}$ to shift the mean to zero. For every edge $(i, j)$ we proceed as follows:

1. The incoming cavity messages are assumed to be Gaussian, parametrized by response functions $R_{k\setminus i}(t, t')$ and correlation functions $C_{k\setminus i}(t, t')$, which are collected in the matrix $\mathsf{G}_{k\setminus i}$, i.e.

$$c_{k\setminus i}(\boldsymbol{x}_k, \hat{\boldsymbol{x}}_k) \propto e^{-\frac{1}{2}\mathsf{X}_k^\top \mathsf{G}_{k\setminus i}^{-1} \mathsf{X}_k}. \tag{66}$$

2. The cavity Gaussian propagator $\mathsf{G}_{i\setminus j}^0$ is computed using the GECaM equations

$$\mathsf{G}_{i\setminus j}^0 = \left( \mathsf{G}_{0,i}^{-1} - \sum_{k \in \partial i \setminus j} \mathsf{J}_{ik} \mathsf{G}_{k\setminus i} \mathsf{J}_{ik}^\top \right)^{-1}. \tag{67}$$

3. At this point, $\mathsf{G}_{i\setminus j}^0$ is known as function of the incoming cavity correlators $\left\{ \mathsf{G}_{k\setminus i} \right\}_{k \in \partial i \setminus j}$, but it is not in a closed form. Performing a perturbative expansion of the cavity measure $c_{i\setminus j}$, which is not Gaussian, and closing it self-consistently, the following Dyson equation is obtained

$$\left( \mathsf{G}_{i\setminus j} \right)^{-1} = \left( \mathsf{G}_{i\setminus j}^0 \right)^{-1} - \Sigma_{i\setminus j}^{MF}, \tag{68}$$

where $\Sigma_{i\setminus j}^{MF}$ is the sum of all the 1PI contributions to the self energy $\Sigma_{i\setminus j}$, giving a so-called *mean-field* or *Hartree-Fock* approximation [34, 78]. The self-energy $\Sigma_{i\setminus j}^{MF}$ is also expressed in terms of $\mathsf{G}_{i\setminus j}^0$, which in turn depends on $\left\{ \mathsf{G}_{k\setminus i} \right\}_{k \in \partial i \setminus j}$ defining a fixed-point message passing set of equations.

The procedure is repeated at every cavity message update, until convergence. Two applications of this self-consistent perturbative algorithm are discussed in sections 3.3-3.4.

## 3 Results

In this Section, we present analytical and numerical results illustrating the versatility of the GECaM approach. We begin by studying the case of linear dynamics with additive noise on random regular graphs (RRGs) [79], comparing the relaxation behavior for different degrees and benchmarking it against the fully connected case. The applicability of our method to graphs with heterogeneous degree distribution is then demonstrated, both in the ferromagnetic and disordered interaction settings.

Next, we explore non-linear stochastic dynamics by considering systems with local cubic forces. This model exhibits a noise-induced phase transition between distinct stationary states, whose critical point can be determined through the perturbative closure scheme introduced in section 2.4. The same technique is employed to study a noise-driven non-equilibrium phase transition in the Bouchaud–Mézard model of wealth condensation on sparse graphs. Finally, we show that GECaM exactly recovers the known relaxation dynamics of the spherical 2-spin model on random regular graphs in the thermodynamic limit, where the tree-like approximation is exact.

### 3.1 Linear dynamics with thermal noise on random regular graphs

Although the cavity equations for response and correlation function can be written on any instance of the underlying graph, to obtain analytical results it is convenient to work in the thermodynamic limit (or at the ensemble level) and assume homogeneity of all other parameters, i.e. $J_{ij} = J_{ji} = J$ for every $i, j = 1, \ldots, N$, $\lambda_i = \lambda$ for every $i = 1, \ldots, N$. Because of the equivalence between different nodes in this limit, we can forget about cavity indices and indicate with $R_c$ and $C_c$ the disorder averaged cavity responses and correlations and with $R$

and $C$ the full ones. Within this simplified setting, it is possible to analytically compute the responses and correlations in Laplace space, at least for a linear dynamics with additive noise. In this case, the cavity functions should satisfy the simplified version of self-consistent equations Eqs. (45) and (47), i.e.

$$\tilde{R}_c(z) = \frac{1}{z + \lambda - (K-1)J^2\tilde{R}_c(z)}\,, \tag{69}$$

$$\tilde{C}_c(z) = \tilde{R}_c(-z)\tilde{R}_c(z)\left[2D + (K-1)J^2\tilde{C}_c(z)\right]\,, \tag{70}$$

where we have defined $R_c := R_{i\setminus j}$ and $C_c := C_{i\setminus j}$ for any edge $(i,j)$ due to the homogeneity of the graph structure and interaction couplings. Solving Eq. (69), we obtain the explicit expression of the cavity correlation in the Laplace space,

$$\tilde{R}_c(z) = \frac{z + \lambda \pm \sqrt{(z+\lambda)^2 - 4(K-1)J^2}}{2(K-1)J^2}\,. \tag{71}$$

The full response function $R := R_i$ for any $i = 1, \ldots, N$ can be straightforwardly computed from a label-free version of Eq. (48), that is

$$\tilde{R}(z) = \frac{1}{z + \lambda - KJ^2\tilde{R}_c(z)} \tag{72a}$$

$$= \frac{1}{2}\frac{(K-2)(z+\lambda) \pm K\sqrt{(z+\lambda)^2 - 4(K-1)J^2}}{K^2J^2 - (z+\lambda)^2}\,. \tag{72b}$$

Depending on the choice of the branch of the square root we obtain two different expressions $\tilde{R}_+(z)$ and $\tilde{R}_-(z)$, respectively. The TTI response function in the long time limit $R(t - t')$ must tend to one as $t - t' \to 0$. Therefore $\tilde{R}(z)$ has to decay as $1/z$ for $z \to \infty$. The correct behavior is exhibited by the $\tilde{R}_-(z)$ branch. Henceforth, we will set $\tilde{R}_c(z) := \tilde{R}_{c-}(z)$ and $\tilde{R}(z) \equiv \tilde{R}_-(z)$ in what follows.

Moreover, solving Eq. (70) for $\tilde{C}_c(z)$, we get

$$\tilde{C}_c(z) = \frac{2D\tilde{R}_c(-z)\tilde{R}_c(z)}{1 - (K-1)J^2\tilde{R}_c(-z)\tilde{R}_c(z)}\,, \tag{73}$$

and using the expression of $\tilde{R}_c(z)$ in Eq. (71), we obtain an explicit expression for the correlation function in the Laplace space,

$$\tilde{C}_c(z) = \frac{D}{(K-1)J^2}\left[\frac{\sqrt{(z+\lambda)^2 - 4(K-1)J^2}}{2z} - \frac{\sqrt{(z-\lambda)^2 - 4(K-1)J^2}}{2z} - 1\right]\,. \tag{74}$$

Finally, the full correlation $\tilde{C}(z) := \tilde{C}_i(z)$ can be computed using Eq. (49), i.e.

$$\tilde{C}(z) = \tilde{R}(z)\tilde{R}(-z)\left[2D + KJ^2\tilde{C}_c(z)\right]\,. \tag{75}$$

It is instructive to show how the corresponding spectral distribution, namely the *Kesten-McKay distribution*, can be recovered from the GECaM response and vice-versa.

The average spectral density $\rho(x)$ of $A$, where $A$ is the adjacency matrix of the RRG, was originally computed by Kesten and McKay starting from general theorems [80, 81]. For an RRG with degree $K$, the spectral density converges in the thermodynamic limit $N \to \infty$ to the Kesten-McKay distribution

$$\rho(x) = \frac{K\sqrt{4(K-1) - x^2}}{2\pi(K^2 - x^2)}\,, \qquad |x| \le 2\sqrt{K-1}\,. \tag{76}$$

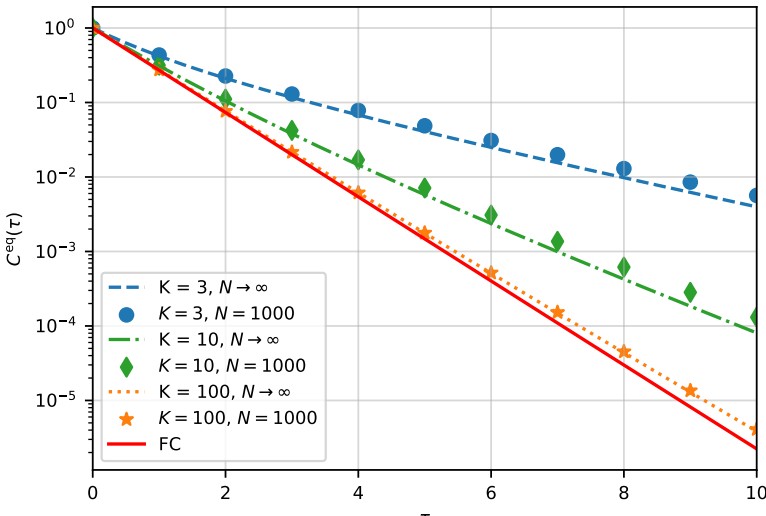

Figure 2: **Average equilibrium correlation on RRGs**. Comparison of the equilibrium correlation function $\langle C(t) \rangle$ in random regular graphs (RRGs) with different connectivities $K$, and in the fully-connected (FC) limit. Results on RRGs are obtained both from the analytic solution in the thermodynamic limit, using the Kesten–McKay spectral density and Eq. (64), and from numerical solution of Eqs. (51) and (52) on graphs with $N = 1000$ nodes. Parameters: $\lambda_i = \lambda = 1.3$, $J = 1/K$, and $D = 1$.

Starting from Eq. (58), and inserting the expression for the GECaM response computed in Eq. (72b), we obtain

$$\rho(x) = \frac{J}{\pi} \lim_{\varepsilon \to 0^+} \text{Im} \tilde{R}(Jx - \lambda - i\varepsilon)$$

$$= \frac{J}{\pi} \frac{\left[ \left( x^2 - 4(K-1) \right)^2 \right]^{1/4}}{2J \left( x^2 - K^2 \right)} \sin \left[ \frac{1}{2} \arg \left( x^2 - 4(K-1) \right) \right], \qquad (77)$$

from which we recover the Kesten-McKay distribution for $|x| \le 2\sqrt{K-1}$.

Proceeding in the opposite direction, the response function can be computed from the knowledge of the Kesten-McKay distribution. The exact expression of the response function in the Laplace space, indeed, can be obtained from Eq. (56),

$$\tilde{R}(z) = \langle \text{Tr} \tilde{\boldsymbol{R}}(z) \rangle = \int dx \, \rho(x) [z - (Jx - \lambda)]^{-1} . \qquad (78)$$

Inserting the Kesten-McKay distribution yields (see supplemental material for details),

$$\tilde{R}(z) = \int_{-2\sqrt{K-1}}^{2\sqrt{K-1}} dx \, \frac{K \sqrt{4(K-1) - x^2}}{2\pi (K^2 - x^2)} [z - (Jx - \lambda)]^{-1} \qquad (79a)$$

$$= \frac{1}{2} \frac{(K-2)(z + \lambda) - K \sqrt{(z + \lambda)^2 - 4(K-1)J^2}}{K^2 J^2 - (z + \lambda)^2} , \qquad (79b)$$

which is identical to expression obtained with the GECaM approach.

Figure 2 compares the behaviour of equilibrium correlation function $C^{\text{eq}}(\tau)$ on RRGs with varying connectivity to the fully-connected case. While the FC case displays purely exponential relaxation, finite connectivity leads to a slight deviation from this behavior. In particular,

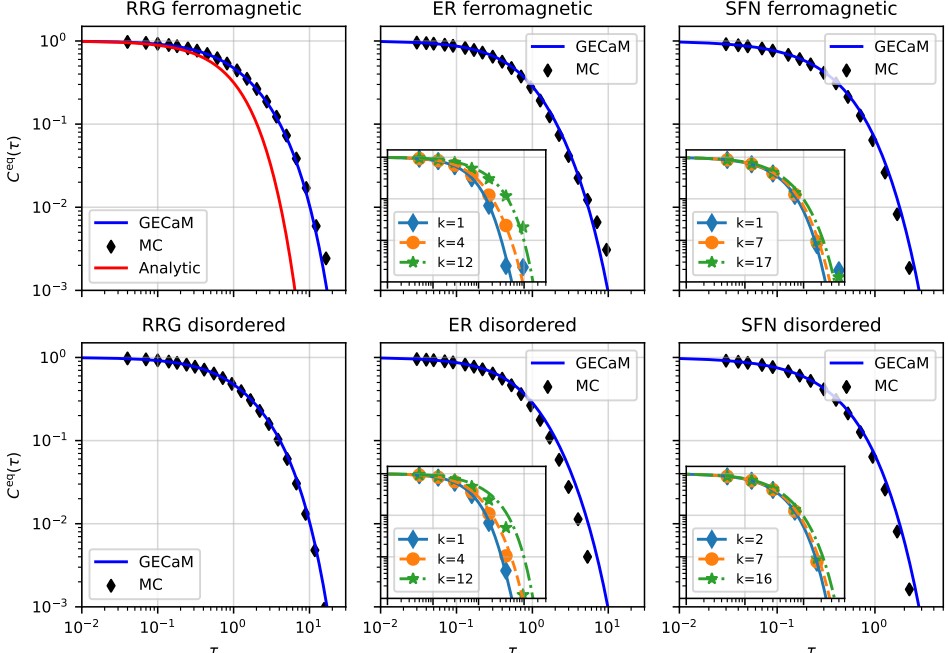

Figure 3: **Average equilibrium correlation on heterogeneous graphs**. Equilibrium correlation functions computed with GECaM (solid lines) and Monte Carlo simulations (diamonds), for different network topologies and interaction types. We consider random regular graphs (RRGs, $N = 200$, $K = 3$, $\lambda = 1.2$), Erdős–Rényi graphs (ER, $N = 100$, $K = 4$, $\lambda = 1.7$), and scale-free networks (SFNs, $N = 200$, $K = 7$, $\alpha = 2.5$, $\lambda = 3.0$), with interaction strength $J = 1/K$ and noise intensity $D = 1$. For the ferromagnetic RRG, the analytical result in the thermodynamic limit is also shown (solid red line). Inset plots show the correlation function for representative nodes with minimum, average, and maximum degrees. Tick labels are omitted in the insets, as the axes are on the same scale as the main plot.

for small values of $K$, the decay becomes slower at longer times, suggesting that sparseness induces a mild non-exponential behaviour in the correlation function. This effect is captured both by the analytic solution in the thermodynamic limit and by numerical solutions of the cavity equations on finite-size graphs, highlighting the influence of network topology on the dynamical relaxation.

## 3.2 Linear dynamics with thermal noise on heterogeneous graphs

To demonstrate the applicability of our method to more realistic network structures, we applied it to heterogeneous sparse graphs with both ferromagnetic and disordered interactions. In particular, we studied systems with bimodal interactions drawn from the symmetric distribution $p(J) = [\delta(J - 1/K) + \delta(J + 1/K)]/2$ as well as purely ferromagnetic couplings $J = 1/K$. We considered three types of network topologies: random regular graphs (RRGs), Erdős–Rényi (ER) graphs [79,82], and scale-free networks (SFNs) with power-law degree distribution [83]. In the ER model, the degree distribution is approximately Poissonian with average degree $K$. For the SFN, the degree distribution follows a power law with exponent $\alpha$ and fixed total number of edges $KN/2$, where $K$ is the average degree.

We computed the average equilibrium correlation function using both the Gaussian Expansion Cavity Method (GECaM), obtained by numerically solving Eqs. (51) and (52), and

Monte Carlo (MC) simulations of the corresponding Langevin dynamics. In the ferromagnetic RRG case, we also show the analytical result in the thermodynamic limit obtained using the Kesten–McKay spectral density and Eq. (64). GECaM accurately captures finite-size corrections absent in the thermodynamic limit, and matches the MC results.

Results for heterogeneous networks (ER and SFN) show that the agreement between GECaM and MC is not always exact. In particular, noticeable deviations can occur in the presence of disorder, as seen in the ER case. To identify the source of the discrepancy, we examined the equilibrium correlation function for representative nodes with minimum, average, and maximum degrees. These results, shown in inset plots, reveal that most of the disagreement arises from nodes with large degrees—those in the tail of the degree distribution. A possible explanation for this behavior is that the GECaM approach assumes a tree-like structure, which is valid in the thermodynamic limit. However, in finite-size graphs, nodes with large degrees can introduce significant correlations that are not captured by the cavity approximation.

### 3.3 Gaussian perturbative closure technique for a cubic perturbation

We consider a system of linearly coupled Ornstein-Uhlenbeck processes with the addition of local cubic forces, described by the system of SDEs

$$\frac{d}{dt}x_i(t) = -\lambda_i x_i(t) - u x_i^3(t) + \sum_{j \neq i} J_{ij} a_{ij} x_j(t) + \eta_i(t), \tag{80}$$

which is equivalent to Eq. (1) with a non-linear drift term $f_i(x_i(t)) = -\lambda_i x_i(t) - u x_i^3(t)$ and additive noise term $g_i(x_i(t)) = D$. This type of system can model thermally diffusing particles constrained in a symmetric double well potential. For clarity, we will consider the underlying interaction graph to be a RRG with homogeneous parameters, i.e. $J_{ij} = J_{ji} = J$ for every edge $(i,j)$, $\lambda_i = \lambda$ for every node $i$. While this simplifies our discussion, extending the results to non-homogeneous cases is straightforward. The cavity probability can be written as in Eq. (65) with non Gaussian action

$$S_{i\backslash j}^{NG}(\boldsymbol{x}_i, \hat{\boldsymbol{x}}_i) = \int_0^{\mathcal{T}} ds\, i\hat{x}_i(s) x_i^3(s). \tag{81}$$

The perturbative parameter $\varepsilon$ is substituted by $u$. The unperturbed system has a steady state fluctuating around zero when stable, i.e. when $\lambda \geq KJ$. We can therefore assume $\mathsf{M}_{i\backslash j} = 0$ for any edge $(i,j)$ without any loss of generality. When the perturbation is switched on, the system exhibits a phase transition between two steady states. For $\lambda \geq \lambda_c(u)$, the stationary mean values $\langle x_i \rangle_i$ vanish for all $i = 1, \dots, N$, with stable noise-driven fluctuations around zero, whereas these values become finite for $\lambda < \lambda_c(u)$.

Using a perturbative expansion, valid for sufficiently small $u$, we obtain a Dyson equation Eq. (68) with a mean-field self energy

$$\Sigma_{i\backslash j}^{MF} = -3u C_{i\backslash j}^{0,0} \sigma_1, \tag{82}$$

where $C_{i\backslash j}^{0,0}$ is the perturbed cavity correlation function at times 0 and $\sigma_1$ is the generalized Pauli matrix

$$\sigma_1 = \begin{pmatrix} 0 & \mathbb{I} \\ \mathbb{I} & 0 \end{pmatrix}. \tag{83}$$

The Dyson equation reads

$$\left(R_{i\backslash j}\right)^{-1} = \left(R_{i\backslash j}^0\right)^{-1} + 3u C_{i\backslash j}^{0,0}, \tag{84}$$

$$\left(R_{i\backslash j}\right)^{-1} C_{i\backslash j} \left(\left(R_{i\backslash j}\right)^{-1}\right)^\top = \left(R_{i\backslash j}^0\right)^{-1} C_{i\backslash j}^0 \left(\left(R_{i\backslash j}^0\right)^{-1}\right)^\top. \tag{85}$$

In the long time limit, the system becomes TTI and Laplace transform can be conveniently applied. Because of the homogeneity assumption, we can neglect node and edge labels and, after inserting the expressions for the cavity response and correlation functions from Eqs. (69) and (70), we obtain

$$\tilde{R}_c^{-1}(z) = z + \lambda + 3uC_c(\tau = 0) - (K-1)J^2\tilde{R}_c^{-1}(z), \tag{86}$$

$$\tilde{C}_c(z) = \tilde{R}_c(z)\left[2D + (K-1)J^2\tilde{C}_c(z)\right]\check{R}_c(-z), \tag{87}$$

where $C_c(\tau = 0)$ should be computed solving self-consistently the equation for $C_c(\tau)$ in the time domain. The perturbation then acts as a renormalization of the parameter $\lambda$ and the problem can be reduced to solving a self-consistent equation for the renormalised parameter $\lambda_R$ such that

$$\lambda_R = \lambda + 3uC_{c,\lambda_R}(\tau = 0), \tag{88}$$

and

$$\tilde{R}_{c,\lambda_R}(z) = \frac{z + \lambda_R - \sqrt{(z + \lambda_R)^2 - 4(K-1)J^2}}{2(K-1)J^2}. \tag{89}$$

Inverting the Laplace transform of the correlation to find $C_{c,\lambda_R}(\tau = 0)$ is challenging. However, one can alternatively compute the spectral density of the cavity graph and use this information to determine the equal-time correlation. Inserting Eq. (89) into Eq. (58) we obtain the "cavity" spectral density

$$\hat{\rho}(x) = \frac{J}{\pi} \lim_{\varepsilon \to 0^+} \mathrm{Im}\,\tilde{R}_{c,\lambda_R}(Jx - \lambda_R - i\varepsilon) \tag{90a}$$

$$= \frac{\sqrt{4(K-1) - x^2}}{2\pi(K-1)}, \tag{90b}$$

where $x \in \left[-2\sqrt{K-1}, 2\sqrt{K-1}\right]$. This quantity is similar to the Wigner semicircle law [71,74] with a dependence on $K$. Inserting the result into Eq. (64) we obtain

$$C_{c,\lambda_R}(\tau = 0) = D\int_{-2\sqrt{K-1}}^{2\sqrt{K-1}} dx\,\frac{\sqrt{4(K-1) - x^2}}{2\pi(K-1)}\int_0^\infty dw\,e^{(Jx - \lambda_R)w} \tag{91a}$$

$$= \frac{D\lambda_R}{2(K-1)J^2}\left[1 - \sqrt{1 - 4\frac{J^2(K-1)}{\lambda_R^2}}\right]. \tag{91b}$$

Plugging this expression into Eq. (88), an equation for $\lambda_R$ is obtained, which is a function of the parameters of the unperturbed problem

$$\lambda_R - 3u\frac{D\lambda_R}{2(K-1)J^2}\left[1 - \sqrt{1 - 4\frac{J^2(K-1)}{\lambda_R^2}}\right] = \lambda, \tag{92}$$

with solution

$$\lambda_R = \lambda\left[1 + \frac{3}{2}\frac{Du}{(K-1)J^2 - 3Du}\left(1 - \sqrt{1 - 4\frac{(K-1)J^2 - 3Du}{\lambda^2}}\right)\right]. \tag{93}$$

For $\lambda_R \geq KJ$, the system has a stable stationary phase, where the order parameter

$$\langle x \rangle = \lim_{t \to \infty}\lim_{N \to \infty}\frac{1}{N}\sum_i \langle x_i(t) \rangle \tag{94}$$

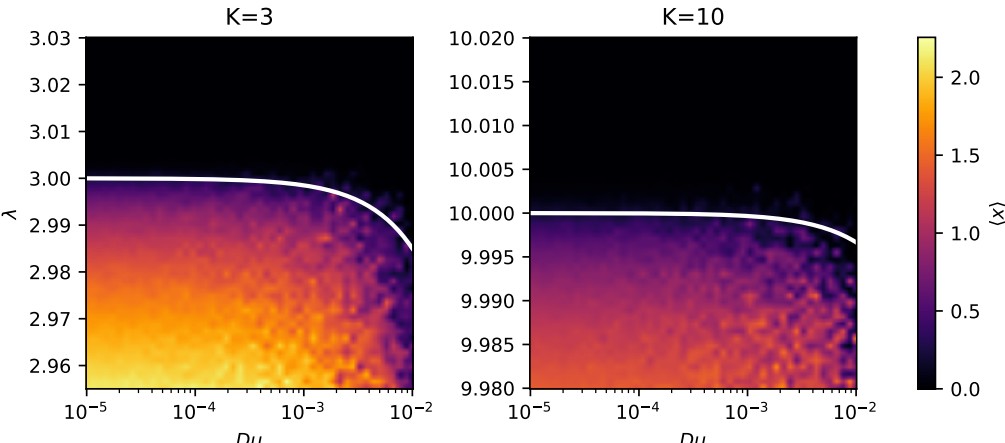

Figure 4: **Phase transition for a cubic perturbation on a RRG**. Estimates of the stationary average value $\langle x \rangle$ obtained from numerical simulations of the dynamical model Eq. (80) on a RRG with $N = 200$ nodes, $K = 3$ (left panel) and $K = 10$ (right panel) neighbors, $J = 1$ and $u = 0.01$ at different values of the product $Du$ and $\lambda$. The stationary value was taken at $t = 1000$, setting the same initial condition $x_i(0) = 1$ for all the nodes. The numerical estimates are compared with the theoretical transition line Eq. (95), represented by the solid white line, demonstrating that our method gives a good prediction of the position of the phase transition.

vanishes. The order parameter is non-zero for $\lambda_R < KJ$. From the critical condition for $\lambda_R$, we can therefore compute the perturbative expression of the critical value

$$\lambda_c(u) = KJ - 3 \frac{Du}{(K-1)J} \, . \tag{95}$$

The analytical transition line in Eq. (95) was compared with numerical simulations of the model on a Random Regular Graph (RRG) with $N = 200$ nodes, considering both $K = 3$ and $K = 10$ neighbors. Numerical simulations were performed using the Euler–Maruyama algorithm, starting with an initial condition of $x_i(0) = 1$ for each degree of freedom, and waiting for the system to reach the stationary state (in practice at time $t = 1000$). Some parameters were kept fixed ($J = 1$ and $u = 0.01$), while the stationary average value $\langle x \rangle$ was computed varying $D$ and $\lambda$. Figure 4 reports a density plot of the stationary average values obtained from simulations alongside the theoretical transition line, demonstrating that the theoretical predictions are consistent with the numerical results.

### 3.4 Noise-driven phase transition with multiplicative noise

Stochastic differential equations with state-dependent ("multiplicative") noise terms are very common in the description of physical, biological, and social systems as they naturally emerge when dealing with intrinsic, as well as environmental, fluctuations [84, 85]. Multiplicative noise interacts with the state variables, leading to more complex and often counterintuitive effects, such as non-equilibrium phase transitions driven by the presence of the noise that alters the stability properties of the dynamics. As a prototype of this behavior, we consider the Bouchaud-Mézard (BM) model [86], although similar stochastic dynamics can be found in various fields of application, from surface growth to ecology. It is known that the wealth distribution in a real economy exhibits a power-law behavior called Pareto's law [87]. The BM model describes the evolution of the wealth in a population of agents, which was originally

assumed to form a fully-connected graph, by the following system of SDEs

$$\frac{d}{dt}x_i(t) = J\sum_{j\neq i} a_{ij}\left(x_j(t) - x_i(t)\right) + \sigma x_i(t)\eta_i(t), \tag{96}$$

where $x_i(t)$ is the wealth of agent $i$ at time $t$, $J$ is the coupling between agents, $\sigma^2$ is the variance of noise and $\eta_i(t)$ is the $i$-th component of a Gaussian white noise with zero average and correlations $\langle\eta_i(t)\eta_{i'}(t')\rangle = \delta_{ii'}\delta(t-t')$, $i, i' = 1,\ldots,N$. Defining $\lambda_i := J\sum_j a_{ij}$ the model corresponds to Eq. (1) with a linear drift term $f_i(x_i(t)) = -\lambda_i x_i(t)$ and a multiplicative noise with $g_i(x_i(t)) = \sigma^2 x_i^2(t)/2$. The SDEs defining the model are invariant under the scaling transformation $x_i \to \alpha x_i$ for any positive constant $\alpha$. Therefore, without loss of generality, we can set $\langle x_i(t)\rangle_i = 1$ for every node $i$ in the theoretical analysis. In numerical studies, we will focus on the statistical properties of the normalized quantity $x_i/\bar{x}$, where $\bar{x}$ is the average over all agents.

In their original paper, Bouchaud and Mézard showed using the mean-field theory that the stationary distribution of normalized wealth $x_i/\bar{x}$ exhibits the expected power-law behavior in a fully connected model. They also found out that the model exhibits a phase transition at $J \leq J_c = \sigma^2$, where the variance of $x/\bar{x}$ diverges. The phenomenon is called *wealth condensation*, since the divergence of the variance implies that wealth condenses to a few rich agents. If instead $J > J_c$ the variance remains finite and most of the agents have a wealth close to the average. Alternatively, by keeping fixed the coupling constant $J$ and varying the amplitude $\sigma^2$ of the noise, the wealth condensation phenomenon can be also interpreted as a realization of a transition from weak-noise to strong-noise regimes [85]. In [88], the wealth condensation phenomenon is studied on sparse random graphs using a combination of adiabatic and independence assumptions. The first one allows the use of self-consistent conditional mean-field closure methods [89, 90], while the independence assumption enables treating neighbors as independent and applying the central limit theorem. This approximation method provides an improved prediction for the location of the phase transition on sparse graphs, such as Random Regular Graphs.

Here we propose an alternative solution employing the perturbative closure technique described in Sect.2.4. Equation (96) can be seen as a linear model with a vanishing additive noise ($D \to 0$ limit) and a multiplicative noise that will be treated using perturbation theory. We can therefore apply our perturbative closure technique expanding in powers of the perturbative parameter $\sigma^2$. In this case, the non Gaussian term of the cavity action is

$$S_{i\backslash j}^{NG}(\boldsymbol{x}_i, \hat{\boldsymbol{x}}_i) = -\frac{1}{2}\int_0^{\mathcal{T}} ds\, x_i^2(s)(\mathrm{i}\hat{x}_i(s))^2. \tag{97}$$

Before proceeding to apply perturbation theory, it is necessary to perform a change of variable in order to eliminate the constant part from $x_i(t)$ [91]. Since $\langle x_i(t)\rangle_i = 1$, we introduce the variable $\phi_i(t)$ defined by $x_i(t) = 1 + \phi_i(t)$ and its conjugate $\hat{\phi}_i = \mathrm{i}\hat{x}_i$. The Gaussian part of the cavity action is unchanged, and since $\langle\phi_i\rangle = 0$, we can safely assume the average vector to be zero. The interaction term becomes

$$S_{i\backslash j}^{NG}(\boldsymbol{\phi}_i, \hat{\boldsymbol{\phi}}_i) = -\frac{1}{2}\int_0^{\mathcal{T}} ds\,(1+\phi_i(s))^2\,\hat{\phi}_i^2(s) \tag{98a}$$

$$= -\frac{1}{2}\int_0^{\mathcal{T}} ds\,\hat{\phi}_i^2(s) - \int_0^{\mathcal{T}} ds\,\phi_i(s)\hat{\phi}_i^2(s) - \frac{1}{2}\int_0^{\mathcal{T}} ds\,\phi_i^2(s)\hat{\phi}_i^2(s). \tag{98b}$$

In the limit $D \to 0$, i.e., in the absence of additive noise, the Gaussian correlation function vanishes identically, $C_{i\backslash j}^0(t, t') = \langle\phi_i(t)\phi_i(t')\rangle_{i\backslash j}^0 = 0$, while the Gaussian response function

$R^0_{i\backslash j}(t,t') = \langle \phi_i(t)\hat{\phi}_i(t')\rangle^0_{i\backslash j}$ satisfies the Laplace space equation

$$\tilde{R}^0_{i\backslash j}(z) = \frac{1}{z + \lambda_i - \sum_{k \in \partial i \backslash j} J^2 \tilde{R}_{k\backslash i}(z)}, \tag{99}$$

which has been already solved for RRGs.

The response function remains unperturbed due to causality restrictions, while the correlation function renormalizes due to the loop corrections coming from the first term in the interaction part of the action. At first order, the contribution is

$$C_{i\backslash j}(t,t') = \sigma^2 \int ds R^0_{i\backslash j}(t,s) R^0_{i\backslash j}(t',s). \tag{100}$$

All the remaining non-vanishing terms can be absorbed in the renormalization of the coupling constant $\sigma^2$ (the others are zero because either $R^0_{i\backslash j}(s,s) = 0$ or $\langle \phi_i(t)\rangle^0_{i\backslash j} = 0$). In matrix form

$$R_{i\backslash j} = R^0_{i\backslash j}, \tag{101}$$

$$C_{i\backslash j} = \sigma^2_{i\backslash j,R} R^0_{i\backslash j}\left(R^0_{i\backslash j}\right)^\top. \tag{102}$$

The renormalization of the coupling constant follows from the perturbative renormalization of the third term in $S^{NG}_{i\backslash j}(\boldsymbol{\phi}_i, \hat{\boldsymbol{\phi}}_i)$, which is easy to express as a geometric series of one-loop corrections (see supplemental material). The re-summation of the series gives

$$\sigma^2_{i\backslash j,R} = \frac{\sigma^2}{1 - \sigma^2 I_{i\backslash j}}, \tag{103}$$

where the term $I_{i\backslash j}$ is given by the integral

$$I_{i\backslash j} = \int_{-\infty}^{\infty} \frac{d\omega}{2\pi} \hat{R}^0_{i\backslash j}(\omega)\hat{R}^0_{i\backslash j}(-\omega), \tag{104}$$

with $\hat{R}^0_{i\backslash j}(\omega) = \tilde{R}^0_{i\backslash j}(z = i\omega)$ being the Gaussian response expressed in the Fourier space.

For a RRG with homogeneous parameters, where $\lambda_i = KJ$ for every node, the integral $I := I_{i\backslash j}$ for every edge $(i,j)$ can be reduced to a simpler one in real space. The final form of the integral is

$$I = \int_{-\infty}^{\infty} \frac{d\omega}{2\pi} \left[\frac{i\omega + KJ - \sqrt{(i\omega + KJ)^2 - 4(K-1)J^2}}{2(K-1)J^2}\right]$$
$$\times \left[\frac{-i\omega + KJ - \sqrt{(-i\omega + KJ)^2 - 4(K-1)J^2}}{2(K-1)J^2}\right] \tag{105a}$$

$$= \frac{2}{\pi\sqrt{K-1}J} \int_{-1}^{1} du \frac{\sqrt{1-u^2}}{\frac{K}{\sqrt{K-1}} - u + \sqrt{\left(\frac{K}{\sqrt{K-1}} - u\right)^2 - 1}}. \tag{105b}$$

A detailed derivation of this expression is reported in the supplemental material.

The fully-connected (FC) limit is easily recovered setting $K = N - 1$, rescaling $J$ to $J/K$ and taking the limit $K \to \infty$,

$$I_{FC} \approx \frac{2}{\pi J}\frac{K}{\sqrt{K-1}} \int_{-1}^{1} du \frac{\sqrt{1-u^2}}{2\frac{K}{\sqrt{K-1}}} \to \frac{1}{2J}. \tag{106}$$

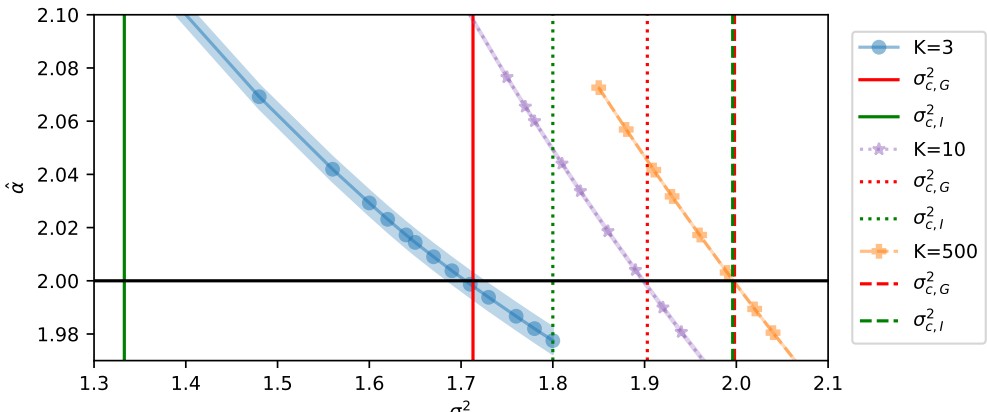

Figure 5: **Tail exponent $\alpha$ of the BM model on a RRG**. Estimates of the tail exponent $\alpha$ obtained by simulation for $K = 3$ (dot, solid lines), $K = 10$ (star, dotted lines) and $K = 500$ (cross, dashed lines) at different values of $\sigma^2$ around the phase transition. Numerical simulations of the BM model on a RRG were carried out for $N = 5000$ nodes and repeated 100 times. Markers correspond to the average value of $\alpha$, while ribbons represent the 95% confidence interval. Vertical lines correspond to the theoretical critical values as predicted by our method $\sigma^2_{c,G}$ (red) and by [88] $\sigma^2_{c,I}$ (green). The horizontal black line corresponds to the critical value $\alpha_c = 2$: a tail exponent below this critical line indicates that the system undergoes wealth condensation.

The renormalized noise parameter diverges at the critical value $\sigma_c = I^{-1}$, indicating a phase transition to the condensed phase. In the FC limit, we recover the value $\sigma^{FC}_c = 2J$, in agreement with the mean-field result [86].

To assess the accuracy of our method, we compared it with numerical simulations. We rescaled $J$ to $J/K$ to facilitate comparison of results across different connectivities $K$. Numerical simulations of the BM model on a RRG were carried out for $N = 5000$ nodes and repeated 100 times using the Milstein algorithm [92] with an initial condition $x_i(0) = 1$ for each parameter value and $J = 1$. The distribution at $t = 1500$ was taken as the stationary distribution. The stationary distribution of $x/\bar{x}$ is expected to have a heavy Pareto tail

$$p\left(\frac{x}{\bar{x}}\right) \overset{x/\bar{x}\to\infty}{\sim} \frac{1}{(x/\bar{x})^{\alpha+1}}, \tag{107}$$

where $\alpha$ is the tail index. The variance of normalized wealth diverges when $\alpha \leq 2$. To estimate the tail exponent we have run a simple Rank-1/2 Ordinary Least Squares (OLS) log-log rank-size regression [93] with a bias reduction term. Figure 5 shows the estimates of $\alpha$ for $K = 3$, $K = 10$ and $K = 500$ at different values of $\sigma^2$ around the phase transition, together with the prediction $\sigma^2_{c,I}$ of [88] and our prediction $\sigma^2_{c,G} = I^{-1}$. Our method provides a better approximation for sparse graphs, while the two methods converge to the same mean-field value when $K$ approaches $N$. Interestingly, the self-consistent method proposed in [88] can be obtained within the GECaM approach when further approximating the cavity quantities by the corresponding full quantities taken at zero time differences (see supplemental material for details).

## 3.5 Relaxation dynamics of the spherical 2-spin model

The spherical $p$-spin model, with $p \geq 3$, has been extensively studied as a prototypical model for the behavior of disordered systems, exhibiting a glassy free-energy landscape as well as

complex dynamical phenomena such as slow relaxation and aging [36]. The $p = 2$ case is structurally much simpler, it does not present a glassy equilibrium free-energy landscape, but a more standard continuous transition into a disguised ferromagnetic phase [94]. The effect of disorder is however enough to ensure non-trivial dynamical properties. In fact, under nearly all initial conditions, the system fails to reach equilibrium, with its relaxation dynamics heavily influenced by the duration of time the system has already spent in the low-temperature phase, indicative of aging behavior. Due to its simplicity, this model can be exactly solved through the diagonalization of the system, provided the spectral properties of the underlying interaction graph are known [7,95,96]. Here, we will show how our method enables the study of this non-trivial relaxation dynamics without the explicit diagonalization of the system or the knowledge of the spectral properties of the underlying interaction graph.

The model is defined by the Hamiltonian

$$H(\vec{x}) = -\frac{1}{2} \sum_{i \neq j} a_{ij} J_{ij} x_i x_j \,, \tag{108}$$

where $x_i(t)$, $i = 1,\dots,N$ are continuous spins with spherical constraint $\sum_{i=1}^{N} x_i^2(t) = N$, $a_{ij}$ are the elements of the adjacency matrix of a RRG with $K$ neighbours and the couplings are independent and identically distributed (iid) symmetric random variables with bimodal distribution

$$P(J_{ij}) = \frac{1}{2} \left( \delta(J_{ij} - J) + \delta(J_{ij} + J) \right) \,. \tag{109}$$

A notable choice for the coupling strength is $J \propto 1/\sqrt{K}$ that ensures to recover known results in the fully-connected limit ($K \to \infty$). The relaxation dynamics satisfies the Langevin equation

$$\frac{d}{dt} x_i(t) = -\frac{\delta H(\vec{x})}{\delta x_i(t)} - \lambda(t) x_i(t) + \eta_i(t) \tag{110a}$$

$$= \sum_{j \neq i} a_{ij} J_{ij} x_j(t) - \lambda(t) x_i(t) + \eta_i(t) \,, \tag{110b}$$

where $\lambda(t)$ is a Lagrange multiplier enforcing the spherical constraint and the thermal noise term $\eta_i(t)$ is Gaussian with zero mean and correlations $\langle \eta_i(t) \eta_{i'}(t') \rangle = 2D\delta_{ii'}\delta(t - t')$ (here $D$ plays the role of temperature $T$ in the usual formulations of the $p$-spin model).

Although the GECaM equations could be derived for any given graph and realization of the coupling constants, we will focus on a simplified setup that enables more straightforward analytic calculations. Due to the spatial homogeneity of the random regular graph, it is reasonable to assume that, in the thermodynamic limit and after averaging over the i.i.d. couplings, both cavity and full quantities become site- and edge-independent. Specifically, we assume $\overline{R_i} := R$ and $\overline{C_i} := C$ for all nodes $i$, and $\overline{R_{i \setminus j}} := R_c$ and $\overline{C_{i \setminus j}} := C_c$ for all edges $(i, j)$. This assumption is justified by the absence of replica symmetry breaking in the 2-spin model—which behaves as a disguised ferromagnet—and by the fact that aging arises solely from dynamical effects. Furthermore, we have verified the validity of this assumption through Monte Carlo simulations of the model; see the supplemental material for details. A crucial consequence of the homogeneity assumption is that the spherical constraint implies a local constraint on the equal time correlation function, i.e. $C(t, t) = 1$. In this homogeneous case, for a sudden quench from the high-temperature phase, the corresponding GECaM equations for the cavity

response functions and cavity correlation functions are

$$\frac{\partial}{\partial t}R_c\left(t,t'\right)=-\lambda\left(t\right)R_c\left(t,t'\right)+\left(K-1\right)J^2\int_{t'}^{t}dt''R_c\left(t,t''\right)R_c\left(t'',t'\right)+\delta\left(t-t'\right), \quad (111)$$

$$\frac{\partial}{\partial t}C_c\left(t,t'\right)=-\lambda\left(t\right)C_c\left(t,t'\right)+\left(K-1\right)J^2\int_{0}^{t}dt''R_c\left(t,t''\right)C_c\left(t'',t'\right)+2DR_c\left(t',t\right)$$

$$+\left(K-1\right)J^2\int_{0}^{t'}dt''R_c\left(t',t''\right)C_c\left(t,t''\right), \tag{112}$$

and for the full response and correlation

$$\frac{\partial}{\partial t}R\left(t,t'\right)=-\lambda\left(t\right)R\left(t,t'\right)+KJ^2\int_{t'}^{t}dt''R_c\left(t,t''\right)R\left(t'',t'\right)+\delta\left(t-t'\right), \tag{113}$$

$$\frac{\partial}{\partial t}C\left(t,t'\right)=-\lambda\left(t\right)C\left(t,t'\right)+KJ^2\int_{0}^{t}dt''R_c\left(t,t''\right)C\left(t'',t'\right)+2DR\left(t',t\right)$$

$$+KJ^2\int_{0}^{t'}dt''R\left(t',t''\right)C_c\left(t,t''\right). \tag{114}$$

Here, for random initial conditions, we neglected the corresponding equations for the cavity means and full means, that would in general explicitly depend on the chosen initial conditions. The spherical constraint $C\left(t,t\right)=1$ implies that $\left(\partial_t C\left(t,t'\right)+\partial_{t'}C\left(t,t'\right)\right)|_{t,t'=s}=0$ from which the expression of the Lagrange multiplier

$$\lambda\left(t\right)=KJ^2\int_{0}^{t}dt''R_c\left(t,t''\right)C\left(t'',t\right)+KJ^2\int_{0}^{t}dt''R\left(t,t''\right)C_c\left(t,t''\right)+D, \tag{115}$$

which gives now coupled equations for (cavity and full) responses and correlations. When setting $J^2\propto 1/K$ and taking the $K\to\infty$ limit, the cavity and full quantities satisfy the same set of equations and the latter represent a $p=2$ version of the Cugliandolo-Kurchan dynamical mean-field equations for the fully-connected $p$-spin model [38].

The GECaM equations can be further simplified by using the following parametric Ansatz,

$$R_c\left(t,t'\right)=\frac{f\left(t'\right)}{f\left(t\right)}R_c^{\text{st}}\left(t-t'\right), \tag{116}$$

$$C_c\left(t,t'\right)=\frac{1}{f\left(t\right)f\left(t'\right)}C_c^1\left(t,t'\right), \tag{117}$$

which are inspired by the form of the exact solution obtained in [95]. The physical meaning of $R_c^{\text{st}}(t-t')$, $C_c^1(t,t')$ and $f(t)$ becomes clear when inserting the Ansatz in Eqs. (111) and (112). In fact, the quantity $R_c^{\text{st}}$ turns out to represent the stationary time-translation invariant cavity response function of a linear system with additive noise and it must satisfy the equation

$$\frac{\partial}{\partial t}R_c^{\text{st}}\left(t-t'\right)=\left(K-1\right)J^2\int_{t'}^{t}dt''R_c^{\text{st}}\left(t-t''\right)R_c^{\text{st}}\left(t''-t'\right)+\delta\left(t-t'\right). \tag{118}$$

On the other hand, $C_c^1$ does not represent the TTI correlation; rather, it must satisfy the following equation

$$\frac{\partial}{\partial t}C_c^1\left(t,t'\right)=2Df^2\left(t\right)R_c^{\text{st}}\left(t'-t\right)+\left(K-1\right)J^2\int_{0}^{t}dt''R_c^{\text{st}}\left(t-t''\right)C_c^1\left(t'',t'\right)$$

$$+\left(K-1\right)J^2\int_{0}^{t'}dt''R_c^{\text{st}}\left(t'-t''\right)C_c^1\left(t,t''\right). \tag{119}$$

The scaling function $f(t)$ is given by

$$f(t) = f(t_0) e^{\int_{t_0}^t \lambda(t')dt'},\tag{120}$$

where $f(t_0) = 1$ for the present choice of initial conditions.

An equivalent parametric Ansatz can be also assumed for the full response and correlation functions

$$R(t,t') = \frac{f(t')}{f(t)}R^{\text{st}}(t-t'),\tag{121}$$

$$C(t,t') = \frac{1}{f(t)f(t')}C^1(t,t'),\tag{122}$$

where $R^{\text{st}}$ is the usual stationary TTI response function

$$\frac{\partial}{\partial t}R^{\text{st}}(t-t') = KJ^2 \int_{t'}^t dt'' R_c^{\text{st}}(t-t'')R^{\text{st}}(t''-t') + \delta(t-t'),\tag{123}$$

while $C^1$ must satisfy the equation

$$\frac{\partial}{\partial t}C^1(t,t') = 2Df^2(t)R^{\text{st}}(t'-t) + KJ^2 \int_0^t dt'' R_c^{\text{st}}(t-t'')C^1(t'',t')$$

$$+ KJ^2 \int_0^{t'} dt'' R^{\text{st}}(t'-t'')C_c^1(t,t'').\tag{124}$$

With some algebraic manipulation, it can be shown that the system of coupled integro-differential equations for $C_c^1(t,t')$ and $C_1(t,t')$ admits the following expressions as a solution

$$C_c^1(t,t') = R_c^{\text{st}}(t+t') + 2D \int_0^{\min(t,t')} dt_1 f^2(t_1)R_c^{\text{st}}(t+t'-2t_1),\tag{125}$$

$$C^1(t,t') = R^{\text{st}}(t+t') + 2D \int_0^{\min(t,t')} dt_1 f^2(t_1)R^{\text{st}}(t+t'-2t_1),\tag{126}$$

where the scaling function $f(t)$ solves the (linear) Volterra integral equation

$$f^2(t) = R^{\text{st}}(2t) + 2D \int_0^t dt_1 f^2(t_1)R^{\text{st}}(2t-2t_1).\tag{127}$$

This equation is formally the same of the exact solution first obtained in [95]. In principle, an explicit solution can be constructed as follows. The TTI equations for $R_c^{\text{st}}$ and $R_{\text{st}}$ can be solved in the Laplace space, and used to solve the equation for $f(t)$ in the Laplace space. If an explicit time-domain expression of $f(t)$ can be computed, then the time behavior of the cavity and full correlation functions can be obtained. Unfortunately, analytically computing the inverse Laplace transform of $f$ is cumbersome. A numerical solution of Eqs. (111) to (114) with the spherical constraint Eq. (115) can be obtained by forward-in-time integration. Figure 6 shows the decay of the correlation function $C(t_w + \tau, t_w)$ for fixed temperature $D = 0.3$ and coupling constant $J = 1.0$, where $\tau$ is the time elapsed after a "waiting time" $t_w$. The correlation function shows the typical "aging" behaviour, i.e. it has an initial fast decay for $\tau \ll t_w$, after which it decays slowly to zero for $\tau > t_w$. The sparseness however decreases the aging effect at fixed waiting time as can be seen from the top-right right and bottom panels. We compared these numerical solutions of the GECaM equations with Monte Carlo simulations of the

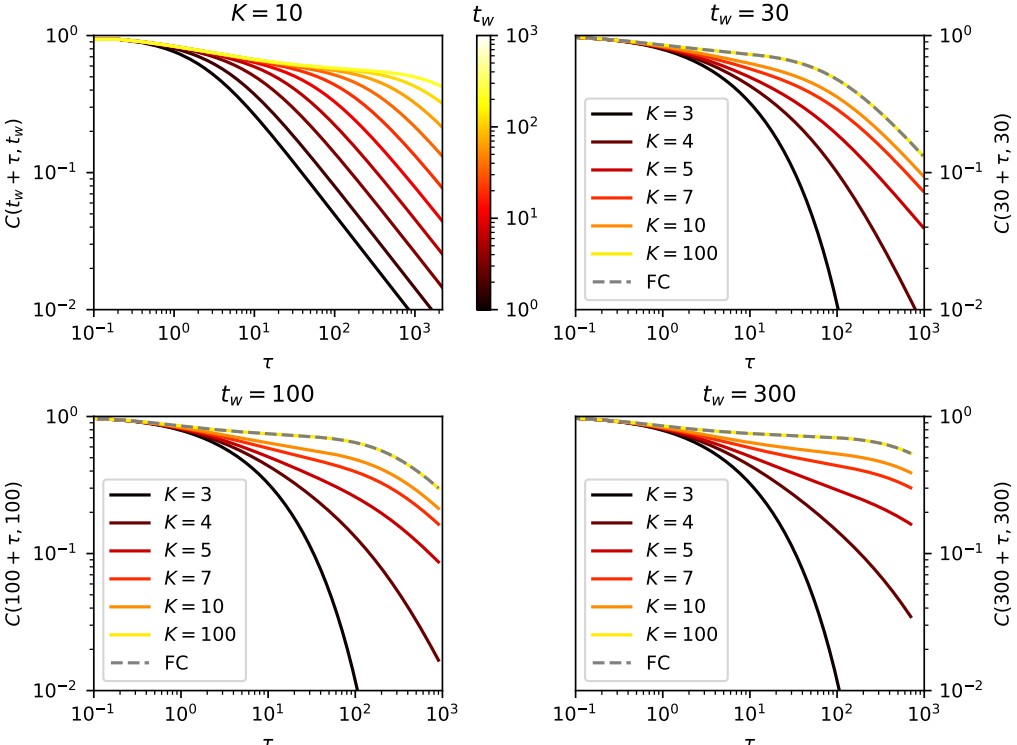

Figure 6: **Correlation function of the 2-spin model on RRGs**. Decay of $C(t_w+\tau, \tau)$ against $\tau$ for the spherical 2-spin model on a RGG with fixed coupling constant $J = 1.0$ and at temperature $D = 0.3$. (Top-left panel) Correlation function decay at fixed $K = 10$ for increasing values of the waiting time $t_w$ (from bottom to top). (Top-right and bottom panels) Correlation function decay at fixed waiting time $t_w = 30, 100, 300$ for $K = 3, 4, 5, 7, 10, 100$ (from bottom to top) together with the fully-connected (FC) case (dashed gray line).

model, finding good agreement despite the presence of strong finite-size effects, as discussed in section 3.2. Further details are provided in the supplemental material. Interestingly, the GECaM equations Eqs. (111) to (115) remain unchanged for a spherical ferromagnet, suggesting that aging behavior should also emerge in this case. The key difference lies in the required scaling of the coupling constant: while the disordered (bimodal) model requires $J \sim 1/\sqrt{K}$ to ensure extensivity of the hamiltonian, the ferromagnetic model requires $J \sim 1/K$. As a consequence, the ferromagnetic model also displays aging in the sparse regime—consistent with the interpretation of the disordered case as a disguised ferromagnet. In contrast, in the large-connectivity limit, aging disappears in the ferromagnetic case due to the vanishing of terms of order $J^2$, while it persists in the disordered case, as established in the literature. We have also verified numerically that this behavior holds for the ferromagnetic model; additional results are presented in the supplemental material.

Further insights on the dynamics of sparse $p$-spin models (also for $p \geq 3$) using GECaM approximation go beyond the purpose of the present work and will be discussed elsewhere.

## 4 Conclusions

This work establishes a flexible and robust framework for studying complex stochastic (Langevin-type) dynamics on networked systems. This is done introducing a novel approxi-

mation scheme for the dynamic cavity method, which is applicable to systems with continuous degrees of freedom and bilinear pairwise coupling between them. By deriving the dynamic cavity equations for a system of linearly-coupled stochastic differential equations through the MSRJD functional theory, we have obtained a path-integral formulation of the dynamic probability measure. This formulation, once represented as a graphical model, enables to identify the cavity marginals with local functionals of single-site trajectories for both the physical variables and their MSRJD conjugate variables. This approach mirrors the quantum cavity method used for quantum many-body systems in the imaginary-time representation.

The main approximation of the method involves a small-coupling expansion of the cavity equations, with a second-order truncation, which corresponds to effectively assuming a Gaussian approximation for the cavity marginals. The resulting integro-differential GECaM equations, depending on cavity versions of response functions and two-time correlation functions, provide an approximate description of the dynamics. For stochastic dynamics on fully-connected graphs, this approximation is equivalent to the extended Plefka expansion and results in a dynamical version of the TAP equations [48]. However, the GECaM equations are also suitable for investigating linearly-coupled SDEs on sparse graphs, where the cavity formulation offers a more precise approximation. In analogy with the standard cavity equations, the GECaM equations serve as an algorithm for studying stochastic dynamics on a specific graph instance. Alternatively, by means of a population dynamics algorithm, these equations can be used to explore general dynamical properties of disordered systems at the level of random graph ensembles.

We applied the GECaM equations to various dynamical problems where the quality of the approximation is controllable. For a linear system of SDEs with additive noise, the GECaM equations provide an exact description of the dynamics, limited by the validity of the local tree-like approximation. This is reflected in the relationship between the GECaM equations for the cavity response functions and those for the resolvent obtained through the cavity method in random matrix theory. Classical results for the calculation of the spectral density of hermitian and non-hermitian random matrices are recovered.

The presence of non-linear forces and state-dependent noise terms in the system of SDEs is addressed using a perturbative closure technique, based on a self-consistent resummation of 1-loop contributions reminiscent of the mode-coupling theory for disorders systems. On RRG, the phase transitions under study have a mean-field nature, but with a non-trivial dependence of the critical point on graph connectivity. In both cases, the perturbative closure method applied to GECaM equations provides a very good description of the dependency of the critical point on the degree $K$ of the RRG. Further studies will be devoted to explore the dependency of the critical behavior on other properties of the graph ensemble, such as degree heterogeneity, perhaps using a population dynamics version of these equations. Finally, we applied the GECaM approximation scheme to the relaxation dynamics of the spherical 2-spin model, where a system of linear SDEs with additive noise is endowed with a self-consistent non-linearity through a global constraint. Despite its simplicity, this model showcases non-trivial dynamical properties like slow relaxation and aging effects. The corresponding GECaM equations appear to provide an accurate description on RRG, suggesting they could be relevant for similar dynamics on more general graphs with local tree-like structures.

Although the current GECaM approach has been effective for the problems under study, a non-perturbative closure scheme appears indispensable to incorporate the local effects of non-linearities without further approximations. One possible improvement could involve using a numerical scheme to sample on-site stochastic trajectories given incoming Gaussian cavity messages, as proposed for random ecosystems [97], and computing updated versions of cavity response and correlation functions from these samples.

A significant limitation of the method is the linear coupling between neighboring degrees of freedom. In many relevant applications, local interactions are not exclusively represented by a weighted sum of the neighboring variables, because of the existence of non-linear activation functions (e.g., in neural networks and learning models) and higher-order interaction patterns (e.g., in ecosystems and reaction networks). Generalizing the path integral approach to model these more complex interactions would necessitate a more intricate structure of the graphical model representation and a more complex functional form for the corresponding cavity messages. This could involve introducing additional auxiliary variables, as demonstrated in [27]. In specific cases, such as multilinear interactions in $p$-spin models with $p \geq 3$, extending the GECaM equation appears straightforward. However, further research is needed to address these limitations and broaden the applicability of the GECaM approach.

## Acknowledgments

The authors would like to thank G. Catania, T. Galla, A. Gamba, A. Gambassi, R. Mulet, P. Sollich for fruitful discussions on the topics of this work.

**Funding informations** This study was carried out within the FAIR - Future Artificial Intelligence Research project and received funding from the European Union NextGenerationEU (Piano Nazionale di Ripresa e Resilienza (PNRR)–Missione 4 Componente 2, Investimento 1.3–D.D. 1555 11/10/2022, PE00000013). This manuscript reflects only the authors' views and opinions, neither the European Union nor the European Commission can be considered responsible for them.

**Computational resources** Computational resources were provided by the SmartData @PoliTO [98] interdepartmental center on Big Data and Data Science.

**Code availability** The code used for this paper can be found at https://github.com/Mattiatarabolo/GaussianExpansionCavityMethod.jl.

## Supplemental material: Gaussian approximation of dynamic cavity equations for linearly-coupled stochastic dynamics

## A General GECaM equations with non-vanishing conjugate variables

We consider the Gaussian Ansatz in Eq. (21) without further assuming that the conjugate averages $\hat{\mu}_{i\backslash j}$ and $B_{i\backslash j}$ vanish due to causality of the dynamics. Writing explicitly the matrix product in Eq. (31), we find the dynamical equations

$$R_{0,i}^{-1} R_{i\backslash j} = \mathbb{I} + 2D\Delta B_{i\backslash j} + \Delta^2 \sum_{k\in\partial i\backslash j} J_{ik} J_{ki} R_{k\backslash i} R_{i\backslash j} + \Delta^2 \sum_{k\in\partial i\backslash j} J_{ik}^2 C_{k\backslash i} B_{i\backslash j}, \tag{A.1}$$

$$R_{0,i}^{-1} C_{i\backslash j} = 2\Delta D R_{i\backslash j}^\top + \Delta^2 \sum_{k\in\partial i\backslash j} J_{ik}^2 C_{k\backslash i} R_{i\backslash j}^\top + \Delta^2 \sum_{k\in\partial i\backslash j} J_{ik} J_{ki} R_{k\backslash i} C_{i\backslash j}, \tag{A.2}$$

$$\left(R_{0,i}^{-1}\right)^\top B_{i\backslash j} = \Delta^2 \sum_{k\in\partial i\backslash j} J_{ik} J_{ki} R_{k\backslash i}^\top B_{i\backslash j} + \Delta^2 \sum_{k\in\partial i\backslash j} J_{ki}^2 B_{k\backslash i} R_{i\backslash j}. \tag{A.3}$$

From Eq. (32) we obtain the dynamical equations

$$R_{0,i}^{-1}\mu_{i\setminus j} = \Delta \sum_{k\in\partial i\setminus j} J_{ik}\mu_{k\setminus i} + \Delta^2 \sum_{k\in\partial i\setminus j} J_{ik}J_{ki}R_{k\setminus i}\mu_{i\setminus j} + 2D\Delta i\hat{\mu}_{i\setminus j}$$
$$+ \Delta^2 \sum_{k\in\partial i\setminus j} J_{ik}^2 C_{k\setminus i}i\hat{\mu}_{i\setminus j}, \tag{A.4}$$

$$\left(R_{0,i}^{-1}\right)i\hat{\mu}_{i\setminus j} = \Delta \sum_{k\in\partial i\setminus j} J_{ki}i\hat{\mu}_{k\setminus i} + \Delta^2 \sum_{k\in\partial i\setminus j} J_{ki}^2 B_{k\setminus i}\mu_{i\setminus j} + \Delta^2 \sum_{k\in\partial i\setminus j} J_{ik}J_{ki}R_{k\setminus i}^\top i\hat{\mu}_{i\setminus j}. \tag{A.5}$$

In the limit $\Delta \to 0$, the non-interacting response becomes $R_{0,i}^{-1}/\Delta \to \left(\frac{\partial}{\partial t} + \lambda_i\right)$ and the complete set of Gaussian Expansion Cavity Method (GECaM) equations is

$$\frac{d}{dt}\mu_{i\setminus j}(t) = -\lambda_i\mu_{i\setminus j}(t) + \sum_{k\in\partial i\setminus j} J_{ik}\mu_{k\setminus i}(t) + \sum_{k\in\partial i\setminus j} \int dt' J_{ik}R_{k\setminus i}(t,t')J_{ki}\mu_{i\setminus j}(t')$$
$$+ 2Di\hat{\mu}_{i\setminus j}(t) + \sum_{k\in\partial i\setminus j} \int dt' J_{ik}^2 C_{k\setminus i}(t,t')i\hat{\mu}_{i\setminus j}(t'), \tag{A.6}$$

$$\frac{d}{dt}i\hat{\mu}_{i\setminus j}(t) = \lambda_i i\hat{\mu}_{i\setminus j}(t) - \sum_{k\in\partial i\setminus j} J_{ki}i\hat{\mu}_{k\setminus i}(t) - \sum_{k\in\partial i\setminus j} \int dt' J_{ki}^2 B_{k\setminus i}(t,t')\mu_{i\setminus j}(t')$$
$$- \sum_{k\in\partial i\setminus j} \int dt' J_{ik}J_{ki}R_{k\setminus i}(t',t)i\hat{\mu}_{i\setminus j}(t'), \tag{A.7}$$

$$\frac{\partial}{\partial t}R_{i\setminus j}(t,t') = -\lambda_i R_{i\setminus j}(t,t') + \sum_{k\in\partial i\setminus j} \int dt'' J_{ik}R_{k\setminus i}(t,t'')J_{ki}R_{i\setminus j}(t'',t')$$
$$+ \sum_{k\in\partial i\setminus j} \int dt'' J_{ik}^2 C_{k\setminus i}(t,t'')B_{i\setminus j}(t'',t') + \delta(t,t'), \tag{A.8}$$

$$\frac{\partial}{\partial t}C_{i\setminus j}(t,t') = -\lambda_i C_{i\setminus j}(t,t') + \sum_{k\in\partial i\setminus j} \int dt'' J_{ik}R_{k\setminus i}(t,t'')J_{ki}C_{i\setminus j}(t'',t') + 2DR_{i\setminus j}(t',t)$$
$$+ \sum_{k\in\partial i\setminus j} \int dt'' R_{i\setminus j}(t',t'')J_{ik}^2 C_{k\setminus i}(t,t''), \tag{A.9}$$

$$\frac{\partial}{\partial t}B_{i\setminus j}(t,t') = \lambda_i B_{i\setminus j}(t,t') - \sum_{k\in\partial i\setminus j} \int dt'' J_{ik}J_{ki}R_{k\setminus i}(t'',t)B_{i\setminus j}(t'',t')$$
$$- \sum_{k\in\partial i\setminus j} \int dt'' J_{ki}^2 B_{k\setminus i}(t,t'')R_{i\setminus j}(t'',t'). \tag{A.10}$$

We can ascertain that if $\hat{\mu}_{k\setminus i}$ and $B_{k\setminus i}$ vanish, the general GECaM equations reduce to the ones reported in the main text. In addition to that, $\hat{\mu}_{i\setminus j}$ and $B_{i\setminus j}$ vanish consequently, and therefore the assumption is self-consistently verified.

# B  Equilibrium cavity equations

We consider the TTI GECaM equations Eqs. (44) and (46)

$$\dot{R}_{i\backslash j}(\tau) = -\lambda_i R_{i\backslash j}(\tau) + \sum_{k\in\partial i\backslash j} J_{ik}J_{ki}\int_0^\tau du R_{k\backslash i}(u)R_{i\backslash j}(\tau-u) + \delta(\tau), \tag{B.1}$$

$$\dot{C}_{i\backslash j}(\tau) = -\lambda_i C_{i\backslash j}(\tau) + \sum_{k\in\partial i\backslash j} J_{ik}J_{ki}\left(\int_\tau^\infty du R_{k\backslash i}(u)C_{i\backslash j}(\tau-u) + \int_0^\tau du R_{k\backslash i}(u)C_{i\backslash j}(\tau-u)\right)$$

$$+ 2DR_{i\backslash j}(-\tau) + \sum_{k\in\partial i\backslash j} J_{ik}^2 \int_\tau^\infty du C_{k\backslash i}(u)R_{i\backslash j}(u-\tau). \tag{B.2}$$

If the interaction matrix $\boldsymbol{J}$ is symmetric, i.e. $J_{ij} = J_{ji}$ for every $i,j = 1,\ldots,N$, the system satisfies detailed balance and it eventually reaches equilibrium after a sufficient long time. Within this regime, applying the Fluctuation Dissipation Theorem (FDT)

$$DR_{i\backslash j}^{\text{eq}}(\tau) = -\dot{C}_{i\backslash j}^{eq}(\tau)\Theta(\tau), \tag{B.3}$$

we obtain an equation for the equilibrium correlations only,

$$\dot{C}_{i\backslash j}^{eq}(\tau)(1-2\Theta(-\tau)) = -\lambda_i C_{i\backslash j}^{eq}(\tau) - \sum_{k\in\partial i\backslash j}\frac{J_{ik}^2}{D}\left(\int_\tau^\infty du \dot{C}_{k\backslash i}^{eq}(u)C_{i\backslash j}^{eq}(\tau-u) + \int_0^\tau du \dot{C}_{k\backslash i}^{eq}(u)C_{i\backslash j}^{eq}(\tau-u)\right)$$

$$- \sum_{k\in\partial i\backslash j}\frac{J_{ik}^2}{D}\int_\tau^\infty du C_{k\backslash i}^{eq}(u)\dot{C}_{i\backslash j}^{eq}(u-\tau). \tag{B.4}$$

Integrating by parts the last integral

$$\int_\tau^\infty du C_{k\backslash i}^{\text{eq}}(u)\dot{C}_{i\backslash j}^{eq}(u-\tau) = C_{k\backslash i}^{\text{eq}}(\infty)C_{i\backslash j}^{eq}(\infty) - C_{k\backslash i}^{\text{eq}}(\tau)C_{i\backslash j}^{eq}(0) - \int_\tau^\infty du \dot{C}_{k\backslash i}^{eq}(u)C_{i\backslash j}^{eq}(u-\tau).$$

The equilibrium correlations decay to zero at infinite time differences $C_{k\backslash i}^{\text{eq}}(\infty)C_{i\backslash j}^{eq}(\infty) = 0$. The cavity equilibrium correlations are therefore obtained by solving the set of equations

$$\text{sgn}(\tau)\dot{C}_{i\backslash j}^{eq}(\tau) = -\lambda_i C_{i\backslash j}^{eq}(\tau) + \sum_{k\in\partial i\backslash j}\frac{J_{ik}^2}{D}\left(C_{k\backslash i}^{\text{eq}}(\tau)C_{i\backslash j}^{eq}(0) - \int_0^\tau du \dot{C}_{k\backslash i}^{eq}(u)C_{i\backslash j}^{eq}(\tau-u)\right). \tag{B.5}$$

The full equilibrium correlations are obtained from the cavity ones as

$$\text{sgn}(\tau)\dot{C}_{i}^{eq}(\tau) = -\lambda_i C_{i}^{eq}(\tau) + \sum_{k\in\partial i}\frac{J_{ik}^2}{D}\left(C_{k\backslash i}^{\text{eq}}(\tau)C_{i}^{eq}(0) - \int_0^\tau du \dot{C}_{k\backslash i}^{eq}(u)C_{i}^{eq}(\tau-u)\right). \tag{B.6}$$

## B.1  Numerical solution

A numerical solution of Eq. Eq. (B.5) can be found by discretizing time with a timestep $\Delta$, i.e. $t = n\Delta$, $n = 0,\ldots,T$ with $T = \mathcal{T}/\Delta$. Within this discretization the equilibrium correlation function becomes a time vector with $T+1$ components $C_{i\backslash j}^{eq,n} = C_{i\backslash j}^{eq}(t = n\Delta)$. Then a

discretized version of Eq. Eq. (B.5) is

$$C_{i\backslash j}^{eq,n+1} = (1 - \lambda_i \Delta) C_{i\backslash j}^{eq,n} + \Delta \sum_{k \in \partial i \backslash j} \frac{J_{ik}^2}{D} C_{k\backslash i}^{eq,n} C_{i\backslash j}^{eq,0}$$

$$- \Delta \sum_{k \in \partial i \backslash j} \frac{J_{ik}^2}{D} \sum_{m=0}^{n-1} \left( C_{k\backslash i}^{eq,m+1} - C_{k\backslash i}^{eq,m} \right) C_{i\backslash j}^{eq,n-m}, \tag{B.7}$$

$$= (1 - \lambda_i \Delta) C_{i\backslash j}^{eq,n} + \Delta \frac{C_{i\backslash j}^{eq,0}}{D} \sum_{k \in \partial i \backslash j} J_{ik}^2 C_{k\backslash i}^{eq,n}$$

$$- \Delta \sum_{m=0}^{n-1} \frac{C_{i\backslash j}^{eq,n-m}}{D} \sum_{k \in \partial i \backslash j} J_{ik}^2 \left( C_{k\backslash i}^{eq,m+1} - C_{k\backslash i}^{eq,m} \right), \tag{B.8}$$

while for the full correlation we obtain

$$C_i^{eq,n+1} = (1 - \lambda_i \Delta) C_i^{eq,n} + \Delta \frac{C_i^{eq,0}}{D} \sum_{k \in \partial i \backslash j} J_{ik}^2 C_{k\backslash i}^{eq,n}$$

$$- \Delta \sum_{m=0}^{n-1} \frac{C_i^{eq,n-m}}{D} \sum_{k \in \partial i \backslash j} J_{ik}^2 \left( C_{k\backslash i}^{eq,m+1} - C_{k\backslash i}^{eq,m} \right). \tag{B.9}$$

# C  Non-Hermitian random matrices

In what follows we will adopt the method proposed in [99] adapting it to our dynamical setting. To maintain generality, we consider complex-valued coupling matrices $\tilde{\boldsymbol{J}} \in \mathbb{C}^{N,N}$. From a dynamical point of view, this is equivalent to duplicating the system introducing an auxiliary degree of freedom $y_i \in \mathbb{C}$ for each node of the graph. The duplicated system evolves according to the system of SDEs

$$\frac{d}{dt} x_i(t) = \sum_j \tilde{J}_{ij} y_j(t) - z y_i(t) + \xi_i(t), \tag{C.1}$$

$$\frac{d}{dt} y_i(t) = -\sum_j \tilde{J}_{ji}^* x_j(t) + z^* x_i(t) + \zeta_i(t), \tag{C.2}$$

where $z \in \mathbb{C}$ is a complex valued parameter, corresponding to the eigenvalue of the coupling matrix. The variables $\xi_i$ and $\zeta_i$ are complex Gaussian noise terms with mean and covariance given by

$$\langle \xi_i(t) \rangle = \langle \zeta_i(t) \rangle = 0, \tag{C.3}$$

$$\langle \xi_i^*(t) \xi_{i'}(t') \rangle = \langle \zeta_i^*(t) \zeta_{i'}(t') \rangle = 2D \delta_{ii'} \delta(t - t'). \tag{C.4}$$

The MSRJD functional integral formalism can now be applied, performing an integration over complex-valued trajectories using the complex Dirac delta function

$$\delta^{(2)}(x) = \delta(\text{Re}\,x)\delta(\text{Im}\,x) \propto \int_{\mathbb{C}} d\hat{x} d\hat{x}^* e^{-\hat{x}^* x + \hat{x} x^*}, \tag{C.5}$$

where $d\hat{x} d\hat{x}^* = d(\text{Re}\,x)d(\text{Im}\,x) = dx^2$ and the integral is taken over the whole complex plane.

The dynamical partition function of the system after discretization of time is

$$
Z \propto \left\langle \int D\vec{x}^2 D\vec{\hat{x}}^2 D\vec{y}^2 D\vec{\hat{y}}^2 \prod_{i,n} \exp\left[ -\left(\hat{x}_i^n\right)^* \left( x_i^{n+1} - x_i^n - \Delta \sum_j \tilde{J}_{ij} y_j^n + \Delta z y_i^n - \Delta \xi_i^n \right) \right.\right.
$$
$$
+ \hat{x}_i^n \left( \left(x_i^{n+1}\right)^* - \left(x_i^n\right)^* - \Delta \sum_j \tilde{J}_{ij}^* \left(y_j^n\right)^* + \Delta z^* \left(y_i^n\right)^* - \left(\Delta \xi_i^n\right)^* \right)
$$
$$
- \left(\hat{y}_i^n\right)^* \left( y_i^{n+1} - y_i^n + \Delta \sum_j \tilde{J}_{ji}^* x_j^n - \Delta z^* x_i^n - \Delta \zeta_i^n \right)
$$
$$
\left.\left.+ \hat{y}_i^n \left( \left(y_i^{n+1}\right)^* - \left(y_i^n\right)^* + \Delta \sum_j \tilde{J}_{ji} \left(x_j^n\right)^* - \Delta z \left(x_i^n\right)^n - \left(\Delta \zeta_i^n\right)^n \right) \right] \right\rangle, \tag{C.6}
$$

where we have introduced $\int D\vec{x}^2 D\vec{\hat{x}}^2 D\vec{y}^2 D\vec{\hat{y}}^2 = \prod_{i,n} \int \left(dx_i^n\right)^2 \left(d\hat{x}_i^n\right)^2 \left(dy_i^n\right)^2 \left(d\hat{y}_i^n\right)^2$. For notational convenience, we set $\Delta = 1$ below. The average over the complex Gaussian noise can be performed applying the identity

$$
\left\langle e^{\vec{\hat{x}}^\dagger \vec{\Delta \xi} - \vec{\Delta \xi}^\dagger \vec{\hat{x}} + \vec{\hat{y}}^\dagger \vec{\Delta \zeta} - \vec{\Delta \zeta}^\dagger \vec{\hat{y}}} \right\rangle = e^{-\vec{\hat{x}}^\dagger g_x \vec{\hat{x}} - \vec{\hat{y}}^\dagger g_y \vec{\hat{y}}}, \tag{C.7}
$$

where $[g_x]_{ii',nn'} = \left\langle \left(\Delta \xi_i^n\right)^* \Delta \xi_{i'}^{n'} \right\rangle = 2D \delta_{ii'} \delta_{nn'} = [g_y]_{ii',nn'} = \left\langle \left(\Delta \zeta_i^n\right)^* \Delta \zeta_{i'}^{n'} \right\rangle$ are the noise covariance matrices. The dynamical partition function can be factorized as

$$
Z \propto \int D\vec{x}^2 D\vec{\hat{x}}^2 D\vec{y}^2 D\vec{\hat{y}}^2 \prod_i \exp\left\{ \sum_n \left[ -\left(\hat{x}_i^n\right)^* \left( x_i^{n+1} - x_i^n + z y_i^n \right) \right.\right.
$$
$$
\left.\left.+ \hat{x}_i^n \left( \left(x_i^{n+1}\right)^* - \left(x_i^n\right)^* + z^* \left(y_i^n\right)^* \right) - 2D \left|\hat{x}_i^n\right|^2 \right] \right\}
$$
$$
\times \prod_{i<j} \exp\left\{ \sum_n \left[ \left(\hat{x}_i^n\right)^* \tilde{J}_{ij} y_j^n + y_i^n \tilde{J}_{ji} \left(\hat{x}_j^n\right)^* - \hat{x}_i^n \tilde{J}_{ij}^* \left(y_j^n\right)^* - \left(y_i^n\right)^* \tilde{J}_{ji}^* \hat{x}_j^n \right] \right\}
$$
$$
\times \prod_i \exp\left\{ \sum_n \left[ -\left(\hat{y}_i^n\right)^* \left( y_i^{n+1} - y_i^n - z^* x_i^n \right) + \hat{y}_i^n \left( \left(y_i^{n+1}\right)^* - \left(y_i^n\right)^* - z \left(x_i^n\right)^* \right) - 2D \left|\hat{y}_i^n\right|^2 \right] \right\}
$$
$$
\times \prod_{i<j} \exp\left\{ \sum_n \left[ -\left(\hat{y}_i^n\right)^* \tilde{J}_{ji}^* x_j^n - x_i^n \tilde{J}_{ij}^* \left(\hat{y}_j^n\right)^* + \hat{y}_i^n \tilde{J}_{ji} \left(x_j^n\right)^* + \left(x_i^n\right)^* \tilde{J}_{ij} \hat{y}_j^n \right] \right\}. \tag{C.8}
$$

It turns out that, in order to properly consider a dynamic cavity Ansatz, the trajectories $x_i$, $\hat{x}_i$, $y_i$ and $\hat{y}_i$ have to be grouped together into a single variable $\mathcal{W}_i^\top = (x_i, \hat{x}_i, y_i, \hat{y}_i)$. We can write the dynamical partition function as a quadratic form

$$
Z \propto \int D\vec{\mathcal{W}}^2 \prod_i e^{-\mathcal{W}_i^\dagger \mathcal{G}_{0,i}^{-1} \mathcal{W}_i} \prod_{i<j} e^{-\mathcal{W}_i^\dagger \tilde{\mathcal{J}}_{ij} \mathcal{W}_j - \mathcal{W}_j^\dagger \tilde{\mathcal{J}}_{ji} \mathcal{W}_i}, \tag{C.9}
$$

where we have introduced $\int D\vec{\mathcal{W}}^2 = \prod_i \int D\mathcal{W}_i^2 = \int D\vec{x}^2 D\vec{\hat{x}}^2 D\vec{y}^2 D\vec{\hat{y}}^2$ and the matrices

$$
\mathcal{G}_{0,i}^{-1} = \begin{pmatrix} \mathsf{G}_{0,i}^{-1} & \mathsf{H} \\ -\mathsf{H}^\dagger & \mathsf{G}_{0,i}^{-1} \end{pmatrix}, \tag{C.10}
$$

$$
\tilde{\mathcal{J}}_{ij} = \begin{pmatrix} 0 & -\tilde{\mathsf{J}}_{ij} \\ \tilde{\mathsf{J}}_{ji}^\dagger & 0 \end{pmatrix} = -\tilde{\mathcal{J}}_{ji}^\dagger, \tag{C.11}
$$

whose elements are the matrices

$$G_{0,i}^{-1} = \begin{pmatrix} 0 & \left[R_{0,i}^{-1}\right]^\dagger \\ R_{0,i}^{-1} & -2D\mathbb{I} \end{pmatrix}, \tag{C.12}$$

$$H = \begin{pmatrix} 0 & z\mathbb{I} \\ z\mathbb{I} & 0 \end{pmatrix}, \tag{C.13}$$

$$\tilde{J}_{ij} = \begin{pmatrix} 0 & \tilde{J}_{ij}\mathbb{I} \\ \tilde{J}_{ij}\mathbb{I} & 0 \end{pmatrix}. \tag{C.14}$$

The cavity messages are

$$c_{i\backslash j}(\mathcal{W}_i) \propto e^{-\mathcal{W}_i^\dagger \mathcal{G}_{0,i}^{-1} \mathcal{W}_i} \prod_{k \in \partial i\backslash j} \int D\mathcal{W}_k^2 \, c_{k\backslash i}(\mathcal{W}_k) e^{-\mathcal{W}_k^\dagger \tilde{\mathcal{J}}_{ki} \mathcal{W}_i - \mathcal{W}_i^\dagger \tilde{\mathcal{J}}_{ik} \mathcal{W}_k}. \tag{C.15}$$

We assume once again that the cavity messages have a Gaussian functional form, that is

$$c_{i\backslash j}(\mathcal{W}_i) \propto e^{-\mathcal{W}_i^\dagger \mathcal{G}_{i\backslash j}^{-1} \mathcal{W}_i}, \tag{C.16}$$

where the cavity propagator is given by

$$\mathcal{G}_{i\backslash j} = \begin{pmatrix} G_{i\backslash j}^{XX} & G_{i\backslash j}^{XY} \\ G_{i\backslash j}^{YX} & G_{i\backslash j}^{YY} \end{pmatrix}, \tag{S12}$$

and $G_{i\backslash j}^{AB}$ is the usual propagator between variables $A_i$ and $B_i$, with A and B being either $X^\top = (x, \hat{x})$ or $Y^\top = (y, \hat{y})$

$$G_{i\backslash j}^{AB} = \begin{pmatrix} \langle a_i b_i^\dagger \rangle_{i\backslash j} & \langle a_i \hat{b}_i^\dagger \rangle_{i\backslash j} \\ \langle \hat{a}_i b_i^\dagger \rangle_{i\backslash j} & \langle \hat{a}_i \hat{b}_i^\dagger \rangle_{i\backslash j} \end{pmatrix} = \begin{pmatrix} C_{i\backslash j}^{AB} & R_{i\backslash j}^{AB} \\ \left(R_{i\backslash j}^{BA}\right)^\dagger & 0 \end{pmatrix}. \tag{C.17}$$

Integrating out the variables $\mathcal{W}_k$, we obtain

$$c_{i\backslash j}(\mathcal{W}_i) \propto e^{-\mathcal{W}_i^\dagger \mathcal{G}_{0,i}^{-1} \mathcal{W}_i} \prod_{k \in \partial i\backslash j} \int D\mathcal{W}_k^2 e^{-\mathcal{W}_k^\dagger \mathcal{G}_{k\backslash i}^{-1} \mathcal{W}_k - \mathcal{W}_k^\dagger \tilde{\mathcal{J}}_{ki} \mathcal{W}_i - \mathcal{W}_i^\dagger \tilde{\mathcal{J}}_{ik} \mathcal{W}_k} \tag{C.18a}$$

$$\propto e^{-\mathcal{W}_i^\dagger \mathcal{G}_{0,i}^{-1} \mathcal{W}_i + \sum_{k \in \partial i\backslash j} \mathcal{W}_i^\dagger \tilde{\mathcal{J}}_{ik} \mathcal{G}_{k\backslash i} \tilde{\mathcal{J}}_{ki} \mathcal{W}_i}, \tag{C.18b}$$

from which we get the extended GECaM matrix equation

$$\mathcal{G}_{i\backslash j}^{-1} = \mathcal{G}_{0,i}^{-1} - \sum_{k \in \partial i\backslash j} \tilde{\mathcal{J}}_{ik} \mathcal{G}_{k\backslash i} \tilde{\mathcal{J}}_{ki}. \tag{C.19}$$

To compute the spectral density, we only need the equations for the response functions. Conveniently expressed in matrix form and in the Laplace space, these equations are

$$R_{i\backslash j}^{-1} = z + \eta I - \sum_{k \in \partial i\backslash j} \tilde{J}_{ik} R_{k\backslash i} \tilde{J}_{ki}, \tag{C.20}$$

where we have introduced the matrices,

$$R_{i\backslash j} = \begin{pmatrix} R_{i\backslash j}^{XX} & R_{i\backslash j}^{XY} \\ R_{i\backslash j}^{YX} & R_{i\backslash j}^{YY} \end{pmatrix}, \tag{C.21}$$

$$z = \begin{pmatrix} 0 & z \\ -z^* & 0 \end{pmatrix}, \tag{C.22}$$

and redefined for convenience the coupling matrix $\tilde{J}_{ij}$ as follows

$$\tilde{J}_{ij} = \begin{pmatrix} 0 & a_{ij} \\ -a_{ji}^* & 0 \end{pmatrix}. \tag{C.23}$$

Equation (C.20) is equivalent to Eq. (64) in [100]. We can therefore compute the spectral distribution of $\boldsymbol{J}$ as

$$\rho_J(z) = \lim_{\eta \to 0} \frac{1}{\pi N} \partial_{z^*} R_i^{\mathsf{YX}}(z - \lambda_i). \tag{C.24}$$

# D Noise-driven phase transition in the Bouchaud-Mézard model

## D.1 Critical point

We consider the integral Eq. (105a) obtained from the Gaussian perturbation closure technique of the BM model on an homogeneous RRG.

$$I = \int_{-\infty}^{\infty} \frac{d\omega}{2\pi} \left[ \frac{i\omega + \lambda - \sqrt{(i\omega + \lambda)^2 - 4(K-1)J^2}}{2(K-1)J^2} \right] \left[ \frac{-i\omega + \lambda - \sqrt{(-i\omega + \lambda)^2 - 4(K-1)J^2}}{2(K-1)J^2} \right] \tag{D.1a}$$

$$= -\frac{2i}{\pi} \int_{-\infty}^{\infty} d(i\omega) \frac{1}{\lambda + i\omega + \sqrt{(\lambda + i\omega)^2 - 4(K-1)J^2}} \frac{1}{\lambda - i\omega + \sqrt{(\lambda - i\omega)^2 - 4(K-1)J^2}}, \tag{D.1b}$$

where $\lambda = KJ$. The integral on the second line can be viewed as a complex integral over the imaginary axis $\mathfrak{Im}$.

$$I = -\frac{2i}{\pi} \int_{\mathfrak{Im}} dz \, F(z), \tag{D.2}$$

where we have defined $F(z) = f(z)f(-z)$ with

$$f(z) = \left( \lambda + z + \sqrt{(\lambda + z)^2 - 4(K-1)J^2} \right). \tag{D.3}$$

$F(z)$ has two symmetric branch cuts associated with the square root validity intervals, i.e $\mathcal{B}_- = [-\lambda - A, -\lambda + A]$ and $\mathcal{B}_+ = [\lambda - A, \lambda + A]$, with $A = 2\sqrt{K-1}J$. The integral along the imaginary axis can be computed from a contour integral on the complex plane in which the complex variable $z$ is integrated along a contour $\mathcal{C}$ that goes along the imaginary axis then closes in a semicircle with a detour to avoid the inclusion of the $\mathcal{B}_-$ branch cut. Figure 7 shows that the contour $\mathcal{C}$ can be decomposed into elementary paths.

More precisely,

$$\oint_{\mathcal{C}} dz \, F(z) = \left( \int_\gamma + \int_{\gamma_R^+} + \int_{\gamma_R^-} + \int_{\gamma_\varepsilon^+} + \int_{\gamma_\varepsilon^-} + \int_{\gamma_\varepsilon'} + \int_{\gamma_1} + \int_{\gamma_2} + \int_{\gamma_3} + \int_{\gamma_4} \right) dz \, F(z) = 0, \tag{D.4}$$

since the contour contains no singularity or branch cuts. Taking the appropriate limits we can therefore write

$$I = \frac{2i}{\pi} \left[ \lim_{R \to +\infty} \left( \int_{\gamma_R^+} + \int_{\gamma_R^-} \right) + \lim_{\varepsilon \to 0^+} \left( \int_{\gamma_\varepsilon^+} + \int_{\gamma_\varepsilon^-} + \int_{\gamma_\varepsilon'} \right) + \int_{\gamma_1} + \int_{\gamma_2} + \int_{\gamma_3} + \int_{\gamma_4} \right] dz \, F(z). \tag{D.5}$$

It can be shown that the only non-vanishing integrals are those along $\gamma_1$, $\gamma_2$, $\gamma_3$ and $\gamma_4$, however the first two cancels out due to analiticity. The integrals along $\gamma_3$ and $\gamma_4$ instead sum

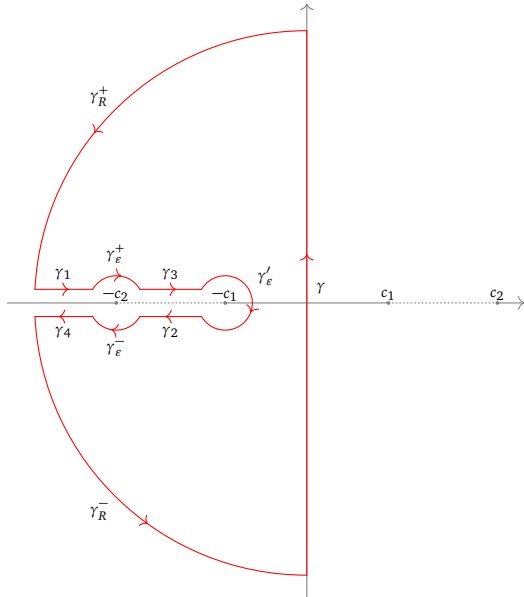

Figure 7: Contour used for the computation of Eq. (D.2). The paths $\gamma_R^+$ and $\gamma_R^-$ are circular arcs of radius $R \to \infty$, while $\gamma_\varepsilon^+$, $\gamma_\varepsilon^-$ and $\gamma_\varepsilon'$ are circular arcs of radius $\varepsilon \to 0$. It can be showed that all of them vanish in the proper limit.

up, since the square roots have opposite signs on the two sides of the branch cut.

$$\int_{\gamma_3} dz\, F(z) = \int_{-\lambda-A}^{-\lambda+A} dx \left[ \frac{1}{\lambda+x+\mathrm{i}\sqrt{A^2-(\lambda+x)^2}} \right]\left[ \frac{1}{\lambda-x+\sqrt{(\lambda-x)^2-A^2}} \right], \quad \text{(D.6a)}$$

$$\int_{\gamma_4} dz\, F(z) = \int_{-\lambda+A}^{-\lambda-A} dx \left[ \frac{1}{\lambda+x-\mathrm{i}\sqrt{A^2-(\lambda+x)^2}} \right]\left[ \frac{1}{\lambda-x+\sqrt{(\lambda-x)^2-A^2}} \right], \quad \text{(D.6b)}$$

where we used the parametrization $z = -xe^{\mathrm{i}\pi}$ for the first integral and $z = -xe^{-\mathrm{i}\pi}$ for the second one.

Gathering all terms,

$$I = \frac{2\mathrm{i}}{\pi}\int_{-\lambda-A}^{-\lambda+A} dx \left[ \frac{1}{\lambda+x+\mathrm{i}\sqrt{A^2-(\lambda+x)^2}} - \frac{1}{\lambda+x-\mathrm{i}\sqrt{A^2-(\lambda+x)^2}} \right]\left[ \frac{1}{\lambda-x+\sqrt{(\lambda-x)^2-A^2}} \right] \quad \text{(D.7a)}$$

$$= \frac{2\mathrm{i}}{\pi}\int_{-\lambda-A}^{-\lambda+A} dx \left[ \frac{-2\mathrm{i}\sqrt{A^2-(\lambda+x)^2}}{\left(\lambda+x+\mathrm{i}\sqrt{A^2-(1+x)^2}\right)\left(\lambda+x-\mathrm{i}\sqrt{A^2-(\lambda+x)^2}\right)} \right]\left[ \frac{1}{\lambda-x+\sqrt{(\lambda-x)^2-A^2}} \right] \quad \text{(D.7b)}$$

$$= \frac{4}{\pi A^2}\int_{-\lambda-A}^{-\lambda+A} dx \left[ \frac{\sqrt{A^2-(\lambda+x)^2}}{\lambda-x+\sqrt{(\lambda-x)^2-A^2}} \right]. \quad \text{(D.7c)}$$

Changing variable $u = (\lambda+x)/A$, the integral is simplified as follows

$$I = I_K = \frac{2}{\pi\sqrt{K-1}J}\int_{-1}^{1} du\, \frac{\sqrt{1-u^2}}{\frac{K}{\sqrt{K-1}}-u+\sqrt{\left(\frac{K}{\sqrt{K-1}}-u\right)^2-1}}, \quad \text{(D.8)}$$

which gives the correct result in the FC limit after rescaling $J = J/K$ and taking the limit $K \to \infty$

$$I_\infty \approx \frac{2}{\pi J}\frac{K}{\sqrt{K-1}}\int_{-1}^{1} du\, \frac{\sqrt{1-u^2}}{2\frac{K}{\sqrt{K-1}}} = \frac{1}{2J}. \quad \text{(D.9)}$$

## D.2 Adiabatic and independent assumption for GECaM

The approximation method proposed in [101–103] is based on the combination of two assumptions: (1) the "local field" $\sum_{j\in\partial i} x_j$ acting on $i$ is assumed to change much more slowly than $x_i$ (*adiabatic assumption*); (2) the neighbouring variables $x_j$ for $j \in \partial i$ are assumed to be independent, so that the local field satisfies the Central Limit Theorem and converges to a Gaussian distribution (*independence assumption*). In this Section, we show that this approximation naturally follows from the GECaM approximation under specific conditions. To simplify the calculations, we will only consider the case of a RRG with degree $K$.

Employing as usual the Gaussian Ansatz in Eq. (21), the local marginal of node $i$ with a perturbation action $S_i^{NG}(\mathsf{X}_i)$ can be written as follows

$$c_i(\mathsf{X}_i) \propto e^{-\frac{1}{2}\mathsf{X}_i^\top \mathsf{G}_{0,i}^{-1}\mathsf{X}_i - S_i^{NG}(\mathsf{X}_i)} \prod_{k\in\partial i} \int D\mathsf{X}_k e^{-\frac{1}{2}(\mathsf{X}_k-\mathsf{M}_{k\backslash i})^\top \mathsf{G}_{k\backslash i}^{-1}(\mathsf{X}_k-\mathsf{M}_{k\backslash i})} e^{-\mathsf{X}_i^\top \mathsf{J}_{ik}\mathsf{X}_k}. \tag{D.10}$$

We also assume that all couplings are the same, i.e. $\mathsf{J}_{ik} = \mathsf{J}$, and introduce a local field $K\mathsf{H}_i = \sum_{j\in\partial i}\mathsf{X}_j$. The previous expression becomes

$$c_i(\mathsf{X}_i) \propto e^{-\frac{1}{2}\mathsf{X}_i^\top \mathsf{G}_{0,i}^{-1}\mathsf{X}_i - S_i^{NG}(\mathsf{X}_i)} \prod_{k\in\partial i} \int D\mathsf{X}_k e^{-\frac{1}{2}(\mathsf{X}_k-\mathsf{M}_{k\backslash i})^\top \mathsf{G}_{k\backslash i}^{-1}(\mathsf{X}_k-\mathsf{M}_{k\backslash i})} e^{-\mathsf{X}_i^\top \mathsf{J}\mathsf{X}_k}$$

$$\times \int D\mathsf{H}_i \delta\left(K\mathsf{H}_i - \sum_{j\in\partial i}\mathsf{X}_j\right) \tag{D.11a}$$

$$\propto e^{-\frac{1}{2}\mathsf{X}_i^\top \mathsf{G}_{0,i}^{-1}\mathsf{X}_i - S_i^{NG}(\mathsf{X}_i)} \prod_{k\in\partial i} \int D\mathsf{X}_k e^{-\frac{1}{2}(\mathsf{X}_k-\mathsf{M}_{k\backslash i})^\top \mathsf{G}_{k\backslash i}^{-1}(\mathsf{X}_k-\mathsf{M}_{k\backslash i})} e^{-\mathsf{X}_i^\top \mathsf{J}\mathsf{X}_k}$$

$$\times \int D\mathsf{H}_i D\hat{\mathsf{H}}_i e^{-i\hat{\mathsf{H}}_i^\top(K\mathsf{H}_i - \sum_{k\in\partial i}\mathsf{X}_k)} \tag{D.11b}$$

$$\propto e^{-\frac{1}{2}\mathsf{X}_i^\top \mathsf{G}_{0,i}^{-1}\mathsf{X}_i - S_i^{NG}(\mathsf{X}_i)} \int D\mathsf{H}_i D\hat{\mathsf{H}}_i e^{-Ki\hat{\mathsf{H}}_i^\top \mathsf{H}_i}$$

$$\times \prod_{k\in\partial i} \int D\mathsf{X}_k e^{-\frac{1}{2}(\mathsf{X}_k-\mathsf{M}_{k\backslash i})^\top \mathsf{G}_{k\backslash i}^{-1}(\mathsf{X}_k-\mathsf{M}_{k\backslash i})+(i\hat{\mathsf{H}}_i - \mathsf{J}^\top \mathsf{X}_i)^\top \mathsf{X}_k}. \tag{D.11c}$$

Performing the Gaussian integral and introducing $\mathsf{M}_i = \sum_{k\in\partial i}\mathsf{M}_{k\backslash i}$ and $\mathsf{G}_i = \sum_{k\in\partial i}\mathsf{G}_{k\backslash i}$, we get

$$c_i(\mathsf{X}_i) \propto e^{-\frac{1}{2}\mathsf{X}_i^\top \mathsf{G}_{0,i}^{-1}\mathsf{X}_i - S_i^{NG}(\mathsf{X}_i)} \int D\mathsf{H}_i D\hat{\mathsf{H}}_i e^{-Ki\hat{\mathsf{H}}_i^\top \mathsf{H}_i + (i\hat{\mathsf{H}}_i - \mathsf{J}^\top \mathsf{X}_i)^\top \mathsf{M}_i + \frac{1}{2}(i\hat{\mathsf{H}}_i - \mathsf{J}^\top \mathsf{X}_i)^\top \mathsf{G}_i(i\hat{\mathsf{H}}_i - \mathsf{J}^\top \mathsf{X}_i)}. \tag{D.12}$$

The integral in $\hat{\mathsf{H}}$ is also Gaussian and it can be done to obtain

$$c_i(\mathsf{X}_i) \propto \int D\mathsf{H}_i e^{-\frac{1}{2}\mathsf{X}_i^\top \mathsf{G}_{0,i}^{-1}\mathsf{X}_i - K\mathsf{X}_i^\top \mathsf{J}\mathsf{H}_i - S_i^{NG}(\mathsf{X}_i)} e^{-\frac{1}{2}(K\mathsf{H}_i - \mathsf{M}_i)^\top \mathsf{G}_i^{-1}(K\mathsf{H}_i - \mathsf{M}_i)}. \tag{D.13}$$

The local marginal can be written in a conditional form, which reminds of the previously discussed adiabatic assumption,

$$c_i(\mathsf{X}_i) = \int D\mathsf{H}_i c_i(\mathsf{X}_i|\mathsf{H}_i) p(\mathsf{H}_i), \tag{D.14}$$

$$c_i(\mathsf{X}_i|\mathsf{H}_i) = \frac{1}{Z_\mathsf{X}(\mathsf{H}_i)} e^{-\frac{1}{2}\mathsf{X}_i^\top \mathsf{G}_{0,i}^{-1}\mathsf{X}_i - K\mathsf{X}_i^\top \mathsf{J}\mathsf{H}_i - S_i^{NG}(\mathsf{X}_i)}, \tag{D.15}$$

$$p(\mathsf{H}_i) = \frac{1}{Z_\mathsf{H}} e^{-\frac{1}{2}(K\mathsf{H}_i - \mathsf{M}_i)^\top \mathsf{G}_i^{-1}(K\mathsf{H}_i - \mathsf{M}_i)}, \tag{D.16}$$

where the partition functions are

$$Z_X(H_i) = \int DX_i e^{-\frac{1}{2}X_i^\top G_{0,i}^{-1} X_i - K X_i^\top J H_i - S_i^{NG}(X_i)}, \tag{D.17}$$

$$Z_H = \left((2\pi)^{2(T+1)} \det G_i\right)^{-1/2}. \tag{D.18}$$

In the case of a random regular graph with homogeneous coupling constants, there is no distinction between nodes in the thermodynamic limit, therefore we drop all the corresponding indices. It follows that the contributions from the neighbours are the same

$$M_i = \sum_{k \in \partial i} M_{k\backslash i} = K M_{\text{cav}}, \qquad G_i = \sum_{k \in \partial i} G_{k\backslash i} = K G_{\text{cav}}, \tag{D.19}$$

therefore $H^\top = (h_1, h_2)$ is a local field with Gaussian statistics, with mean $\langle H \rangle = M_{\text{cav}}$ and variance $\langle H^2 \rangle - \langle H \rangle^2 = G_{\text{cav}}/K$.

The first two moments of $x$ can be computed as

$$\langle x(t) \rangle = \int DX c(X) x(t) \tag{D.20a}$$

$$= \int DH p(H) \int DX \frac{x(t)}{Z_X(H)} e^{-\frac{1}{2}X^\top G_0^{-1} X - K X^\top J H - S^{NG}(X)} \tag{D.20b}$$

$$= -\frac{1}{KJ} \left\langle \frac{1}{Z_X(H)} \frac{\delta}{\delta h_2(t)} Z_X(H) \right\rangle_H, \tag{D.20c}$$

and

$$\langle x^2(t) \rangle = \int DX c(X) x^2(t) \tag{D.21a}$$

$$= \int DH p(H) \int DX \frac{x^2(t)}{Z_X(H)} e^{-\frac{1}{2}X^\top G_0^{-1} X - K X^\top J H - S^{NG}(X)} \tag{D.21b}$$

$$= \frac{1}{(KJ)^2} \left\langle \frac{1}{Z_X(H)} \frac{\delta^2}{\delta h_2^2(t)} Z_X(H) \right\rangle_H. \tag{D.21c}$$

### D.2.1 Linear dynamics with additive noise

We now focus on the simplest case of a linear dynamics with additive noise ($S^{NG} = 0$), where $\lambda = KJ$ is the drift coefficient and $D$ is the noise coefficient. The partition function is easily obtained as a Gaussian integral

$$Z_X(H) = \int DX e^{-\frac{1}{2}X^\top G_0^{-1} X - K X^\top J H} \propto e^{\frac{K}{2}(J^\top H)^\top G_0(J^\top H)}, \tag{D.22}$$

where $G_0$ is the free particle propagator

$$G_0(t, t') = \begin{pmatrix} C_f(t, t') & R_f(t, t') \\ R_f(t', t) & 0 \end{pmatrix}, \tag{D.23}$$

with $C_f(t, t') = D/\lambda \exp(-\lambda|t - t'|)$ being the free particle correlation function and $R_f(t, t') = \Theta(t - t') \exp(-\lambda(t - t'))$ the free particle response function. It follows that the partition function can be written as

$$Z_X(H) \propto \exp\left\{ \frac{(KJ)^2}{2} \int dt \int dt' \left( h_2(t) C_f(t, t') h_2(t') + 2 h_2(t) R_f(t, t') h_1(t') \right) \right\}, \tag{D.24}$$

from which the first and second moments of $x$ follow

$$\langle x(t)\rangle = -\frac{1}{KJ}\left\langle \frac{1}{Z_X(H)}\frac{\delta}{\delta h_2(t)}Z_X(H)\right\rangle_H \tag{D.25a}$$

$$= -\frac{KJ}{2}\int dt'\left(2C_f(t,t')\langle h_2(t')\rangle_H + 2R_f(t,t')\langle h_1(t')\rangle_H\right) \tag{D.25b}$$

$$= -KJ\int dt'\left(C_f(t,t')i\hat{\mu}_{cav}(t')\rangle_H + R_f(t,t')\mu_{cav}(t')\right) \tag{D.25c}$$

$$= -KJ\int dt' R_f(t,t')\mu_{cav}(t'), \tag{D.25d}$$

$$\langle x^2(t)\rangle = \frac{1}{(KJ)^2}\left\langle \frac{1}{Z_X(H)}\frac{\delta^2}{\delta h_2^2(t)}Z_X(H)\right\rangle_H \tag{D.25e}$$

$$= C_f(t,t) + \frac{(KJ)^2}{4}\left\langle \left[\int dt'\left(2C_f(t,t')h_2(t') + 2R_f(t,t')h_1(t')\right)\right]^2\right\rangle_H \tag{D.25f}$$

$$= C_f(t,t) + \frac{(KJ)^2}{4}\int dt'dt''\left(4C_f(t,t')C_f(t,t'')\langle h_2(t')h_2(t'')\rangle_H\right.$$
$$\left. +4R_f(t,t')R_f(t,t'')\langle h_1(t')h_1(t'')\rangle_H + 2C_f(t,t')R_f(t,t'')\langle h_2(t')h_1(t'')\rangle_H\right.$$
$$\left. + 2R_f(t,t')C_f(t,t'')\langle h_1(t')h_2(t'')\rangle_H\right) \tag{D.25g}$$

$$= C_f(t,t) + (KJ)^2\int dt'dt''\left(R_f(t,t')R_f(t,t'')\frac{\tilde{C}_{cav}(t',t'')}{K}\right.$$
$$\left. +R_f(t,t')C_f(t,t')\frac{R_{cav}(t',t'')}{K}\right), \tag{D.25h}$$

where $C_{cav}^{dc}(t',t'') = K\langle h_1(t')h_1(t'')\rangle_H$ is the cavity disconnected correlation. In summary, in the stationary state,

$$s^2 = \frac{D}{\lambda} + (KJ)^2\int_0^\infty d\tau \int_0^\infty ds\left(R_f(\tau)R_f(s)\frac{C_{cav}(s-\tau)}{K} + R_f(\tau)C_f(s)\frac{R_{cav}(s-\tau)}{K}\right). \tag{D.26}$$

It can be shown that the "adiabatic and independent" approximation of [101] corresponds to approximating the cavity quantities by the full quantities at zero time differences, i.e $C_{cav}(s-\tau) \approx C(0) = s^2$ and $R_{cav}(s-\tau) \approx R(0) = 0$, so that we obtain

$$s^2 = \frac{D}{\lambda} + \frac{s^2}{K}(KJ)^2\int_0^\infty d\tau e^{-\lambda\tau}\int_0^\infty ds e^{-\lambda s} \tag{D.27a}$$

$$= \frac{D}{\lambda} + \frac{s^2}{K}\left(\frac{KJ}{\lambda}\right)^2, \tag{D.27b}$$

and, since $\lambda = KJ$, we get the relation

$$s^2 = \frac{D}{J(K-1)}, \tag{D.28}$$

which can be obtained using the approximation in [101].

### D.2.2 Bouchaud-Mézard model

We apply the previous approximation to the BM model, in which case $\lambda = KJ$ and $D = 0$ in the Gaussian part of the action. The interacting part of the action is non-Gaussian

$$S^{NG}(X) = -\frac{1}{2}\sigma^2\int dt x^2(t)(i\hat{x}(t))^2 = -\frac{1}{8}\sigma^2\left(X^\top\sigma_1 X\right)^2, \tag{D.29}$$

and it has to be computed perturbatively. In order to properly perturb the system, we have to rescale the variables as usual, defining $X^\top = (1 + \phi, \hat{\phi})$. The field $K\tilde{H}_i = \sum_{j \in \partial i} \Phi_j$ has now a different average. The site label will be dropped in the rest of the calculation. The field $\tilde{H}^\top = (h_1 - 1, h_2)$ is a Gaussian random variable with mean $\langle \tilde{H} \rangle = \tilde{M}_{\text{cav}}$ and variance $\langle \tilde{H}^2 \rangle - \langle \tilde{H} \rangle^2 = G_{\text{cav}}/K$, where $\tilde{M}_{\text{cav}}^\top = (\mu_{\text{cav}} - 1, 0)$. In summary the distribution of $\Phi$ is

$$c(\Phi) = \int D\tilde{H} c(\Phi|\tilde{H}) p(\tilde{H}), \tag{D.30}$$

$$c(\Phi|\tilde{H}) = \frac{1}{Z_\Phi(\tilde{H})} e^{-\frac{1}{2}\Phi^\top G_0^{-1}\Phi - K\Phi^\top J\tilde{H} - S^{NG}(\Phi)}, \tag{D.31}$$

$$p(\tilde{H}) = \frac{1}{Z_{\tilde{H}}} e^{-\frac{1}{2}(\tilde{H} - \tilde{M}_{\text{cav}})^\top K G_{\text{cav}}^{-1}(\tilde{H} - \tilde{M}_{\text{cav}})}, \tag{D.32}$$

where the partition functions are

$$Z_\Phi(\tilde{H}) = \int D\Phi \, e^{-\frac{1}{2}\Phi^\top G_0^{-1}\Phi - K\Phi^\top J\tilde{H} - S^{NG}(\Phi)}, \tag{D.33}$$

$$Z_{\tilde{H}} = \left( \left( \frac{2\pi}{K} \right)^{2(T+1)} \det G_{\text{cav}} \right)^{-1/2}. \tag{D.34}$$

In terms of the field $\Phi$, the interacting part of the action can be written as

$$
\begin{aligned}
S^{NG}(\Phi) &= -\frac{1}{2}\sigma^2 \int dt (1 + \phi(t))^2 \hat{\phi}(t)^2 \\
&= -\frac{\sigma^2}{2} \int dt \, \hat{\phi}^2(t) - \sigma^2 \int dt \, \hat{\phi}^2(t)\phi(t) - \frac{\sigma^2}{2} \int dt \, \hat{\phi}^2(t)\phi^2(t).
\end{aligned}
\tag{D.35}
$$

The partition function $Z_\Phi(\tilde{H})$ can be computed perturbatively. At first order in $\sigma^2$

$$
\begin{aligned}
Z_\Phi(\tilde{H}) &\approx \int D\Phi \, e^{-\frac{1}{2}\Phi^\top G_0^{-1}\Phi - K\Phi^\top J\tilde{H}} \left( 1 + \frac{\sigma^2}{2} \int dt \, \hat{\phi}^2(t) + \sigma^2 \int dt \, \hat{\phi}^2(t)\phi(t) \right. \\
&\qquad\qquad\qquad\qquad\qquad\qquad\qquad \left. + \frac{\sigma^2}{2} \int dt \, \hat{\phi}^2(t)\phi^2(t) \right)
\end{aligned}
\tag{D.36a}
$$

$$
\begin{aligned}
&= \left\{ 1 + \frac{\sigma^2}{2} \frac{1}{(KJ)^2} \int dt \frac{\delta^2}{\delta \tilde{h}_1^2(t)} - \sigma^2 \frac{1}{(KJ)^3} \int dt \frac{\delta^3}{\delta \tilde{h}_2(t)\delta \tilde{h}_1^2(t)} \right. \\
&\qquad \left. + \frac{\sigma^2}{2} \frac{1}{(KJ)^4} \int dt \frac{\delta^4}{\delta \tilde{h}_2^2(t)\delta \tilde{h}_1^2(t)} \right\} Z_\Phi^0(\tilde{H})
\end{aligned}
\tag{D.36b}
$$

$$= \left\{ 1 + \frac{\sigma^2}{2}\hat{T}_1 - \sigma^2 \hat{T}_2 + \frac{\sigma^2}{2}\hat{T}_3 \right\} Z_\Phi^0(\tilde{H}), \tag{D.36c}$$

where $\hat{T}_1$, $\hat{T}_2$ and $\hat{T}_3$ are operators acting on the Gaussian single particle partition function

$$Z_\Phi^0(\tilde{H}) = \int D\Phi \, e^{-\frac{1}{2}\Phi^\top G_0^{-1}\Phi - K\Phi^\top J\tilde{H}} = e^{-\tilde{S}^0(\tilde{H})}, \tag{D.37}$$

$$\tilde{S}^0(\tilde{H}) = -(KJ)^2 \int dt_1 \int dt_2 \, \tilde{h}_2(t_1) R_f(t_1, t_2) \tilde{h}_1(t_2), \tag{D.38}$$

where we have used $C_f(t, t') = 0$ for the BM model. The operators act on it as

$$
\begin{aligned}
\hat{T}_1 Z_\Phi^0(\tilde{\mathsf{H}}) &= (KJ)^2 \int dt \int dt_1 R_f(t_1, t) \tilde{h}_2(t_1) \int dt_2 R_f(t_2, t) \tilde{h}_2(t_2) Z_\Phi^0(\tilde{\mathsf{H}}) \\
&= T_1(\tilde{\mathsf{H}}) Z_\Phi^0(\tilde{\mathsf{H}}),
\end{aligned}
\tag{D.39}
$$

$$
\begin{aligned}
\hat{T}_2 Z_\Phi^0(\tilde{\mathsf{H}}) &= (KJ)^3 \int dt \int dt_1 R_f(t_1, t) \tilde{h}_2(t_1) \int dt_2 R_f(t_2, t) \tilde{h}_2(t_2) \int dt_3 R_f(t, t_3) \tilde{h}_1(t_3) Z_\Phi^0(\tilde{\mathsf{H}}) \\
&= T_2(\tilde{\mathsf{H}}) Z_\Phi^0(\tilde{\mathsf{H}}),
\end{aligned}
\tag{D.40}
$$

$$
\begin{aligned}
\hat{T}_3 Z_\Phi^0(\tilde{\mathsf{H}}) &= (KJ)^4 \int dt \int dt_1 R_f(t_1, t) \tilde{h}_2(t_1) \int dt_2 R_f(t_2, t) \tilde{h}_2(t_2) \\
&\quad \times \int dt_3 \tilde{h}_1(t_3) R_f(t, t_3) \int dt_4 R_f(t, t_4) \tilde{h}_1(t_4) Z_\Phi^0(\tilde{\mathsf{H}}) \\
&= T_3(\tilde{\mathsf{H}}) Z_\Phi^0(\tilde{\mathsf{H}}),
\end{aligned}
\tag{D.41}
$$

where we have used $R_f(t, t) = 0$ in the Itô case. In summary,

$$
Z_\Phi(\tilde{\mathsf{H}}) \approx Z_\Phi^0(\tilde{\mathsf{H}}) \left( 1 + \frac{\sigma^2}{2} T_1(\tilde{\mathsf{H}}) - \sigma^2 T_2(\tilde{\mathsf{H}}) + \frac{\sigma^2}{2} T_3(\tilde{\mathsf{H}}) \right)
\tag{D.42a}
$$

$$
\approx e^{-\tilde{S}^0(\tilde{\mathsf{H}}) + \frac{\sigma^2}{2} T_1(\tilde{\mathsf{H}}) - \sigma^2 T_2(\tilde{\mathsf{H}}) + \frac{\sigma^2}{2} T_3(\tilde{\mathsf{H}})}
\tag{D.42b}
$$

$$
= e^{-\tilde{S}(\tilde{\mathsf{H}})}.
\tag{D.42c}
$$

It is now possible to compute the first and second moments of the field $\phi$. We neglect contributions in the local self-consistent field beyond second order. The mean is

$$
\langle \phi(t) \rangle = -\frac{1}{KJ} \left\langle \frac{1}{Z_\Phi(\tilde{\mathsf{H}})} \frac{\delta}{\delta \tilde{h}_2(t)} Z_\Phi(\tilde{\mathsf{H}}) \right\rangle_{\tilde{\mathsf{H}}}
\tag{D.43a}
$$

$$
= -(KJ) \left( \int dt' R_f(t, t') \langle \tilde{h}_1(t') \rangle_{\tilde{\mathsf{H}}} - 2\sigma^2 \int dt_1 R_f(t, t_1) \int dt_2 R_f(t_2, t_1) \right.
$$
$$
\left. \times \int dt_3 R_f(t_1, t_3) \langle \tilde{h}_2(t_2) \tilde{h}_1(t_3) \rangle_{\tilde{\mathsf{H}}} \right)
\tag{D.43b}
$$

$$
= 2\sigma^2 (KJ) \int dt_1 R_f(t, t_1) \int dt_2 R_f(t_2, t_1) \int dt_3 R_f(t_1, t_3) \frac{R_{\text{cav}}(t_3, t_2)}{K} = 0,
\tag{D.43c}
$$

due to causality ($R_f(t, t') = R_{\text{cav}}(t, t') = 0$ if $t' > t$). Computing the second moment is more complicated. We use that

$$
\langle \phi(t)^2 \rangle = \frac{1}{(KJ)^2} \left\langle \frac{1}{Z_\Phi(\tilde{\mathsf{H}})} \frac{\delta^2}{\delta \tilde{h}_2^2(t)} Z_\Phi(\tilde{\mathsf{H}}) \right\rangle_{\tilde{\mathsf{H}}}
\tag{D.44a}
$$

$$
= \frac{1}{(KJ)^2} \left[ \left\langle \left( \frac{\delta}{\delta \tilde{h}_2(t)} \tilde{S}(\tilde{\mathsf{H}}) \right)^2 \right\rangle_{\tilde{\mathsf{H}}} - \left\langle \frac{\delta^2}{\delta \tilde{h}_2^2(t)} \tilde{S}(\tilde{\mathsf{H}}) \right\rangle_{\tilde{\mathsf{H}}} \right],
\tag{D.44b}
$$

where the first average is given by

$$\left\langle\left(\frac{\delta}{\delta\tilde{h}_2(t)}\tilde{S}(\tilde{\mathsf{H}})\right)^2\right\rangle_{\tilde{\mathsf{H}}} = (KJ)^4\left\{\int dt_1 \int dt_2 R_{\mathrm f}(t,t_1)R_{\mathrm f}(t,t_2)\langle\tilde{h}_1(t_1)\tilde{h}_1(t_2)\rangle_{\tilde{\mathsf{H}}}\right. \tag{D.45a}$$

$$\left.+2\sigma^2\int dt_1 \int dt_2 R_{\mathrm f}(t,t_2)\int dt_3 R_{\mathrm f}(t,t_3)R_{\mathrm f}(t_1,t_3)\langle\tilde{h}_2(t_1)\tilde{h}_1(t_2)\rangle_{\tilde{\mathsf{H}}}\right\}$$

$$= (KJ)^4\left\{\int dt_1 \int dt_2 R_{\mathrm f}(t,t_1)R_{\mathrm f}(t,t_2)\frac{C_{\mathrm{cav}}(t_1,t_2)}{K}\right. \tag{D.45b}$$

$$\left.+2\sigma^2\int dt_1 \int dt_2 R_{\mathrm f}(t,t_2)\int dt_3 R_{\mathrm f}(t,t_3)R_{\mathrm f}(t_1,t_3)\frac{R_{\mathrm{cav}}(t_2,t_1)}{K}\right\},$$

and the second one is

$$\left\langle\frac{\delta^2}{\delta\tilde{h}_2^2(t)}\tilde{S}(\tilde{\mathsf{H}})\right\rangle_{\tilde{\mathsf{H}}} = (KJ)^2\sigma^2\int dt_1 R_{\mathrm f}^2(t,t_1)$$

$$\times\left(1+(KJ)^2\int dt_2 \int dt_3 R_{\mathrm f}(t_1,t_2)R_{\mathrm f}(t_1,t_3)\langle\tilde{h}_1(t_2)\tilde{h}_1(t_3)\rangle_{\tilde{\mathsf{H}}}\right) \tag{D.46a}$$

$$= (KJ)^2\sigma^2\int dt_1 R_{\mathrm f}^2(t,t_1)$$

$$\times\left(1+(KJ)^2\int dt_2 \int dt_3 R_{\mathrm f}(t_1,t_2)R_{\mathrm f}(t_1,t_3)\frac{C_{\mathrm{cav}}(t_1,t_2)}{K}\right). \tag{D.46b}$$

In summary, at stationarity,

$$s^2 = \langle x^2\rangle - \langle x\rangle^2 = \langle\phi^2\rangle \tag{D.47a}$$

$$= \sigma^2\int_0^\infty d\tau R_{\mathrm f}^2(\tau) + (KJ)^2\int_0^\infty d\tau R_{\mathrm f}(\tau)\int_0^\infty ds R_{\mathrm f}(s)\frac{C_{\mathrm{cav}}(s-\tau)}{K}$$

$$+\sigma^2(KJ)^2\int_0^\infty d\tau\int_0^\infty ds\int_0^{\min(\tau,s)}d\theta R_{\mathrm f}^2(\theta)R_{\mathrm f}(\tau-\theta)R_{\mathrm f}(s-\theta)\frac{C_{\mathrm{cav}}(\tau-\theta)}{K}$$

$$+2\sigma^2(KJ)^2\int_0^\infty d\tau R_{\mathrm f}(\tau)\int_0^\tau ds R_{\mathrm f}(\tau-s)\int_0^s d\theta R_{\mathrm f}(\theta)\frac{R_{\mathrm{cav}}(s-\theta)}{K}. \tag{D.47b}$$

Making the choice of response and correlation functions that corresponds to the adiabatic and independent approximation of [101], i.e setting $C_{\mathrm{cav}}(s-\tau)\approx C(0)=s^2$ and $R_{\mathrm{cav}}(s-\tau)\approx R(0)=0$, we obtain

$$s^2 = \frac{\sigma^2}{2KJ}+\left(1+\frac{\sigma^2}{2KJ}\right)\frac{s^2}{K}. \tag{D.48}$$

This formula leads to a wrong prediction of the critical point $\sigma_c^2 = 2KJ(K-1)$. This is due to the fact we neglected relevant higher-order contributions in the loop expansion generated by the noise term. Performing the resummation of all terms of the geometric series expansion containing enchained 1-loops diagrams only, the previous relation becomes

$$s^2 = \left(1+\frac{\sigma^2}{2KJ}+\left(\frac{\sigma^2}{2KJ}\right)^2+\ldots\right)\left(1+\frac{s^2}{K}\right)-1 \tag{D.49a}$$

$$= \left(1-\frac{\sigma^2}{2KJ}\right)^{-1}\left(1+\frac{s^2}{K}\right)-1, \tag{D.49b}$$

from which we get the same prediction $\sigma_c^2 = 2J(K-1)$ already obtained in [101].

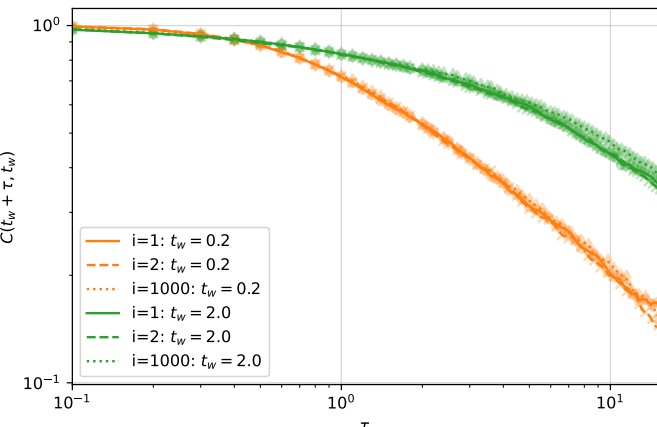

Figure 8: **Verification of site-independence of the correlation function in a homogeneous sparse network**. Disorder-averaged correlation functions for three representative nodes of a random regular graph (RRG) with $N = 1000$ nodes and fixed degree $K = 5$ are shown. The system is governed by a homogeneous coupling constant $J = 1.0$ and temperature $D = 0.3$. The dynamics were integrated using an Euler-Maruyama scheme. Each curve was averaged over $10^4$ independent realizations of the noise, coupling disorder, and uniform initial conditions. The near-perfect overlap confirms the assumption that, due to the spatial homogeneity of the RRG, both cavity and full correlation functions can be treated as site-independent in the thermodynamic limit.

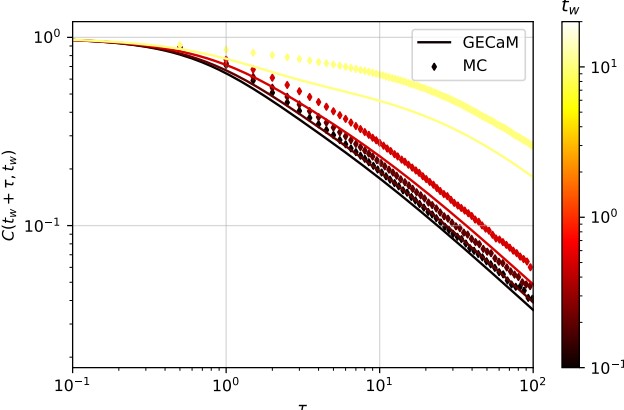

Figure 9: **Comparison between GECaM predictions and Monte Carlo simulations on a random regular graph**. The figure shows the correlation function $C(t_w + \tau, t_w)$ obtained from the numerical solution of the GECaM equations (solid lines) and from direct Monte Carlo simulations (dots) on a RRG with $N = 5000$ nodes, degree $K = 4$, coupling constant $J = 1.0$, and temperature $D = 0.3$. Although finite-size effects—present only in the Monte Carlo simulations—lead to deviations for large waiting times, the qualitative agreement between the two methods remains good, supporting the validity of the GECaM approach.

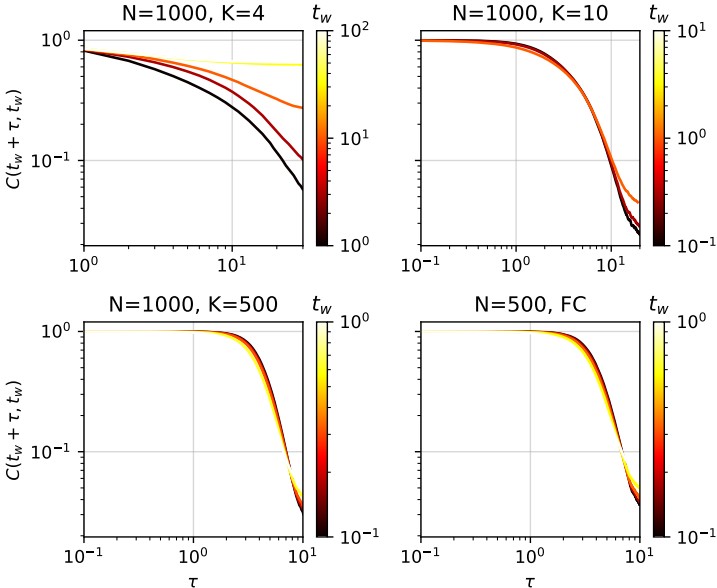

Figure 10: **Aging behavior in sparse spherical ferromagnets**. Correlation functions $C(t_w + \tau, t_w)$ obtained from Monte Carlo simulations on random regular graphs of varying size and connectivity are shown for the spherical ferromagnetic model with homogeneous coupling $J = 1.0$ and temperature $D = 0.3$. The presence of aging in the sparse regime ($K$ small) confirms the theoretical expectation that the GECaM equations also describe ferromagnetic dynamics in this limit. As the connectivity increases, aging disappears due to the scaling $J \sim 1/K$, in contrast to the disordered case where aging persists even in the dense limit.

# E  Additional numerical results on the spherical $p = 2$-spin model

In this section, we provide additional numerical results supporting the assumptions and claims discussed in the main text concerning the spherical 2-spin model introduced in section 3.5. The results presented below validate key hypotheses underlying the Gaussian Expansion Cavity Method (GECaM) and illustrate the qualitative agreement between analytical predictions and numerical simulations. To justify the assumption that cavity and full quantities in the GECaM framework are independent of node and edge indices, we studied the disorder-averaged correlation functions of three representative nodes in a random regular graph (RRG). As shown in Fig. 8, the correlation functions coincide within numerical accuracy, confirming that the dynamics is effectively homogeneous across the graph, even for finite-size systems. This result supports the validity of assuming site- and edge-independence in the thermodynamic limit. The data were obtained by simulating a system of $N = 1000$ nodes and fixed degree $K = 5$ with homogeneous coupling $J = 1.0$ and temperature $D = 0.3$, using Euler-Maruyama integration. Averages were taken over $10^4$ independent realizations of the noise, coupling disorder, and initial conditions. We then compared the GECaM numerical solutions described in the main text with MC simulations on a larger RRG with $N = 5000$ and degree $K = 4$ (see Fig. 9). While the agreement is not quantitative for large waiting times possibly due to finite-size effects present only in the MC data—an effect already discussed in section 3.2—the qualitative behavior is consistent across both methods. This further supports the correctness of the GECaM equations in capturing the aging dynamics of the system. Finally, we investigated whether aging behavior also appears in the spherical ferromagnetic model on sparse graphs. As shown in Fig. 10, MC simulations confirm the presence of aging in the sparse regime, with the effect

gradually disappearing as the degree $K$ increases. This is in agreement with the analysis in the main text: although the GECaM equations Eqs. (112), (114) and (115) are formally identical for the disordered and ferromagnetic cases, the required coupling scaling differs. In particular, the ferromagnetic model requires $J \sim 1/K$, so that in the dense limit aging vanishes due to the suppression of terms of order $J^2$. In contrast, the disordered case—with $J \sim 1/\sqrt{K}$—continues to exhibit aging even for large $K$, consistent with known results.

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
