# Peer review of "Gaussian approximation of dynamic cavity equations for linearly-coupled stochastic dynamics"

_SciPost Physics, doi:SciPost Phys. 19, 019 (2025)_

## Round 1 · Referee Report · Anonymous (Referee 1) · 2025-3-21

Strengths

1) The paper puts forward a new method to study the dynamics of sparse interacting systems of continuous degrees of freedom.

2) The paper presents detailed results for several interesting examples.

Weaknesses

1) The paper does not test the method on graphs with heterogeneous structures.

Report

In this work, the authors combine ideas from generating functional analysis with the local tree-like structure of sparse interacting systems to put forward cavity equations for the dynamics of systems described by coupled stochastic differential equations. By performing a small coupling expansion, they show that the cavity equations are solved by a Gaussian ansatz, yielding a system of closed dynamical equations for macroscopic parameters. These equations apply only to systems with linear interactions, while nonlinearities can, in principle, be incorporated via a perturbative approach. The method is then illustrated with four different examples of dynamical systems on regular random graphs.

The dynamical cavity method for sparse interacting systems with binary variables has been developed more than ten years ago. The present work proposes an interesting extension of the dynamical cavity method for systems modelled by continuous degrees of freedom. Even though the approach is restricted to linear dynamics, this work represents and important step towards a more complete understanding of the dynamical cavity method for coupled differential equations.

However, I would like to raise one important point. A key feature of the cavity approach is its versatility, as it allows to solve problems on graphs with heterogeneous structures, including random degrees and/or coupling strengths. In the present work, all examples discussed by the authors apply to regular random graphs, whose spectral density is analytically known. To make the approach more robust and general, I believe it would be valuable to test the method on a linear dynamical system with heterogeneous interactions, where the spectral density of the interaction matrix does not have an analytic closed form.

When resubmitting their manuscript, the authors should also address the following points and modify the paper where necessary:

1) I might have misunderstood figure 1, but it seems that the interactions between variables at different times do not correspond to the graphical structure shown. I don't see why the graph in figure 1 includes an interaction between $\hat{x}_{i}^n$ and $x_{i}^{n-1}$ at the previous time step, since this term is absent from eq. (5). The authors also mention on page 5 that the local tree-like structure of the graph is a consequence of the linear coupling between variables. I think it'd be important to elaborate further on this point and better explain why linear interactions allow one to disentangle the loopy structure.

2) Regarding eq. (17), I have the following issue. If one expands eq. (17) up to second order in $\alpha$, one should recover the expansion of eq. (6), but this does not seem to be the case. The reason is that the $O(\alpha^2)$ contribution coming from the term with $\mu_{k \setminus i}^n$ in eq. (17) does not appear to cancel out with the $O(\alpha^2)$ contribution in eq. (6). Please clarify this point.

3) On page 8 the authors mention that $B_{i \setminus j}(t,t^{\prime})$ is zero because the dynamics is causal. In the generating functional formalism, the moments of conjugate variables are zero because the generating functional is normalized. It would therefore be important to elaborate further on why causality implies that these quantities vanish in the present context.

4) In the long time limit, eq. (50) has a stable solution if all eigenvalues of $\boldsymbol{J}-\boldsymbol{\lambda}$ have negative real parts. Besides that, the spectral density of, for instance, a sparse and symmetric random matrix $\boldsymbol{J}-\boldsymbol{\lambda}$ typically has unbounded support for $N \rightarrow \infty$, which makes the linear dynamics described by eq. (50) unstable. It is thus important to make explicit the conditions on $\boldsymbol{J}-\boldsymbol{\lambda}$ that render the linear dynamics stable.

5) The authors mention on page 14 that the relation between random matrix theory and the Gaussian solution for the cavity equations is double-sided. Equations (45) and (48) for the response function are exactly the same as the cavity equations for the diagonal elements of the resolvent of sparse symmetric matrices (in this sense, it'd be instructive to cite a few relevant papers where these equations have been originally derived, e.g., works of Perez-Castillo, Rogers, Metz, Biroli, etc). On the other hand, the spectral density of $\boldsymbol{J}-\boldsymbol{\lambda}$ determines the correlation and response of the linear dynamical system, but this is only valid for graphical models whose spectral density has a bounded support (see point 4). Is it not correct to think that this restriction should also manifest itself in the Gaussian solution of the dynamical cavity equations?

6) In section 3.1, the authors solve the equations for the Laplace transform of the response and the correlation functions in the TTI regime, in the case of random regular graphs. Technically speaking, this problem is equivalent to solving the resolvent equations for RRG's, so this section does not present new material. It would be interesting to explore further the consequences of the solution. For instance, how do the behaviours of the two-point functions compare with those in the fully-connected case?

7) On page 24, it is not clear why one can assume that the local quantities are independent of the site index. Although the graph is regular, the coupling strengths are random variables, and local quantities should fluctuate from site to site. Please, clarify the status of the homogeneity assumption in this case.

8) Equations (108-111) are equivalent to those for a ferromagnetic model with $J_{ij} = J$. The solution of eqs. (108-111) exhibits aging, which shows that in the sparse regime there is no need for randomness in the coupling strengths to observe aging effects. I believe this is an interesting conclusion from eqs. (108-111). It would be also important to clarify the meaning of the fully-connected limit in this context, because the model is ferromagnetic for finite $K$, while we recover the Cugliandolo-Kurchan equations for $p=2$ spherical model with Gaussian couplings in the limit $K \rightarrow \infty$.

Minor comments:

a) After eq. (7), what do the authors mean by the term "quasi-probability distribution"?

b) In the text after eq. (9), the upper limit of the summation seems wrong.

c) Equation (10) is a functional. I suggest the authors replace the brackets $(\dots)$ by $[\dots]$, as this is the standard notation for the argument of a functional.

d) In eq. (54), I think it would be more consistent with the notation in the paper to use a different type of index to identify the eigenvalues.

e) Typo on page 14: "Hermitization method".

f) Typo on page 15: "proceed as follows".

g) Typo on page 23: "bimodal distribution" instead of "binomial distribution".

h) The sentence "We can therefore safely substitute $z$..." on page 13 is misleading, since we cannot compute the resolvent directly on the real line (the resolvent is singular at the eigenvalues). Note also that you keep a small imaginary part in eq. (54).

Recommendation

Publish (easily meets expectations and criteria for this Journal; among top 50%)

  • validity: high
  • significance: high
  • originality: high
  • clarity: good
  • formatting: excellent
  • grammar: good

Author:  Mattia Tarabolo  on 2025-05-19  [id 5493]

(in reply to Report 1 on 2025-03-21)
Category:
answer to question

Comment on heterogeneous networks

We thank the referee for this important suggestion. In the revised manuscript, we extend our analysis to heterogeneous interaction networks—Erdős-Rényi (ER) and Random Configuration Models (RCMs) graphs. We consider ferromagnetic couplings in all cases and also include bimodal disordered interactions specifically for the ER case.

This extension is significant, as only in the regular random graph with ferromagnetic couplings the interaction matrix has an analytical closed-form spectral density. In contrast, the spectral properties of the heterogeneous cases are not analytically tractable. We show that the Gaussian Expansion Cavity Method (GECaM) accurately reproduces equilibrium correlations and captures finite-size effects, confirming its applicability to a broad class of networks.

Major comments

1) I might have misunderstood figure 1, but it seems that the interactions between variables at different times do not correspond to the graphical structure shown. I don't see why the graph in figure 1 includes an interaction between $\hat{x}^n_i$ and $x^{n−1}i$ at the previous time step, since this term is absent from eq. (5). The authors also mention on page 5 that the local tree-like structure of the graph is a consequence of the linear coupling between variables. I think it'd be important to elaborate further on this point and better explain why linear interactions allow one to disentangle the loopy structure.

We thank the referee for pointing this out. Regarding the figure, the referee is correct; there was an inconsistency in the graphical structure, which we have now corrected in the revised version. As for the second part, we agree that our original wording may have been unclear. What we intended to convey is that, in the case of linear couplings, the disentanglment of the locally-loopy structure of the factor graph along a locally tree-like structure is straightforward. For nonlinear interactions—such as when the dynamics involves a term like $h\left(\sum_{j=1}^N a_{ij} J_{ij} x_j(t)\right)$ with $h(x)$ a generic nonlinear function—this decoupling is still in principle possible but requires the introduction of auxiliary fields. This significantly complicates the factor graph representation and the derivation of the dynamical cavity equations.

2) Regarding eq. (17), I have the following issue. If one expands eq. (17) up to second order in $\alpha$, one should recover the expansion of eq. (6), but this does not seem to be the case. The reason is that the $O(\alpha^2)$ contribution coming from the term with $\mu_{k\setminus i}^n$ in eq. (17) does not appear to cancel out with the $O(\alpha^2)$ contribution in eq. (6). Please clarify this point.

The key aspect here is that the correlation function $C_{i\setminus j}^{n,n'}$ in eq. (17) is defined as connected, i.e.,

$$C_{i\setminus j}^{n,n'} = \langle x_i^n x_i^{n'} \rangle_{i\setminus j} - \mu_{i\setminus j}^n \mu_{i\setminus j}^{n'}.$$
When expanding eq. (17) to second order in $\alpha$, two terms involving $\mu_{k\setminus i}^n$ appear, one from the connected correlation function (with sign $-$) and the other from the second order contribution of the integral involving $\mu_{k\setminus i}^n$ (with sign $+$). These terms cancel out, and the resulting expression matches the second-order expansion of eq. (6) after averaging over cavity messages.

3) On page 8 the authors mention that $B_{i\setminus j}(t,t')$ is zero because the dynamics is causal. In the generating functional formalism, the moments of conjugate variables are zero because the generating functional is normalized. It would therefore be important to elaborate further on why causality implies that these quantities vanish in the present context.

We agree that stating causality alone implies vanishing conjugate moments may be misleading. In the exact generating functional formalism, these moments vanish due to normalization. However, since cavity messages are derived from an approximation, we cannot rigorously prove their normalization. Instead, we show self-consistently that conjugate moments vanish under the assumption of causal dynamics. This argument would not hold for non-causal dynamics—e.g., observation-constrained dynamics as in Ref. [27]—where constraints introduce feedback from future times and the conjugate moments are indeed non-zero.

4) In the long time limit, eq. (50) has a stable solution if all eigenvalues of $J−\lambda$ have negative real parts. Besides that, the spectral density of, for instance, a sparse and symmetric random matrix $J−\lambda$ typically has unbounded support for $N \to \infty$, which makes the linear dynamics described by eq. (50) unstable. It is thus important to make explicit the conditions on $J−\lambda$ that render the linear dynamics stable

We now clarify in the manuscript that stability requires all eigenvalues of $J - \lambda$ to have negative real parts, and that the analysis is restricted to parameter regimes where this condition holds.

5) The authors mention on page 14 that the relation between random matrix theory and the Gaussian solution for the cavity equations is double-sided. Equations (45) and (48) for the response function are exactly the same as the cavity equations for the diagonal elements of the resolvent of sparse symmetric matrices (in this sense, it'd be instructive to cite a few relevant papers where these equations have been originally derived, e.g., works of Perez-Castillo, Rogers, Metz, Biroli, etc). On the other hand, the spectral density of $J−\lambda$ determines the correlation and response of the linear dynamical system, but this is only valid for graphical models whose spectral density has a bounded support (see point 4). Is it not correct to think that this restriction should also manifest itself in the Gaussian solution of the dynamical cavity equations?

Yes, this is correct. If the system is unstable, the TTI regime is never reached, and the Laplace-transformed GECaM equations are not defined. We clarify this point in the revised manuscript.

6) In section 3.1, the authors solve the equations for the Laplace transform of the response and the correlation functions in the TTI regime, in the case of random regular graphs. Technically speaking, this problem is equivalent to solving the resolvent equations for RRG's, so this section does not present new material. It would be interesting to explore further the consequences of the solution. For instance, how do the behaviours of the two-point functions compare with those in the fully-connected case?

We used our method to compute the equilibrium correlation functions for random regular graphs with ferromagnetic couplings and compared them with the fully-connected (FC) case. We observed that sparseness leads to a slower relaxation of the system. In particular, for small enough connectivity values, the decay of the correlation function deviates from a purely exponential form, exhibiting a fast initial decay followed by a slower relaxation tail. This illustrates how network topology—specifically, the finite connectivity—can qualitatively affect the dynamical behavior even in absence of disorder.

7) On page 24, it is not clear why one can assume that the local quantities are independent of the site index. Although the graph is regular, the coupling strengths are random variables, and local quantities should fluctuate from site to site. Please, clarify the status of the homogeneity assumption in this case.

This remark is certainly valid for a single instance, because of the random realization of couplings, and it could be relevant for ensemble averages. In order to obtain ensemble average results for sparse random regular graphs, that could be compared with classical results for fully-connected ones, we assumed that the disordered-averaged cavity response functions and cavity correlation functions are identical on every edge. This is reasonable because the 2-spin model does not show replica symmetry breaking—it is a disguised ferromagnet—and the aging behavior has a purely dynamical nature. In the manuscript we now state clearly this is an hypothesis. We checked through Monte Carlo simulations that the disorder-averaged correlation functions of different nodes are comparable, which suggests that our hypothesis is correct in the thermodynamic limit.

8) Equations (108-111) are equivalent to those for a ferromagnetic model with $J_{ij}=J$. The solution of eqs. (108-111) exhibits aging, which shows that in the sparse regime there is no need for randomness in the coupling strengths to observe aging effects. I believe this is an interesting conclusion from eqs. (108-111). It would be also important to clarify the meaning of the fully-connected limit in this context, because the model is ferromagnetic for finite $K$, while we recover the Cugliandolo-Kurchan equations for $p=2$ spherical model with Gaussian couplings in the limit $K\to\infty$.

The equations for a ferromagnet would be indeed the same as Eqs. (108-11). The catch is that while for the bimodal disordered case the coupling $J$ needs to scale as $1/\sqrt{K}$, for the ferromagnetic case it needs to scale as $1/K$. Hence, it is correct that the ferromagnetic model also shows the same sort of aging in the sparse regime (in fact, also the disordered case is just a disguised ferromagnet). In the large connectivity limit, instead, the ferromagnetic model does not show any aging because the terms of order $J^2$ vanish (because of the scaling $J\sim 1/K$), while the disordered model still preserves aging behavior as known from literature. We also verified numerically that this behavior is observed for the ferromagnetic model.

Minor comments

a) After eq. (7), what do the authors mean by the term "quasi-probability distribution"?

By quasi-probability distribution, we refer to a mathematical object similar to a probability distribution that relaxes some of Kolmogorov’s axioms. In our case, normalization holds, but non-negativity is not guaranteed—there may be regions where the function takes negative values.

b)–h)

We thank the referee for the suggestions and have accordingly corrected the manuscript.

---

## Round 1 · Referee Report · Anonymous (Referee 3) · 2025-4-2

Strengths

(i) The work expands the dynamical cavity formalism to systems with continuous degrees of freedom, pairwise (random) interactions, defined on sparse graphs. This framework includes a variety of systems of interest and allows for several applications and perspectives, which are mentioned in the work;

(ii) Despite its technical nature, the exposition remains clear and self-contained. The division of material between the main text and appendices makes the paper accessible and readable.

Report

The work provides a generalization of the dynamical cavity formalism to systems with continuous degrees of freedom and pairwise (random) interactions, defined on sparse graphs. Eq. (17) expresses the cavity marginals in terms of the cavity local averages, correlations, and responses, defined self-consistently as averages over the cavity marginals. This equation is derived by performing a second-order expansion in the coupling parameter α in the general Eq. (6). For linear forces and additive thermal noise, this expansion leads to Gaussian marginals, allowing the authors to derive equations for the cavity order parameters (mean, correlation, and response), which generalize the previously derived dynamical TAP equations for fully connected graphs. The paper also discusses a perturbative closure scheme to account for nonlinearity in the force or noise terms (which spoil the Gaussianity). The derivation is supported by a series of illustrative applications discussed in Sec. 3.

The content of this work is novel and potentially interesting for a rather broad audience. The content of the manuscript it technical, but I find the presentation of the material quite clear and self-contained.I recommend the publication of this work in SciPost.

A few minor questions or comments:

(i) In the derivation, Sec. 2.1, I would be more explicit on the precise reason why linear interactions allow for graphical model construction presented in the paper, maybe contrasting this with the more complicated situation that would occur for non-linear interactions ( this is briefly mentioned in the conclusions, bit I feel that this point should be stressed more in the first part of the paper).

(ii) In the abstract and introduction, the authors state that the formalism developed in this work can be applied to systems subject to global constraints, and in Sec. 3, they discuss the example of a spherical constraint. Could the authors comment on whether they could treat the case in which constraints are imposed uniformly on the continuous variables (for example, their positivity, as needed in applications to ecosystems dynamics), which are perhaps not straightforward to enforce using Lagrange multipliers?

(iii) In the conclusion, the authors claim that while extending their equations to include nonlinearities in interactions—such as those found in neural networks and learning models—is challenging and intricate, this is not the case for multi-body nonlinearities, such as those in p-spin models with p≥3. Could they provide some intuition for why this is the case and how general is this statement?

Requested changes

Suggested changes:

  1. In the introduction, it is mentioned that “DMFT has been successfully applied to study the dynamics of spherical p-spin models for aging and glassy dynamics in disordered systems”, and some references are mentioned. I feel that the work characterizing the aging solution in spherical p-spin models should be referenced to at this point:

Cugliandolo, L. F., & Kurchan, J. (1993). Analytical solution of the off-equilibrium dynamics of a long-range spin-glass model. Physical Review Letters, 71(1), 173.

  1. In the introduction, when referring to the characterization of dynamical phases in neural networks via DMFT, in addition to Ref. [38] the authors may consider the very recent work:

A Montanari, P Urbani, Dynamical Decoupling of Generalization and Overfitting in Large Two-Layer Networks, arXiv preprint arXiv:2502.21269

where transitions between different dynamical learning regimes are characterized via DMFT.

  1. Eq. (10), upper index of the integral should be “curly” T

Recommendation

Publish (easily meets expectations and criteria for this Journal; among top 50%)

  • validity: -
  • significance: -
  • originality: -
  • clarity: -
  • formatting: -
  • grammar: -

Author:  Mattia Tarabolo  on 2025-05-19  [id 5495]

(in reply to Report 3 on 2025-04-02)

Major comments

We thank the reviewer for suggesting additional references. We have carefully considered them and decided to include them in the revised manuscript.

Minor comments

1) In the derivation, Sec. 2.1, I would be more explicit on the precise reason why linear interactions allow for graphical model construction presented in the paper, maybe contrasting this with the more complicated situation that would occur for non-linear interactions (this is briefly mentioned in the conclusions, bit I feel that this point should be stressed more in the first part of the paper).

In the case of linear couplings, the disentanglement of the locally loopy structure of the factor graph into a locally tree-like structure is straightforward. For nonlinear interactions—such as terms of the form $h\left(\sum_{j=1}^N a_{ij} J_{ij} x_j(t)\right)$, with $h(x)$ a generic nonlinear function—this decoupling remains, in principle, possible but requires the introduction of auxiliary fields. This significantly complicates the factor graph representation and the derivation of the dynamical cavity equations.

2) In the abstract and introduction, the authors state that the formalism developed in this work can be applied to systems subject to global constraints, and in Sec. 3, they discuss the example of a spherical constraint. Could the authors comment on whether they could treat the case in which constraints are imposed uniformly on the continuous variables (for example, their positivity, as needed in applications to ecosystems dynamics), which are perhaps not straightforward to enforce using Lagrange multipliers?

In the manuscript, we focused on constraints that are either global but analytically tractable—such as the spherical constraint, which can be enforced via a Lagrange multiplier—or local constraints acting on individual nodes, as in observation-constrained dynamics. Uniform constraints such as positivity are more challenging to enforce within this framework. However, for models relevant to ecosystem dynamics, we believe that explicit enforcement of positivity is not strictly necessary. The presence of an absorbing state at zero ensures that variables initialized in the positive regime either remain positive or are absorbed at zero, which effectively constrains their dynamics without requiring hard constraints.

3) In the conclusion, the authors claim that while extending their equations to include nonlinearities in interactions—such as those found in neural networks and learning models—is challenging and intricate, this is not the case for multi-body nonlinearities, such as those in p-spin models with p≥3. Could they provide some intuition for why this is the case and how general is this statement?

In the case of multi-body nonlinearities, such as those arising in $p$-spin models with $p \geq 3$, the structure of the dynamical generating functional remains amenable to a factor graph representation. Specifically, one can write a factorized form of the dynamical partition function similar to Eq. (5), where the interaction term involves $p$ variables—for instance, $\hat{x}_i$, $x_{j_1},\dots,x_{j_{p-1}}$. This naturally leads to a bipartite graphical model in which factor nodes encode the $p$-body interactions. The resulting graph remains locally tree-like in sparse topologies, and a procedure analogous to that used in our work—based on the cavity method and a small-coupling expansion—can be applied. One thus obtains a closed set of equations for the local means, correlations, and responses.

---

## Round 1 · Referee Report · Anonymous (Referee 2) · 2025-4-2

Strengths

The approach is new and was tested non-only in cases where the solution is known, but it also demonstrates to outperform previous approaches for the BM in an homogeneous regular graphs

Weaknesses

All the model considered are simple, the interaction is always linear, and when applied to disordered models, one of the approximation is not obvious to me (see the report)

Report

Report: Gaussian approximation of dynamic cavity equations for linearly coupled stochastic dynamics

I read the paper carefully and followed the derivations (without looking for typos) up to page (15). Then I found the discussion of the results reasonable and consistent.

The manuscript is very well written and represents an original contribution to the field. The authors studied the dynamics of continuos variables in Random Regulars graphs using the simplest possible kind of intereacting models, linear interactions. They also introduced non-linearities through perturbation theory.

Besides reproducing known results from the literature, Figure 3 shows the relevance of the approach. For the Bouchaud-Mezard model, the computation of the authors clearly outperform previous approaches coinciding with the simulations.

Some points to be clarified:

In section 3.4, the authors assume that the cavity quantities do not depend on the edjes. This is not trivial at all due to the disorder, specially for small K. The authors should clarify on this.

I suggest the authors to introduce a short discussion in the introduction making explicit the difference between their approach and the ones in [44] and [48].

After these minor modifications the paper should be published.

Requested changes

1- In section 3.4, the authors assume that the cavity quantities do not depend on the edjes. This is not trivial at all due to the disorder, specially for small K. The authors should clarify on this.

2- I suggest the authors to introduce a short discussion in the introduction making explicit the difference between their approach and the ones in [44] and [48].

Recommendation

Publish (easily meets expectations and criteria for this Journal; among top 50%)

  • validity: good
  • significance: good
  • originality: high
  • clarity: top
  • formatting: excellent
  • grammar: excellent

Author:  Mattia Tarabolo  on 2025-05-19  [id 5494]

(in reply to Report 2 on 2025-04-02)
Category:
answer to question

Comments

1) In section 3.4, the authors assume that the cavity quantities do not depend on the edjes. This is not trivial at all due to the disorder, specially for small $K$. The authors should clarify on this.

This remark is certainly valid for a single instance, because of the random realization of couplings, and it could be relevant for ensemble averages. In order to obtain ensemble average results for sparse random regular graphs, that could be compared with classical results for fully-connected ones, we assumed that the disordered-averaged cavity response functions and cavity correlation functions are identical on every edge. This is reasonable because the 2-spin model does not show replica symmetry breaking—it is a disguised ferromagnet—and the aging behavior has a purely dynamical nature. In the manuscript we now state clearly this is an hypothesis. We checked through Monte Carlo simulations that the disorder-averaged correlation functions of different nodes are comparable, which suggests that our hypothesis is correct in the thermodynamic limit.

2) I suggest the authors to introduce a short discussion in the introduction making explicit the difference between their approach and the ones in [44] and [48].

In short, our method is developed as a small-coupling expansion at the single-disorder-instance level. This distinguishes it from [44], which performs a large-connectivity (i.e., small-coupling) expansion on the disorder-averaged problem. Our approach retains sample-specific information and is thus more general, allowing it to be employed as an algorithm capable of capturing finite-size effects and node-level fluctuations. In addition, while the DMFT framework proposed in [48] is designed for directed sparse graphs with nonlinear interactions, our method can deal with both directed and undirected graphs with linear interactions. Furthermore, our formulation provides a closed set of equations for the average quantities (means, response functions, and correlations), whereas [48] relies on a population dynamics algorithm to compute disorder-averaged observables.

---

## Round 2 · Referee Report · Anonymous (Referee 1) · 2025-5-31

Report

The authors have addressed all my previous comments, and the
paper has been revised accordingly. In particular, the resubmitted
version includes results for the correlation function in the case of
sparse heterogeneous graphs, offering an even more
compelling demonstration of the potential applications of the
Gaussian cavity method.

This paper develops an efficient approach to studying the dynamics
of coupled differential equations on sparse random graphs, with
illutrative applications in different models. Although the method is
currently restricted to linear interactions, the authors introduce a
valuable and original theoretical framework for analyzing the dynamics
of continuous systems on complex networks. I therefore recommend
the paper for publication in SciPost Physics.

Recommendation

Publish (easily meets expectations and criteria for this Journal; among top 50%)

---

## Round 2 · Referee Report · Anonymous (Referee 3) · 2025-6-4

Report

The authors have addressed my questions and they have improved the clarity of the exposition, in particular regarding its connection to previous works. In my opinion, the current version of the manuscript meets the standards for publication in SciPost.

Recommendation

Publish (easily meets expectations and criteria for this Journal; among top 50%)

---

## Round 2 · Author Response

We thank the editors and reviewers for their careful reading of our manuscript and for the constructive feedback they provided. Their comments have led to a significant improvement of both the clarity and the scope of our work.

In response to the reports, we have revised the manuscript accordingly. The new version includes extensions of the original analysis, additional numerical results, clarifications of key assumptions, and several textual and graphical improvements. These changes are aimed at enhancing the accessibility and rigor of our presentation.
We believe that the revised manuscript now better communicates the ideas and results of our study, and we hope it will be of interest to researchers working on stochastic processes, complex networks, and disordered systems.

We gratefully acknowledge the valuable input of the reviewers and the editorial team in the preparation of this revised version.

---

## Round 2 · List of Changes

1) Added a comparison with related works in the introduction (page 4), specifically discussing Refs. [47] and [51]. 2) Corrected the left panel of Figure 1. 3) Clarified the role of nonlinear interactions in the derivation of the equations at the end of page 6. 4) Standardized the notation for functionals by consistently using square brackets instead of round brackets. 5) Fixed various typographical errors throughout the manuscript. 6) Explained more clearly why hatted averages are expected to vanish, at the end of page 8. 7) Included the derivation of the equilibrium equations within the Gaussian Expansion Cavity Method (GECaM) framework at the end of page 13. 8) Clarified that stable solutions for linearly coupled Ornstein-Uhlenbeck processes exist only when all eigenvalues of the interaction matrix have negative real parts, and stated explicitly that this assumption is adopted throughout the paper (page 14, between Eqs. (54) and (55)). 9) Removed the sentence “We can therefore safely substitute z with his real part x = \text{Re}(z)” preceding Eq. (57). 10) Revised the notation in Eqs. (57) and (58) by using different indices for the eigenvalues. 11) Rewrote the introduction of Section 3 to improve clarity and context. 12) Added a quantitative comparison between equilibrium correlation functions obtained using GECaM and those obtained from Monte Carlo (MC) simulations on Random Regular Graphs (RRGs) with varying degrees. These results are also compared with the Fully Connected (FC) regime. The discussion is included at the end of Section 3.1 (page 19), and the data are presented in the new Figure 2. 13) Introduced a new Section 3.2, where GECaM is applied to study linear dynamics with thermal noise on heterogeneous graphs. This section examines the effects of topological heterogeneity and finite-size corrections. The results are summarized in the new Figure 3. 14) Justified the assumption of homogeneity in the analysis of the 2-spin model on RRGs, at the end of page 27. 15) Added a comment at the end of Section 3.5 pointing the reader to the Supplemental Material, where further numerical results on the 2-spin model are provided. These include a comparison between GECaM predictions and MC simulations, and validation of the homogeneity assumption. 16) Added a code availability statement in the Acknowledgments section. 17) Included additional references. 18) Added Section S2 in the Supplemental Material, providing a detailed derivation and implementation strategy for the equilibrium GECaM equations. 19) Added Section S5 in the Supplemental Material, presenting additional numerical results for the spherical 2-spin model. These include: (i) MC validation of the homogeneity assumption on RRGs (Figure 8); (ii) a direct comparison between MC and GECaM results (Figure 9); and (iii) MC simulations of the spherical ferromagnet (Figure 10), which indicate the presence of aging for small degree and its disappearance at higher connectivity.

---

## Editorial Decision

published